# Hierarchical Li electrochemistry using alloy-type anode for high-energy-density Li metal batteries

Jiaqi Cao[1,3], Yuansheng Shi [ID][1,3], Aosong Gao[2], Guangyuan Du[1], Muhtar Dilxat[1], Yongfei Zhang[1], Mohang Cai[1], Guoyu Qian [ID][1], Xueyi Lu[1], Fangyan Xie[2], Yang Sun [ID][1] & Xia Lu [ID][1] ✉

Exploiting thin Li metal anode is essential for high-energy-density battery, but is severely plagued by the poor processability of Li, as well as the uncontrollable Li plating/stripping behaviors and Li/electrolyte interface. Herein, a thickness/capacity-adjustable thin alloy-type Li/LiZn@Cu anode is fabricated for high-energy-density Li metal batteries. The as-formed lithophilic LiZn alloy in Li/LiZn@Cu anode can effectively regulate Li plating/stripping and stabilize the Li/electrolyte interface to deliver the hierarchical Li electrochemistry. Upon charging, the Li/LiZn@Cu anode firstly acts as Li source for homogeneous Li extraction. At the end of charging, the de-alloy of LiZn nanostructures further supplements the Li extraction, actually playing the Li compensation role in battery cycling. While upon discharging, the LiZn alloy forms just at the beginning, thereby regulating the following Li homogeneous deposition. The reversibility of such an interesting process is undoubtedly verified from the electrochemistry and in-situ XRD characterization. This work sheds light on the facile fabrication of practical Li metal anodes and useful Li compensation materials for high-energy-density Li metal batteries.

Since their first commercialization in the 1990s, lithium-ion batteries (LIBs) with high safety endowed by the unique Li⁺ extraction/insertion mechanism, have established the dominant role in the energy storage market for more than 30 years[1–3]. However, although significant optimization has been put into practice, a grand gap still exists between the energy density of "rocking-chair"-type LIBs and the requirement of emerging industries[4–6]. Regarding the ultimate choice for a new-generation high-energy-density system, the metallic lithium (Li) is remarkable for its high capacity (10 times higher than the graphite anode), the lowest redox potential (−3.04 V vs. SHE), and light mass density (0.534 g cm⁻³)[7,8]. While, the Li-based battery chemistry involves repeatedly Li plating and stripping processes, where the Li metal anode (LMA) will inevitably trigger the dendrite-like deposition and uncontrollable volume change due to the imperfect substrate and "frameless" nature, forestalling its further progress from the

laboratory to the commercialization[9,10]. To date, several effective strategies have been proposed to develop the practical LMA or LMA-based composites for high-energy-density LIBs. For example, encapsulating metallic Li into three-dimensional (3D) hosts with high specific surface area and enough pores is a feasible way to reinforce the Li plating/stripping reversibility and volume stability[11–13]. Moreover, rationally designing electrolyte composition (e.g. the high-concentration Li salts and novel additives) or introducing artificial solid electrolyte interphase (SEI) can upgrade the robustness of the Li/electrolyte interface as an indirect strategy to confine the LMA and Li dendrite growth[14–17]. Nevertheless, almost all modifications of LMAs are based on thick Li foils (>100 μm, corresponding to an areal capacity of >20.5 mAh cm⁻²) or thick 3D hosts[18]. Despite the better cycling performance they have achieved, the overstock LMAs can not reasonably match the capacity of the current commercial cathodes (e.g.

[1]School of Materials, Sun Yat-sen University, Shenzhen 518107, PR China. [2]Instrumental Analysis & Research Center, Sun Yat-sen University, Guangzhou 510275, PR China. [3]These authors contributed equally: Jiaqi Cao, Yuansheng Shi. ✉e-mail: luxia3@mail.sysu.edu.cn

3 mAh cm$^{-2}$) in a low N/P ratio (<3), resulting in a waste of most Li sources with compromising the energy density[19]. Furthermore, the thick modified LMAs, as infinite Li sources, can not reflect the effectiveness of the improvement strategies actually due to the constant replenishment of the Li loss during cycling. Therefore, of special importance is how to fabricate the thin LMAs (e.g. <50 μm) to better match the commercial cathodes with appropriate N/P ratios[18,20,21]. Generally, the Li foils (50 to hundreds of microns) are fabricated by mechanical rolling in industry, but some problems are deep-rooted in the metallic Li when it withstands mechanical deformation. Whereby the strong influence of diffusion creep caused by high homologous temperature, the metallic Li becomes highly sticky and poor mechanical processibility, posing great challenge in producing thin LMAs[22,23]. In addition, it is easy to generate mechanical scratches on the soft Li surface during the rolling process, resulting in the Li$^+$ ions tending to deposit at these uneven protrusions. Thus, when preparing Li anode with a thickness of less than 50 μm, the requirements for the accuracy of the rolling machine and preparation cost and difficulty will increase sharply.

As a matter of fact, reprocessing Li metal upon its liquid state is a promising orientation to prepare Li composite anodes, such as infusing molten Li into 3D hosts to reshape the Li morphology[24,25]. At this point, the thickness of the Li composite anode is closely related to that of the 3D host. By controlling the thicknesses of the 3D hosts, the thin carbon nanotube-based Li-MnO$_x$/CNT (10 μm) and reduced graphene oxide-based Li@eGF (0.5–20 μm) with the limited Li sources can be readily prepared[26,27]. However, the preparation for the thin 3D hosts is complex and high-cost. Another feasible and cost-effective way is directly reshaping the thickness of Li at liquid state. Unfortunately, the high surface tension of molten Li hampers the Li spreading away on the substrate (i.e. bad Li wettability) and makes it difficult to keep the thickness of the molten Li. Once the external force is lost, the flattened molten Li will be spherical again, which is the shape with minimum surface energy under high surface tension[18]. To lower the surface tension and improve the Li wettability, a viable strategy is weakening the interior atomic interaction by doping heteroatoms to form the Li-M alloy phases (M: Ag, Sn, Mg, Al, In, etc.)[28,29]. These as-formed Li-M alloy phases possess a high affinity with metallic Li and rapid Li$^+$ ion diffusion coefficients, which could provide fast charge transfer kinetics and effectively regulate Li nucleation/deposition behaviors to promote electrochemical performance[15,30–33]. For instance, by adding Sn, In, or Mg into molten Li, the formed Li-M alloys with thin thicknesses could firmly adhere to many substrates (Cu foil, stainless steel, Ti foil, glass, Kapton film, Garnet solid-state electrolyte, and so on) via the direct contact between the alloy and these substrates[24,29,34,35]. While, the resultant thin Li-M alloy anodes can still not achieve good uniformity and variable thickness, not to mention the scale-up production. Accordingly, it is highly essential for high-energy-density LMAs or Li-M alloy anodes with adjustable thickness, well-guided Li deposition behaviors, and optimized electrochemical performance.

In terms of Li-M alloys, the Zn equips with unique traits, such as chemical stability in the ambient environment (vs. Na, Ca, Mg, etc.), nontoxicity, environmental friendliness (vs. Hg, Cd, etc.), low cost (vs. n, Ag, Sn, Ga, Au, etc.), and lower melting temperature of ~161 °C when the Zn atom concentration is 4.5% (>180 °C of similar Li-Al alloy)[36,37], which corresponds to a decreased safety concern during the processing of Li-Zn alloy. In addition, the as-formed LiZn alloy can also provide a relatively high Li$^+$ diffusion coefficient to ensure excellent charge transfer kinetics[31]. In this work, the spontaneously formed LiZn and Li$_2$ZnCu$_3$ alloying composites are fabricated by introducing the Zn into the molten Li on the Cu substrate (Li/LiZn@Cu) as LMA for LIBs. The results demonstrate that the molten Li-Zn mixture can consecutively spread away over 100 cm$^2$ on the Cu substrate and its thickness is adjustable (5–48 μm, corresponding to 0.89–8.7 mAh cm$^{-2}$) via the doctor-blade assistance at 200 °C. Such excellent

wettability is attributed to the spontaneous formation of LiZn and Li$_2$ZnCu$_3$ alloys, giving a thermodynamically-favored driven force for Li diffusion. Moreover, electrochemical tests indicate that the Li stripping/plating behaviors are significantly improved by the as-formed LiZn alloys due to the fine Li affinity, low barrier energy for Li diffusion, and hierarchical Li electrochemistry, which substantively contributes to the dendrite-free morphology and stabilizes the Li/electrolyte interface. The resultant double-layer pouch cell delivers a high energy density (284.0 Wh kg$^{-1}$, expecting a rise to 366.5 Wh kg$^{-1}$ at 10 stacked layers) under lean electrolyte (1.83 g Ah$^{-1}$) and low N/P ratio (1.35) conditions.

## Results
### Fabrication of Li/LiZn@Cu anode
The fabrication routine for the thin Li/LiZn@Cu anode is illustrated in Fig. 1a. Firstly, the precursor of the molten Li-Zn mixture (Li/LiZn) is fabricated by co-heating the mixed Li and Zn foils (mass ratio: 4:1) under 250 °C in a glovebox. After a thoroughly stirring process, the precursor is transferred onto a Cu foil surface under 200 °C. It is found that with regard to the strong surface tension in the molten Li (difficulty to expand), the molten Li-Zn mixture can spread easily on the Cu foil, where the additive of the metallic Zn seems to be improving the multiple interface contacts among the Li|Li-Zn|Cu system as shown in Fig. 1b, c. Therefore, it is effortless to reshape the thickness of the molten Li-Zn layer on the Cu substrate using a doctor blade. As shown in Fig. 1d, the average thickness of the as-obtained Li/LiZn@Cu anode is around 35 μm (Li/LiZn layer: ~25 μm; Cu substrate: ~10 μm) with a dense Li/LiZn and Cu interface structure as are confirmed by scanning electron microscope (SEM) images in Fig. 1e, f and Supplementary Fig. 1. As shown in Supplementary Fig. 2, such a thin Li/LiZn@Cu anode delivers an appropriate specific capacity of 4.4 mAh cm$^{-2}$, corresponding to the volumetric and gravimetric energy densities of ~3100 mAh g$^{-1}$ and 1760 mAh cm$^{-3}$, which are ca. 8 and 2 times higher than the commercial graphite anode as shown in Supplementary Table 1, respectively[18]. What is more, the thickness of the cast Li/LiZn layer is adjustable (ranging from 5 to 48 μm, corresponding to the capacity from 0.89 to 8.7 mAh cm$^{-2}$), the available capacities of which are compatible with most of the cathodes as shown in Supplementary Figs. 3–5. On the other hand, even using the 6 μm Cu foil, the fabrication of the thin Li/LiZn@Cu anode is also successful, which sheds new light on further reducing the inactive mass and thereupon increasing both volumetric and gravimetric energy densities as shown in Supplementary Figs. 6–8. Moreover, this thin Li/LiZn@Cu anode is more flexible with respect to the Li foil that can be steadily extended to the flexible energy storage devices as shown in Supplementary Fig. 9. Of special interest is that as shown in Fig. 1g, it is easy to cast a thin Li/LiZn@Cu anode with an of 17 × 6 cm$^{-2}$ in area to demonstrate the potential scale-up for LIBs industrialization application.

Regarding the bare Cu surface as shown in Supplementary Fig. 10, the island-like morphology of the Li/LiZn@Cu anode is displayed where numerous nanorod-like and nanodot-like structures are distributed randomly onto the flat Li/LiZn@Cu substrate as shown in Fig. 1h, i. Actually, the interaction between Li and Zn is the reconstitution alloying reaction, which means that the Li/LiZn can be regarded as a mixture of the intermetallic compound LiZn and metallic Li at a concentration range of Li atoms from 50% to 99%[38]. Here, the atomic concentration of Li in Li/LiZn is ~97.3%. Thus, the nanorod- and nanodot-like structures mainly correspond to the LiZn alloy skeletons as shown in Fig. 1j, which is verified by the energy dispersive X-ray spectroscopy (EDS) measurement of the Li/LiZn@Cu surface, presenting the concentrative distribution of the Zn element around the nanostructures. As shown in Supplementary Fig. 11, such nanostructures are buried by the metallic Li, forming a 3D composite Li anode supported by the alloy skeletons. Interestingly, the Cu element also exists on the nanostructures, where the atomic concentration is

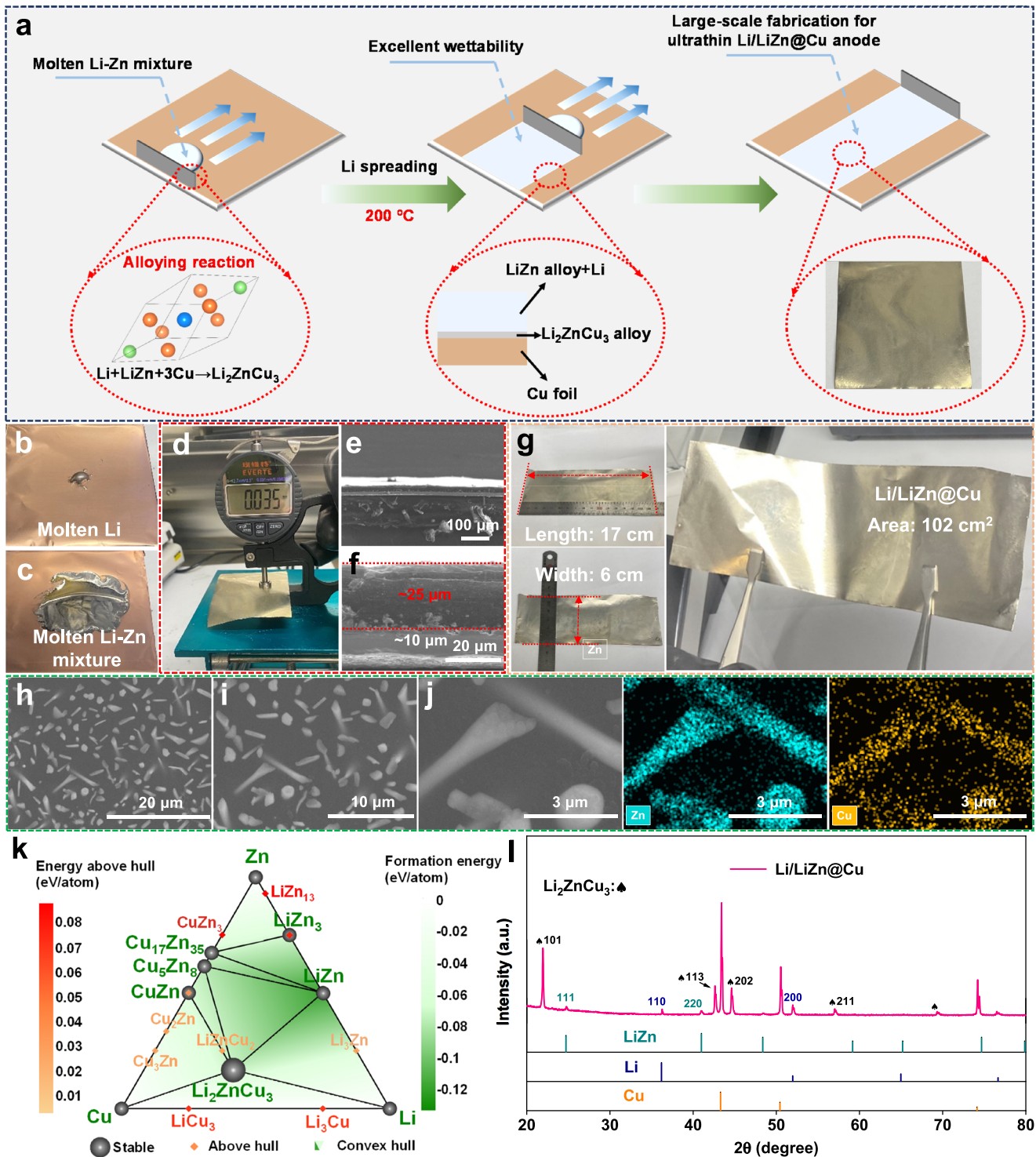

**Fig. 1 | Fabrication of Li/LiZn@Cu anode. a** Schematic illustration of the fabrication process. The Li wettability for (**b**) molten Li and (**c**) molten Li-Zn mixture on Cu foils. **d** Thickness, **e**, **f** cross-section SEM images, **g** size, **h**, **i** top-view SEM images, **j** corresponding EDS mappings of the thin Li/LiZn@Cu anode. **k** The calculated ternary phase diagram of the Li-Zn-Cu system at 473 K. **l** XRD pattern of the thin Li/LiZn@Cu anode.

10.91% as listed in Supplementary Table 2. As shown in Fig. 1k and Supplementary Figs. 12–14, the simulated ternary phase diagram and chemical potentials of the Li-Zn-Cu system forecast that the LiZn alloy, Li, and Cu substrate would re-alloy and convert into the rarely reported $Li_2ZnCu_3$ alloy, which is an energy-favorable phase at a wide temperature range. Thus, these nanostructures contain LiZn and a small amount of $Li_2ZnCu_3$ (~4.1%). As shown in Fig. 1l and Supplementary Figs. 15 and 16, the X-ray diffraction (XRD) peaks indicate the

coexistence of LiZn alloy (PDF#03-0954), metallic Li (PDF#15-0401), Cu (PDF#04-0836) substrate and the newly formed $Li_2ZnCu_3$ alloy in the Li/LiZn@Cu anode. Excepting these peaks, no obvious metallic Zn peaks are detected to indicate the thorough Li and Zn alloying process. Of special importance is that the $Li_2ZnCu_3$ alloy mainly exists at the interface between the Li/LiZn layer and Cu substrate as shown in Supplementary Fig. 17. Such an existent $Li_2ZnCu_3$ alloy can further help the fast spreading of the molten Li-Zn mixture on the Cu substrate.

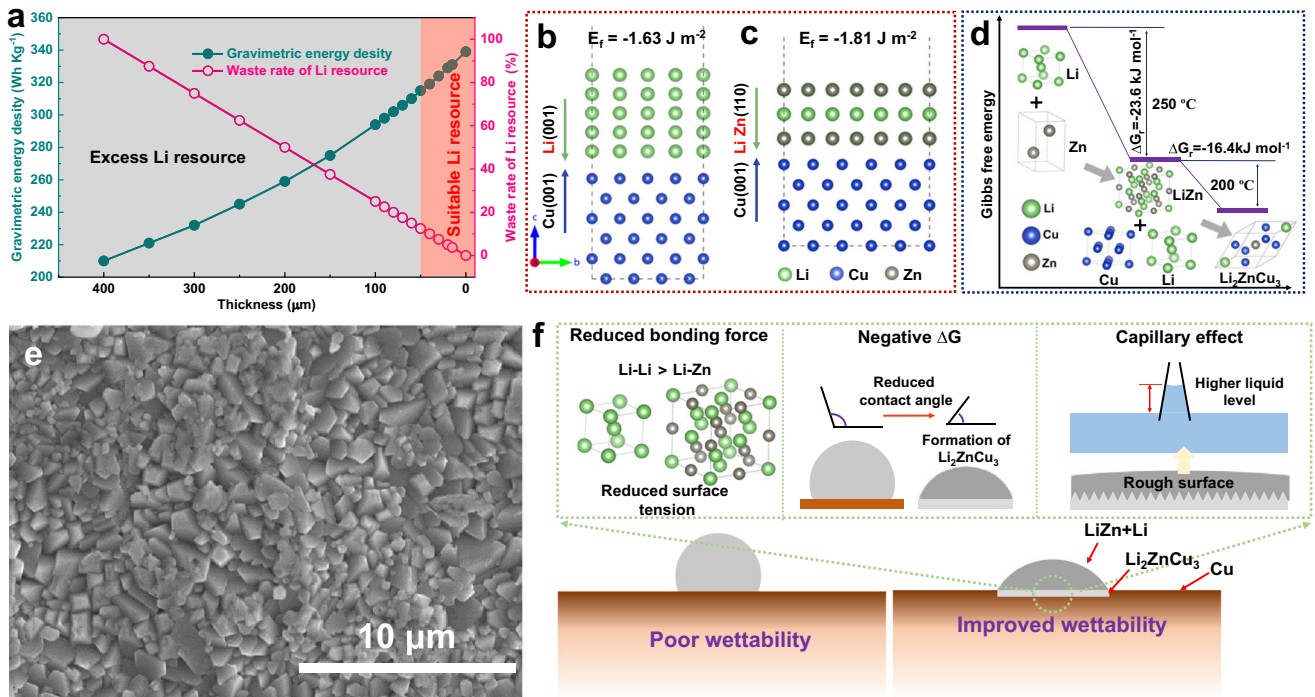

**Fig. 2 | Economic Li composite anode by alloying. a** Calculated gravimetric energy density and waste rate of Li resource of pouch cell with different thicknesses of Li metal anodes. The obtained data is based on Supplementary Table 4. Formation energies of the (**b**) Cu|Li and (**c**) Cu|LiZn interfaces. **d** Schematic illustration of the formation of LiZn and Li$_2$ZnCu$_3$ alloys and their corresponding Gibbs free energy (ΔG$_r$). **e** top-view SEM image of the Li$_2$ZnCu$_3$ alloy. **f** Factors influence the Li wettability with the LiZn and Li$_2$ZnCu$_3$ on the Cu substrate.

This multicomponent mixed Li/LiZn@Cu film shows a unique composite structure, functioning not only as the Li nondirectional disposition and nucleation sites to minimize the influence from the notorious dendrites, also as the Li replenishment and framework to contribute the chemical/electrochemical stability upon cycling.

## Economic Li composite anode by alloying

Figure 2 and Supplementary Table 4 display the possible waste of Li source at the increased thickness, and the design strategy of Li alloy-based anodes for Li metal batteries. As a matter of fact, the energy density of the resultant pouch cell decreases as well if the excessive Li can not reasonably match the capacity of the current commercial cathodes as shown in Fig. 2a. Hence, developing the thin Li-based anodes contributes not only to the highly anticipated high energy density, but also to the economic Li utilization for LIBs. To this end, the thin Li-Zn alloys, not the pure Li metal due to the frameless of Li itself are proposed here as a practical anode at a prerequisite of well understanding the Li wettability and alloying mechanisms.

Substantially, the wettability of liquid towards the substrate mainly involves two factors, including the surface tension of molten Li and the surface texture of the substrate. Firstly, the aforementioned LiZn alloys can effectively reduce the surface tension of the molten Li due to the weaker interaction of Li-Zn with regard to the Li-Li bonds, visibly augmenting the Li wettability on the substrate. The alloying reaction between Li and Zn at 250 °C is as follows:

$$\text{Li} + \text{Zn} \rightarrow \text{LiZn}, \Delta G_r = -23.6\,kJ\,mol^{-1}\,(-0.246\text{eV/f.u.}) \quad (1)$$

Note that the released Gibbs free energy (ΔG$_r$) at high temperatures is estimated using the machine learning model SISSO (sure independence screening and sparsifying operator) approach[39]. Compared with the Li-Li bond, the bonding force of Li-Zn is smaller, resulting in the reduction of the interior atomic interactions and surface tension for molten Li to effectively improve the Li wettability on

the substrate, consistent with the reported results[28]. Therefore, upon preparation, the molten Li could in-situ react with Zn to form the desired LiZn alloy, which in turn accelerates the homogeneous spread of excessive Li on the surface of Cu foil. Then, the partial binary LiZn alloy further reacts with the Cu and excessive Li to generate the ternary Li-Zn-Cu alloy. As shown in Fig. 2b, c, f, the formation energies of Cu|Li and Cu|Li-Zn interfaces are determined to be *ca.* −1.63 and −1.81 J m$^{-2}$, respectively, indicating the enhanced chemical contacts between Cu and Li-Zn with respect to that between Cu and Li[40]. As shown in Fig. 2d, this improved Cu|Li-Zn interface contact helps the subsequent alloying process of LiZn, Cu, and excessive Li at 200 °C to generate the ternary Li$_2$ZnCu$_3$ phase as follows:

$$\text{Li} + \text{LiZn} + 3\text{Cu} \rightarrow \text{Li}_2\text{ZnCu}_3, \quad \Delta G_r = -16.4\,kj\,mol^{-1}\,(-0.171\text{eV/f.u.}) \quad (2)$$

In fact, this exothermal reaction among the molten Li, LiZn alloy, and Cu is conducive to further improving the Li wettability on the substrate upon the alloying process.

From a thermodynamic point of view, the Li wettability can be determined by the contact angle using the following formula[41,42]:

$$\cos\theta^1 = \cos\theta^0 - \Delta\gamma_{sl}/\gamma_{gl} - \Delta G_r/\gamma_{gl} \quad (3)$$

in which the θ$^1$ and θ$^0$ denote the contact angle after or before the reaction, the Δγ$_{sl}$ is a change in solid-liquid interfacial energy by the reaction, and the γ$_{gl}$ is the gas-liquid interfacial energy. The smaller the θ, the better the wettability[43]. Thus, the exothermal alloying (negative ΔG$_r$) of Li-Zn and Li$_2$ZnCu$_3$ could decrease the θ to enhance the Li wettability on Cu foil, providing an extra driven force for the molten Li spreading, in line with previous reports[29,43]. More significantly, the formation of the Li$_2$ZnCu$_3$ alloy can transform the morphology of Cu foil and significantly improve its roughness as shown in Fig. 2e. The contact between the molten Li-Zn mixture and Cu substrate can be

regarded as a Wenzel state due to the compact interface within the Li/LiZn layer and Cu in Li/LiZn@Cu, which can further improve the lithiophilic surface wettability by the roughness increasing[44]. Such a pyramidal-shape morphology of $Li_2ZnCu_3$ alloy demonstrates the abundant nano-structures to form the effective capillary effect, which also contributes to the Li wettability by Laplace pressure ($\Delta P$; $\Delta P = 2\sigma/r$, where the $\sigma$ is the surface tension of molten Li and the $r$ refers to the capillary radius), as shown in Fig. 2e, f[15]. Under the above synergetic effects, the contact angle of ~62° between molten Li-Zn mixture and Cu substrate is much smaller than that of ~110° between molten Li and Cu substrate as shown in Supplementary Fig. 18.

After all, the Li wettability process of the Li-Zn mixture towards the Cu substrate is, to our best, much different from the other reported Li-M mixtures[23,29,38]. Excepting for the reduced surface tension by Li-M bond formation, the improved Li wettability of the Li-Zn mixture also involves the formation of ternary $Li_2ZnCu_3$ alloy via a re-alloying reaction to further facilitate Li spreading on Cu substrate, where the accompaned exothermal reaction and unique morphology will synergistically contribute to Li wettability as shown in Fig. 2f. This wettability process offers a probability to ensure a uniform and thickness-adjustable Li/LiZn layer on Cu substrate for the desired Li deposition/stripping process.

## Characterizations on Li plating/stripping

The lower Li adsorption energy and diffusion barrier on the substrate are essential for homogeneous Li deposition and subsequent Li dissolution. When the substrate/electrode surface has more negative adsorption energy towards $Li^+$ ion (i.e. excellent lithiophilicity), the Li nuclei sites will be more easily formed to promote the Li deposition. Otherwise, the Li may be dissolved again into the electrolyte, resulting in the heterogeneous Li nucleation/deposition morphology[46]. Therefore, uniform Li deposition behaviors could be expected if the surface lithiophilic sites are distributed homogeneously. The intrinsic Li diffusion behaviors of the Li/LiZn@Cu anode are first visited through DFT calculations, where the stabilized Li (100) and LiZn (110) surfaces are established to comprehend the Li plating behaviors on Li and LiZn[47–50]. The possible adsorption sites on the Li(100) and LiZn (110) surfaces are demonstrated in Supplementary Figs. 19 and 20 (the atomic coordinates of one $Li^+$ ion absorbed at the LiZn(110) and Li(100) surfaces are provided in Supplementary Data 1 and 2). As shown in Fig. 3a, k, the Li absorption energy of the LiZn (110) surface is −2.07 eV, much lower than the −1.50 eV of Li(100) surface, indicating the $Li^+$ ions would preferentially nucleate on the LiZn alloy. In detail, the absorption of Li atoms around the Zn sites exhibits much lower adsorption energies, which probably dominate the Li plating process on the LiZn (110) surface. As the deposition proceeds, the randomly distributed lithiophilic LiZn alloy particles as shown in Fig. 1h–j provide a chance of Li homogeneous deposition to suppress the oriented growth of Li dendrites. Moreover, the Li diffusion barrier along the LiZn (110) surface is ~0.03 eV, only half of that along the Li(100) surface as shown in Fig. 3b, c, k, although such diffusion barriers imply the fluid-like Li diffusion behavior. But, a lower Li diffusion barrier can facilitate the lateral Li transportation to deposit the Li in the vicinity of the substrate rather than the protuberances, mitigating the dendrite growth[46]. Additionally, given that the presence of a small amount of $Li_2ZnCu_3$ alloy in the Li/LiZn layer and its superior lithiophilicity, it will also exert positive influences on Li plating behaviors as shown in Supplementary Table 5 and Fig. 21. As shown in Fig. 3d–i, k, from the SEM images of Li/LiZn@Cu after quantitative Li plating, at a Li deposition of 0.5 mAh cm$^{-2}$, the homogeneous Li nuclei appear on the LiZn alloy surface and further at 1 and 3 mAh cm$^{-2}$, a dense and smooth Li layer is planted on the Li/LiZn@Cu anode without obvious dendrite growth, in sharp contrast to the Li@Cu anode as shown in Supplementary Fig. 22a, b.

Furthermore, the in-situ XRD characterization furnishes an in-depth understanding of the Li stripping/plating behaviors in the Li/LiZn@Cu anode. Due to the excellent electrochemical stability of $Li_2ZnCu_3$ alloy and to prevent interference of Cu substrate on XRD signals as shown in Supplementary Figs. 23, 24, the working electrode is set to be the Li/LiZn anode for simplification. As shown in Fig. 3j, upon the initial Li stripping (voltage <0.15 V), there are no apparent peak shifts or new peak generation to demonstrate the electrochemical stability of LiZn alloy, while the metallic Li is continuously dissolved into the electrolyte as shown by the gradual intensity decrease of Li (110) and (200) peaks. At the end of Li stripping, the XRD peak intensity of LiZn alloy fades and it decomposes partially into Zn and Li-deficient Li-Zn alloys such as $Li_2Zn_5$ component (black XRD pattern) at a voltage higher than 0.15 V after 4.5 h as shown in Fig. 3j. This implies that due to a higher oxidization potential of LiZn alloy with respect to the metallic Li, the LiZn alloy will maintain its composition and crystal structure before the complete stripping of the surrounding Li ions to act firstly as a framework as shown in Supplementary Fig. 11, and then as a Li source, but leaving abundant nucleation sites (such as Zn, LiZn, and Li-deficient Li-Zn alloys) for the subsequent Li non-directional deposition. As shown in Fig. 3k, such a hierarchical Li stripping process in Li/LiZn@Cu anode establishes the perfect Li stripping process, the necessary Li supplement as well as the upcoming Li plating process, similar to the retained alloy-type framework[35,51,52]. Then upon Li plating, the LiZn alloy is regenerated once the Li begins to deposit as shown in Fig. 3j. Of special importance is that the deposition surface of Li changes to the (110) plane rather than the starting (200) plane, which is conducive to the formation of planar Li morphology[53–55].

## Electrochemical performance of Li/LiZn@Cu anode

Figure 4 displays the electrochemical performance of thin Li/LiZn@Cu and Li@Cu anodes using symmetric cells. The cycled capacity is fixed at 1 mAh cm$^{-2}$, corresponding to a discharge of depth of 22.7%, which is greater than that of the reported works as listed in Supplementary Table 6. As shown in Fig. 4a, both of the two cells exhibit stable voltage profiles, while the visible difference is that the overpotential of Li/LiZn@Cu is lower than that of the Li@Cu anodes in the initial 90 cycles at 0.5 mA cm$^{-2}$/1 mAh cm$^{-2}$. This can be attributed to a higher Li nucleation/deposition barrier on the Li@Cu surface caused by a poor interface process as shown by the electrochemical impedance spectroscopy (EIS) spectra in Supplementary Figs. 25, 26 and Supplementary Table 7, where the dendrite-like and nonuniform-thickness Li@Cu anode has an initial unstable Li/electrolyte interface with the higher SEI ($R_{SEI}$) and charge-transfer ($R_{CT}$) resistances than that of the Li/LiZn@Cu anode. Moreover, the voltage polarization of Li@Cu anode increases at approx. 192 h and the failure of the symmetric cell happens after ~380 h as shown in Fig. 4a, b. In sharp contrast, the symmetric cell of Li/LiZn@Cu anode delivers a stable cycling performance over 1200 h in response to the aforementioned homogeneous Li plating/stripping process and robust Li/electrolyte interface endowed by the excellent lithiophilicity, low $Li^+$ ion diffusion energy barrier, and framework-like function of LiZn alloy. Even at a higher current density of 1 mA cm$^{-2}$, the Li/LiZn@Cu anode still delivers a similar electrochemical performance without voltage fluctuations over 690 h as shown in Fig. 4c, d. Note that such exceptional performance of the thin Li/LiZn@Cu anode (35 μm, ~4.4 mAh cm$^{-2}$) is even better than the commercial Li foil (450 μm, ~90 mAh cm$^{-2}$) as shown in Supplementary Figs. 27, 28.

The rate capability of the Li/LiZn@Cu and Li@Cu symmetric cells is then carried out under different current densities of 0.5, 1, 2, 3, and 5 mA cm$^{-2}$ with a stationary cycling capacity (1 mAh cm$^{-2}$). As shown in Fig. 4e, the Li/LiZn@Cu cell displays overwhelmingly more stable voltage profiles and lower overpotentials than those of the Li@Cu cell throughout the whole cycling process. After the rate test, the overpotential of the Li/LiZn@Cu symmetric cell can also correspondingly return to its initial state of ~9 mV, totally different from the Li@Cu

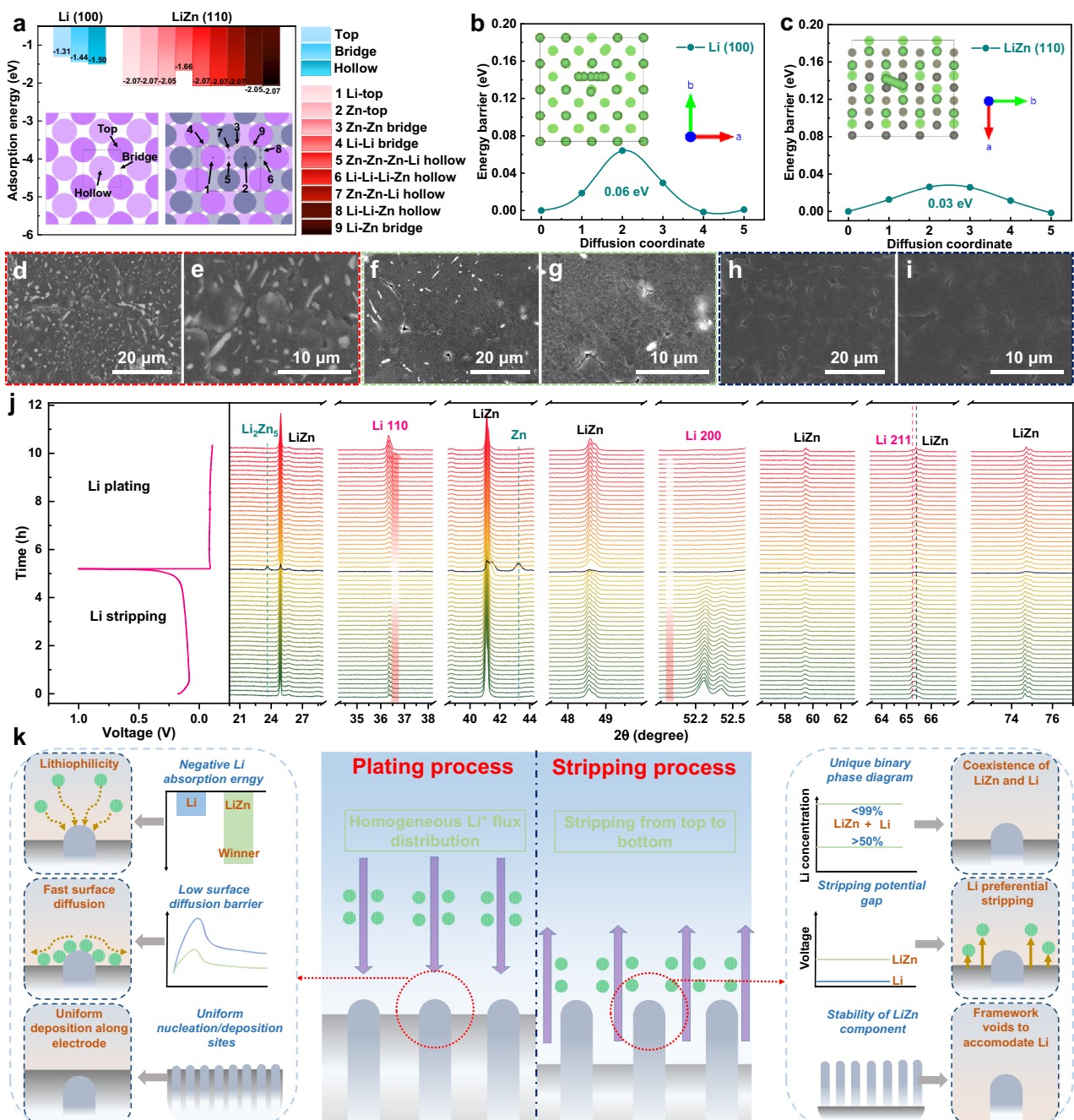

**Fig. 3 | Li plating/stripping behaviors on Li/LiZn@Cu anode. a** Li absorption energies on Li (100) and LiZn (110) surfaces. The diffusion barriers of Li along (**b**) Li (100) and (**c**) LiZn (110) surfaces. SEM images of Li/LiZn@Cu anode after Li deposition at (**d**, **e**) 0.5, (**f**, **g**) 1, and (**h**, **i**) 3 mAh cm⁻². **j** In-situ XRD analysis and (**k**) schematic illustration of the Li stripping and plating processes of the Li/LiZn@Cu anode.

symmetric cell at the same 0.5 mA cm⁻². Furthermore, Fig. 4f demonstrates the average Coulombic efficiency (ACE) tests to appraise the reversibility of repetitive Li plating and stripping behaviors in the symmetric cells. Compared with the low ACE of 96.8% for the Li@Cu cell, the Li/LiZn@Cu cell delivers a high ACE of 99.2% after 50 cycles at 1 mA cm⁻²/1 mAh cm⁻², indicating a high Li utilization with a low accumulation of "dead Li".

To understand the excellent Li deposition and stripping behaviors, the Ar⁺ ion etched in-depth X-ray photoelectron spectroscopy (XPS) analysis is carried out on the cycled Li/LiZn@Cu and Li@Cu electrodes. As shown in Fig. 5a, the pristine Li *1s* XPS spectrum of the cycled Li/LiZn@Cu can be deconvoluted into the LiF, Li₂CO₃, and Li₂O, which is similar to that of the cycled Li@Cu electrode as shown in

Fig. 5b. While, the content of LiF in Li/LiZn@Cu is higher than the Li@Cu, probably giving a profitable effect for uniform Li⁺ flux distribution and electron insulation[7,56], which is also verified in the spectra of F *1s* as shown in Fig. 5f. Of special interest is that at the etching time of 120 s, a new peak at 52.0 eV is assigned to the metallic Li in Li/LiZn@Cu, while it does not show up in Li@Cu, which plausibly indicates the thinner SEI growth of Li/LiZn@Cu with respect to that of Li@Cu electrodes. Alongside the etching, the metallic Li appears in both electrodes, but the amount of the parasitic Li₂CO₃ by side reactions is obviously higher in Li@Cu than that in Li/LiZn@Cu electrodes. Figure 5c–e demonstrates the characteristic C *1s* peaks at 284.8, 286.4, 288.6, 289.7, and 292.8 eV that are identified as the C-C, COR (R: radicals), HCO₂Li/COOR, Li₂CO₃, and C-F₃, respectively[57,58]. Especially,

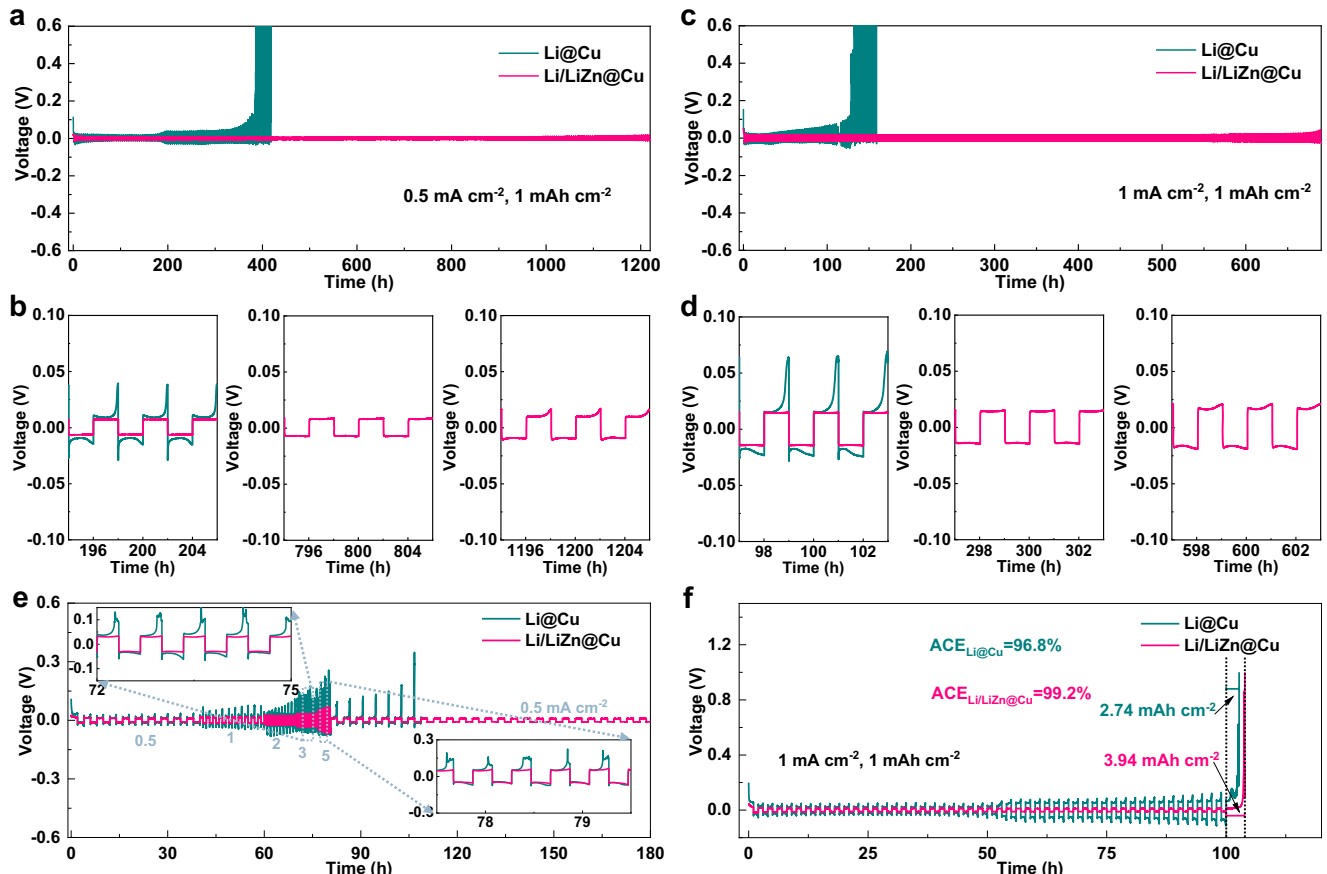

**Fig. 4 | Symmetric cell performance of thin Li/LiZn@Cu and Li@Cu anodes.** Galvanostatic cycling and detailed voltage profiles at (**a**, **b**) 0.5 mA cm⁻²/1 mAh cm⁻² and (**c**, **d**) 1 mA cm⁻²/1 mAh cm⁻². **e** Rate performances at the current densities from 0.5 to 5 mAh cm⁻² with 1 mAh cm⁻². Insets are the detailed voltage profiles at 3 and 5 mA cm⁻². **f** Average Coulombic efficiency (ACE) for 50 cycles of Li plating/stripping at 1 mA cm⁻²/1 mAh cm⁻².

the C $1s$ peaks higher than 286 eV almost disappear at 480 s in Li/LiZn@Cu, while visibly present in Li@Cu throughout the etching process, which corresponds to the carbonaceous species (COR, HCO$_2$Li/COOR, Li$_2$CO$_3$, C-F$_3$, etc.) mainly from the irreversible side reactions. This reveals that the Zn-based alloys influence the SEI formation as well as the building of Li/LiZn@Cu/electrolyte interface to effectively protect the Li against electrolyte corrosion and provide reversible Li plating/stripping behaviors for a prolonged cycling lifespan as demonstrated in Fig. 4. In addition, there are no XPS peaks in Zn $2p$ spectra of Li/LiZn@Cu even after 960 s etching, indicating the uniform Li deposition layer guided by the LiZn alloy substrate due to the excellent lithiophilicity as shown in Supplementary Fig. 29.

Figure 5g–j and Supplementary Fig. 30 show the surface morphology evolution of Li@Cu and Li/LiZn@Cu electrodes after repeated Li plating/stripping. As shown in Supplementary Fig. 30a, b, a thick SEI layer with huge cracks is observed on the Li@Cu surface after 50 cycles at 1 mA cm⁻² and 1 mAh cm⁻². This thick and exfoliated SEI is composed of the stacked "dead Li", arising probably from the undesired Li dendrite and the repeated rupture of SEI. Moreover, the cross-section SEM images of Li@Cu exhibit a completely deteriorated interface morphology, which would hamper the mass transfer process for the ion/electron transportations as shown in Supplementary Fig. 30c, d. In contrast, a homogeneous deposition layer (~34 μm) consisting of numerously compact Li grains is formed on the Li/LiZn@Cu surface as shown in Fig. 5g–j. This signifies that the multi-functionalized LiZn alloy can well regulate the homogeneous Li deposition and restrict the uncontrollable volume expansion to promote the electrochemical performance. It can be further seen from the EIS analysis that as shown in Supplementary Fig. 31, the semicircle at high frequency corresponds

to the Li-ion transport inside the SEI (R$_{SEI}$), which decreases and almost stabilizes in Li/LiZn@Cu electrode, while it gets smaller first and then bigger from the 1$^{st}$ cycle to the 50$^{th}$ cycle, reflecting the deterioration of interface dynamic process in Li@Cu.

Figure 6 displays the full cell performance with the thin Li/LiZn@Cu anode, where the cathode is LiFePO$_4$ (LFP) with an average active mass loading of 8.7 mg cm⁻² (corresponding to ~1.3 mAh cm⁻²) and the capacity ratio between the anode and cathode (N/P ratio) is set to be ~3.4, a low value than most of the reported ones as shown in Supplementary Table 6. As shown in Fig. 6a and Supplementary Fig. 32, the Li/LiZn@Cu||LFP cell delivers not only a stable Coulombic efficiency of ~100% and negligible capacity decay throughout the total cycle, but also a prolonged lifespan over 230 cycles with the improved capacity retention of 98%. In sharp contrast, the Li@Cu anode-based full cell can normally operate for only 70 cycles, then quickly fades to 113.1 mAh g⁻¹ after 80 cycles. Such an inferior cycling performance of Li@Cu||LFP cell is directly ascribed to the overgrown dendrite-induced interface failure, whereas the morphology of Li/LiZn@Cu is still smooth and undamaged during cycling as shown in Fig. 6b, c and Supplementary Fig. 33. Additionally, considering that the inevitable introduction of the rugged Li morphology and "dead Li" in Li@Cu anode during electrochemical preparation, which will exert innate disadvantages in full cell tests, the mechanical rolling-based thin Li@Cu (M-Li@Cu, ~24 μm-thick Li) anode is further set to be the control group as shown in Supplementary Fig. 34. When paired with the high-loading LFP (13.1 mg cm⁻², corresponding to ~2 mAh cm⁻², N/P = 2.2), the prominent performance of Li/LiZn@Cu||LFP full cell is also inherited, where it displays enhanced capacity retention of 90.1% and a prolonged lifespan with respect to those of the Li@Cu||LFP and M-

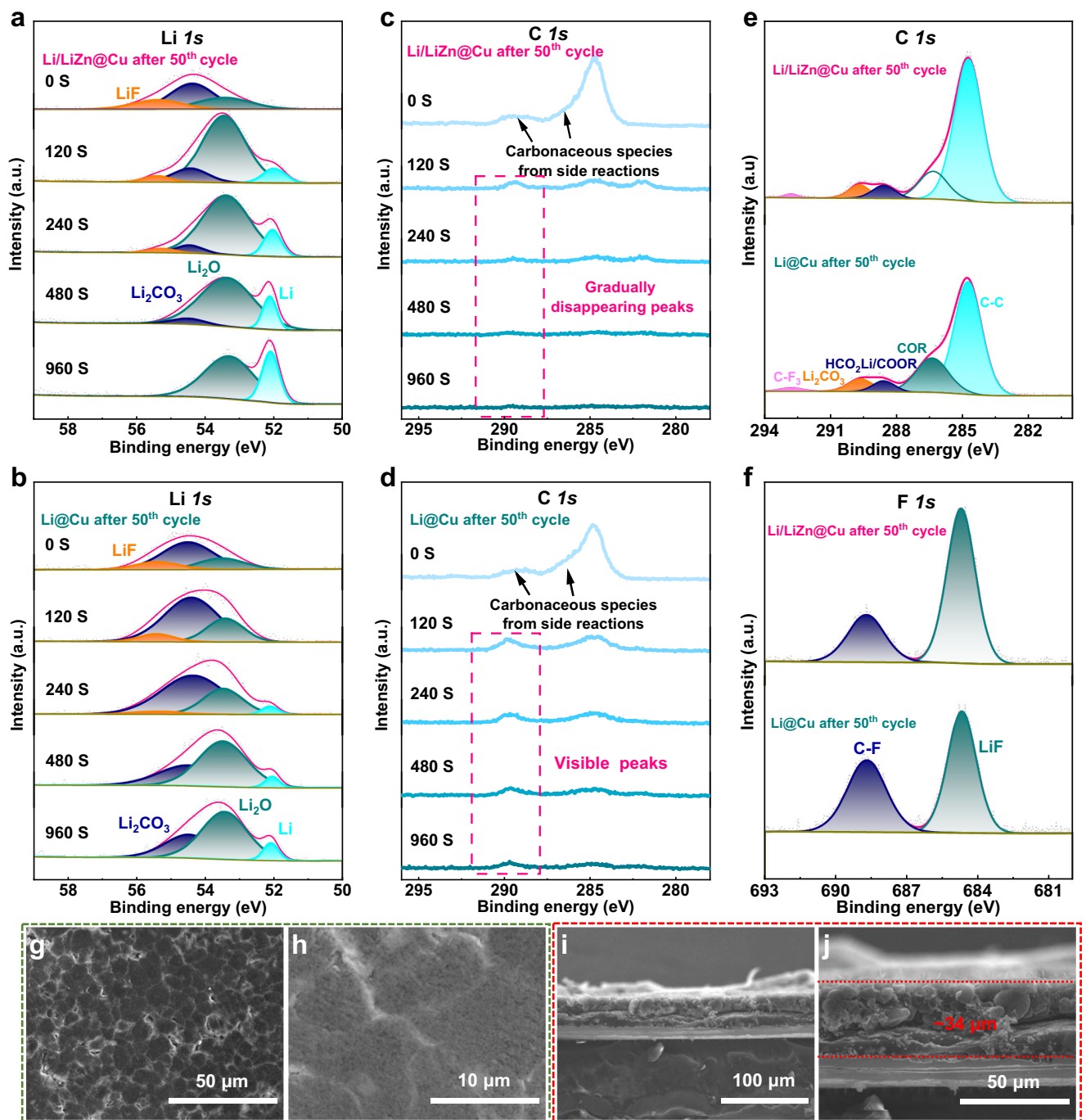

**Fig. 5 | Postmortem analysis of the symmetric cells after the 50th cycle. a** Li *1s*, **c** C *1s* XPS spectrum of Li/LiZn@Cu anode at different etching times. **b** Li *1s*, **d** C *1s* XPS spectrum of Li@Cu anode at different etching times. **e** C *1s*, **f** F *1s* XPS spectrum of Li/LiZn@Cu and Li@Cu anodes without etching. **g**, **h** Top-view and **i**, **j** cross-section SEM images of the cycled Li/LiZn@Cu anode.

Li@Cu‖LFP cells as shown in Supplementary Fig. 34o. Subsequently, the rate performance of Li/LiZn@Cu‖LFP is implemented as shown in Fig. 6d, where the Li/LiZn@Cu‖LFP delivers the high discharging capacities of ~148.7, 144.3, 137.2, 117.4, and 99.1 mAh g⁻¹ at 0.2, 0.5, 1, 3, and 5 C, respectively, higher than that of the Li@Cu‖LFP full cell as shown in Supplementary Fig. 35.

Moreover, highly demanding operation conditions are exerted on Li/LiZn@Cu anodes. By replacing the 25 μm-thick Li/LiZn layer with 10 or 5-μm-thick Li/LiZn layers, the N/P ratios of the as-assembled Li/LiZn@Cu‖LFP cells can be further reduced to ~1.4 or 0.7. Typically, the Li/LiZn@Cu‖LFP cells with lower N/P ratios can still maintain high average capacities of 144 and 145 mAh g⁻¹ over 80 and 60 cycles at

0.5 C as shown in Supplementary Figs. 36, 37. In the meantime, the full cell with LiCoO₂ cathode (LCO, 1.8 mAh cm⁻²) is cycled under a voltage window of 2.8–4.5 V (N/P = ~2.5), which also exhibits a high initial capacity of 1.79 mAh cm⁻² and good capacity retention as shown in Supplementary Fig. 38. As shown in Fig. 6e, the full cell with the Li/LiZn@Cu anode possesses enhanced cycling stability and capacity retention of ~74% after 125 cycles at 0.5 C (1 C = 274 mA g⁻¹). While with the Li@Cu anode, the Li@Cu‖LCO full cell shows a similar specific capacity in the first 50 cycles but accompanies a continuous capacity loss in the subsequent cycles. The degradation of Li@Cu‖LCO cell is mainly induced by the finite Li depletion and inactive SEI accumulation, which exerts a large barrier for Li⁺ transportation across the SEI

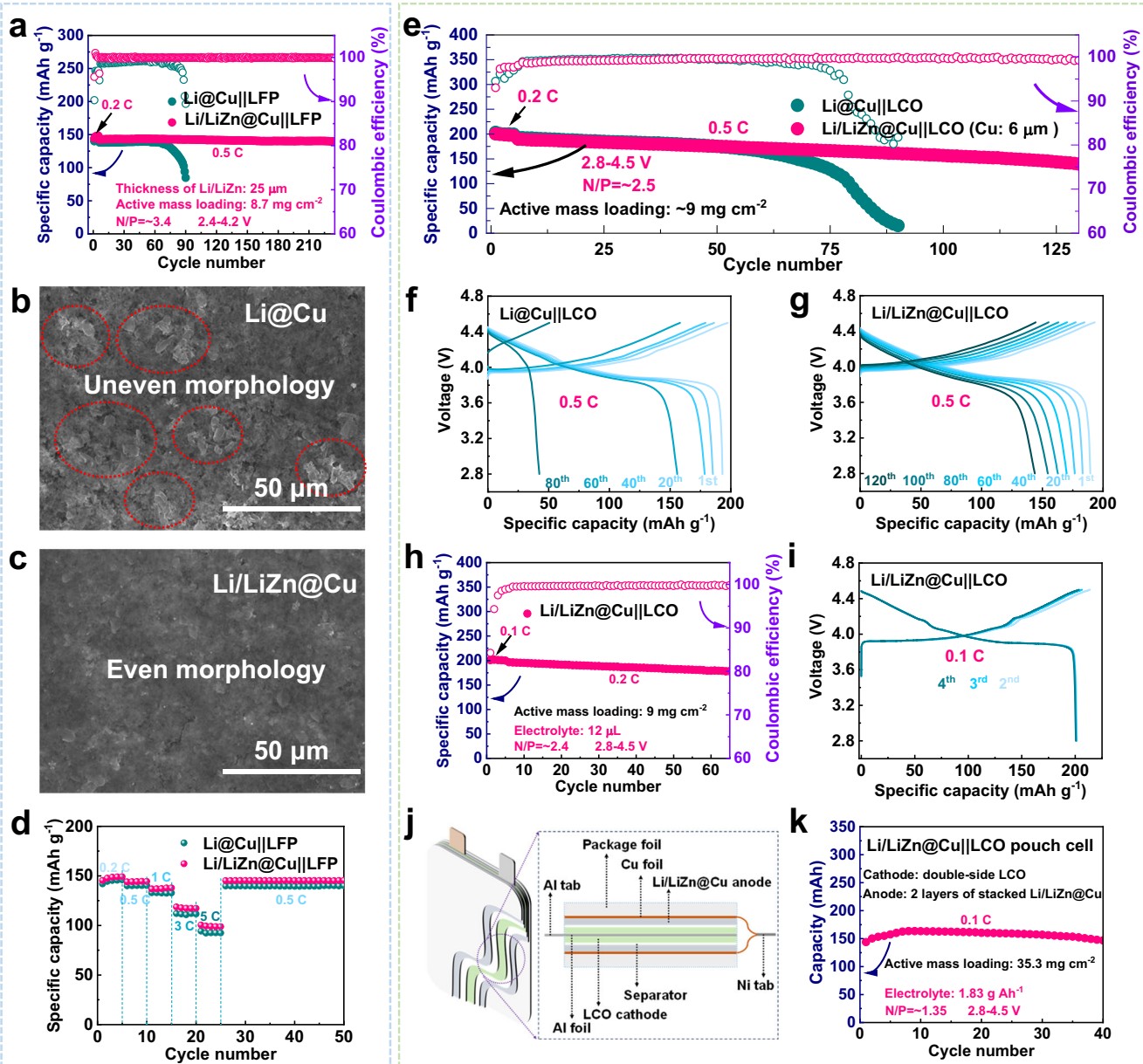

**Fig. 6 | Electrochemical performance of Li@Cu and Li/LiZn@Cu full cells.**
**a** Cycling performance of Li@Cu||LFP and Li/LiZn@Cu||LFP full cells with an N/P ratio of ~3.4 at 0.5 C between 2.8 and 4.2 V. Top-view SEM images of the (**b**) Li@Cu and (**c**) Li/LiZn@Cu anodes after the 30th cycle in the LFP-based full cells. **d** Rate performance of Li@Cu||LFP and Li/LiZn@Cu||LFP full cells. The thickness of Cu foil is ~10 μm. **e** Cycling performance and (**f**, **g**) corresponding voltage profiles of Li@Cu||LCO and Li/LiZn@Cu||LCO full cells with an N/P ratio of ~2.8 at 0.5 C between 2.8 and 4.5 V. **h** Cycling performance of Li/LiZn@Cu||LCO full cell with a lean electrolyte of 12 μL and **i** corresponding voltage profiles for selected cycles at 0.1 C. **j** Schematic illustration and **k** corresponding cycling performance of the Li/LiZn@Cu||LCO pouch cell. The thickness of Cu foil is ~6 μm.

and gradually increases the voltage hysteresis as shown in Fig. 6f, g. To further optimize the energy density, the electrolyte is decreased down to 12 μL per cell. As shown in Fig. 6h, i and Supplementary Table 8, the as-prepared Li/LiZn@Cu||LCO full cell delivers a high discharging capacity of 201.6 mAh g⁻¹ at 0.1 C, corresponding to a gravimetric energy of 283.7 Wh kg⁻¹ (~700 Wh kg⁻¹ if only considering the active mass of cathode and anode), which is much higher than the commercial LIBs.

Furthermore, a 0.16 Ah-level Li/LiZn@Cu||LCO pouch cell is packaged to validate the cycling superiority in real conditions as shown in Supplementary Fig. 39, Fig. 6j, k, and Supplementary Table 9. The pouch cell consists of two single-sided Li/LiZn@Cu (4.4 mAh cm⁻²) anodes and one double-sided LCO cathode (active mass loading: 35.3 mg cm⁻²), in which the N/P ratio is ~1.35 and the amount of

electrolyte is 1.83 g Ah⁻¹. Such a stacked cell with only two layers could achieve a high energy density of 284.0 Wh kg⁻¹, surpassing most of the available practical LIBs as indicated in Supplementary Table 9. As a matter of fact, taking the performance of this pouch cell as a reference, the energy density can be further increased to ~366.5 Wh kg⁻¹ at about 10 staked double-sided LiCoO₂ cathodes as shown in Supplementary Table 10.

The finally well-regulated Li/LiZn@Cu anode demonstrates the hierarchical Li electrochemistry to be well compatible with the high-performance LIBs. Upon charging, the Li/LiZn@Cu anode firstly acts as the Li source to establish the even Li stripping. Then, at the end of charging, the de-alloy of LiZn particle further supplements the Li extraction, actually playing the Li compensation role in battery cycling, which has a profound influence on the practical application of

available high energy density materials, such as the Si or $SiO_x$ based anodes and NCM-based high energy density layered cathodes. Furthermore, upon discharging, the LiZn alloy will form just at the beginning, and then the LiZn alloy-regulated Li homogeneous deposition follows along with the cycling. The reversibility of such an interesting process has been undoubtedly verified from the aforementioned electrochemistry and in-situ XRD characterization. These results provide a rewarding avenue toward the utilization of Li metal anode, the Li supplementary strategy, and the prevailing high-energy-density Li metal batteries

## Discussion

In conclusion, the thin Li/LiZn@Cu anodes with large sizes (>100 cm$^{-2}$) and adjustable thicknesses/capacities (5–48 μm of Li/LiZn, 0.89–8.7 mAh cm$^{-2}$) are successfully fabricated by a facile and low-cost approach for high-energy-density LIBs. Benefiting from the LiZn alloy, the thin Li/LiZn@Cu anode demonstrates the homogeneous and abundant surface functional sites to intrigue the Li stripping/plating and delivers the hierarchical Li electrochemistry to regulate the dendrite-free morphology and robust Li/electrolyte interface. As expected, the Li/LiZn@Cu anode not only exhibits excellent cycling stability over 1200 h at 0.5 mA cm$^{-2}$/1 mAh cm$^{-2}$ in the symmetric cell, but also realizes a prolonged lifespan of 230 cycles under a suitable N/P ratio when paired with the LFP cathode. Moreover, a high energy density battery of 284.0 Wh kg$^{-1}$ (366.5 Wh kg$^{-1}$ when containing 10 stacked layers) is obtained in LCO-based pouch cell, consisting of the lean electrolyte (1.83 g Ah$^{-1}$), low N/P ratio (1.35), thinner Cu foil-based Li/LiZn@Cu anode, and double-sided LCO cathode (36.6 mg cm$^{-2}$). Such a promising alloy-type technology for thin Li metal anodes sheds a practicable and up-and-coming routine for future high-performance and high-energy-density Li metal batteries.

## Methods
### Preparation of the thin Li anode

Thin Li/LiZn@Cu anode was fabricated via a doctor-blade casting process and all steps were performed in the glovebox (MBRAUN, contents of $H_2O$ and $O_2$ were less than 0.1 ppm). First, the polished Li and Zn foil were mixed together (mass ratio: 4:1) and heated to 250 °C in a stainless-steel container. Then, the molten Li would alloy with the Zn foil to form the Li-Zn mixture (Li+LiZn alloy). After full stirring, the molten mixture was transferred to the Cu foil surface (temperature: 200 °C). Due to the further alloying reaction (Li, LiZn, and Cu) and reduced surface tension of formed LiZn, the quantitatively thin Li/LiZn@Cu anode could be readily fabricated with the assistance of the doctor-blade method on Cu foil. In addition, the e-Li/LiZn@Cu electrode can be obtained after etching off the Li/LiZn layer with alcohol. To acquire the thin Li@Cu anode, a half cell using Cu foil as the working electrode and Li foil as the counter electrode was assembled, and then 4.5 mAh cm$^{-2}$ Li was electroplated on the Cu surface at 0.5 mA cm$^{-2}$. The electrolyte was 1 M lithium bis(trifluoromethanesulphonyl) imide (LiTFSI) in 1,3-dioxolane (DOL)/1,2-dimethoxyethane (DME) (v/v = 1:1) with 1 wt% LiNO$_3$. After disassembling the half cell and rinsing the Li-plated Cu foil with DOL, the preparation of the Li@Cu anode was completed. The thin M-Li@Cu anode (thickness: ~24μm-thick Li+10 μm-thick Cu foil) was purchased from China Energy Lithium Co., Ltd.

### Materials characterization

SEM images were obtained using HITACHI SU5000 field emission scanning electron microscopes. To observe the top-view or cross-section morphologies of the cycled electrodes, they were rinsed with DOL to remove the residual electrolyte in advance and dried in the glovebox. XRD measurements were conducted via Rigaku SmartLab and Bruker D8 Advance X-ray diffractometers with Cu Kα radiation. To reduce the air influence on highly reactive Li, the surface of Li/LiZn@Cu samples was covered with Kapton tape before XRD measurements. In-depth XPS spectra were explored through Thermofisher Scientific ESCALAB Xi+ with monochromatic Al Kα radiation. All samples were processed in glove box and transferred to the XPS chamber using a transfer box.

### In-situ XRD measurement

The in-situ XRD diffraction was carried out on a Bruker D8 Advance X-ray diffractometer (Cu Kα radiation) with the successive scanning mode (2θ range: 20° to 80°). The cathode and anode were the same and prepared by cooling down the molten Li-Zn mixture and then pressing it into thin foil (Li/LiZn). Then the shaped Li/LiZn foils were assembled into a symmetric cell in a homemade in-situ cell with a beryllium window for further XRD characterization. In the in-situ Li||e-Li/LiZn@Cu cell, the cathode was e-Li/LiZn@Cu (Cu thickness: 6 μm) and the anode was Li foil. The electrolyte was 1M LiTFSI in DOL/DME (v/v = 1:1) with 1 wt% LiNO$_3$.

### Electrochemical measurements

All the electrochemical measurements were investigated using CR2025-type coin cells and conducted on the LANHE battery testing system (CT 3002A). In symmetric cell tests, two identical electrodes (Li@Cu or Li/LiZn@Cu) were used as the counter and working electrode. In average Coulombic efficiency (ACE) tests, the symmetric cells with different electrodes were firstly cycled 50 cycles under 1 mA cm$^{-2}$/1 mAh cm$^{-2}$, and then stripping all residual Li at 1 mA cm$^{-2}$ until the voltage reached 1 V. The value of ACE can be calculated as the following formula[59]:

$$ACE = (nC_F + C_S)/(nC_F + C_P) \qquad (4)$$

where $n$ is the cycle number, $C_P$ is the pristine areal capacity of the tested electrode, $C_F$ is the fixed areal capacity used in cycles, and $C_S$ is the final stripping areal capacity of the tested electrode. EIS tests of symmetric cells were conducted on an electrochemical workstation (Princeton Applied Research, PARSTAT MC) with a frequency range from 200 kHz to 0.1 Hz. The involved electrolyte in the above tests was 1M LiTFSI in DOL/DME (v/v = 1:1) with 1 wt% LiNO$_3$. In full cell tests, all cathodes were prepared by the blade-casting approach. For the LFP cathode with ~10.9 mg cm$^{-2}$, the mass ratio of LFP, carbon black, and polyvinylidene fluoride (PVDF) was 8:1:1. The LFP cathode with a higher mass loading of ~14 mg cm$^{-2}$ was provided by Kejing (mass ratio of LFP: 93.4%). For the LCO cathode, the mass loading was ~10 mg cm$^{-2}$ (mass ratio of LCO: 90%). All full cells were first cycled 5 cycles at 0.2 or 0.1 C for activation before further tests. The electrolyte for full cells was 1 M lithium hexafluorophosphate (LiPF$_6$) in ethylene carbonate (EC)/dimethyl carbonate (DMC)/ethyl methyl carbonate (EMC) (v/v/v = 1:1:1) with 5% fluoroethylene carbonate (FEC). Unless otherwise specified, the amount of electrolyte for coin-type cells was 35 μL. In the pouch cell test, the used double-sided LCO cathode possessed a higher mass loading of ~38.8 mg cm$^{-2}$ (mass ratio of LCO: 91%), and the applied size was 5 × 5 cm$^{-2}$. The operating voltage window and rate were set as 2.8–4.5 V and 0.1 C. Notably, the charging operation contained two processes: the first step was a galvanostatic charging process at 0.1 C until the voltage reached 4.5 V, while the next step was a constant-voltage charging process until the charging rate reduced to 0.03 C. The detailed information is displayed in Supplementary Tables 9, 10. The energy densities (Wh kg$^{-1}$) of the coin-type cell and pouch cell were calculated by the following formula:

$$E_g = \frac{E_D}{m_T} \qquad (5)$$

where $E_g$ is the specific energy of the cell (Wh kg$^{-1}$), $E_D$ is the discharging energy per unit area given by the data processing software of the LANHE battery testing system, and the $m_T$ is the total weight per unit area based on the sum of pouch cell bag, separator, electrolyte, current collector, cathode, and anode. All the electrochemical measurements were conducted in the incubator at 27 °C.

## Theoretical calculations

The Vienna Ab Initio Simulation Package[60,61] (vasp.5.4.4) based on the projected augmented wave[62] (PAW) potentials was utilized for first-principles calculations. The generalized gradient approximation[63] (GGA) in the parameterization of Perdew, Burke, and Ernzerhof (PBE) functional was adapted to describe the electronic exchange-correlation interaction. The cut-off energy of 450 eV and Γ-centered $4 \times 4 \times 1$ k-point mesh was applied, respectively. Geometry optimizations were performed until the convergence criteria of energy and force reached $1 \times 10^{-6}$ eV and 0.01 eV/Å, respectively. The vacuum thickness of 15 Å for all slab structures was implemented to prevent the interaction between the periodic images. The Li-ion migration barriers were calculated by climbing image nudged elastic band (CINEB)[64,65] method with 4 images as the intermediate states, where the CINEB simulation is considered complete when the magnitude of force per atom is smaller than 0.03 eV/Å. The optimized 8-layer slab was utilized for the calculation of the Li-ion migration barriers, where the four bottom layers were fixed at bulk positions while all other layers were allowed to relax. The crystal structures were built using VESTA software[66]. The compositional phase diagrams were built using the *phase_diagram module* of Pymatgen[67]. The Gibbs free energy G(T) of the formation of solids at high temperatures (200 or 250 °C) can be estimated using the machine learning model SISSO (sure independence screening and sparsifying operator) approach[39]. And the Gibbs free energy per atom (ΔG) between reactants and products can be calculated as follows:

$$\Delta G = (G_{products} - G_{reactants}) \qquad (6)$$

The Li adsorption energy $E_{adsorb}$ is described as follows:

$$E_{adsorb} = E_{slab+Li} - (E_{slab} + E_{Li}) \qquad (7)$$

Where the $E_{slab+Li}$ is the total energy of the slab structure with one Li adsorbed on the surface, $E_{slab}$ is the total energy of the surface structure, and $E_{Li}$ is the total energy of one Li.

The interface formation energies of Cu(001)|Li(001) and Cu(001)|LiZn(110) interfaces systems can be calculated as follows:

$$E_f = (E_{ab} - E_a - E_b)/S \qquad (8)$$

Where the $E_{ab}$ is the complete system containing the interface, $E_a$ and $E_b$ denote the separated slabs (Cu(001), Li(001), and LiZn(110) slab), and $S$ refers to the interfacial area.

The surface energy σ of Li$_2$ZnCu$_3$ can be obtained as follows:

$$\sigma = \frac{1}{2A}[E_{slab} - nE_{bulk}] \qquad (9)$$

where the $E_{slab}$ is the energy of relaxed slab models with no restriction of the atomic position, the $E_{bulk}$ is the total energy of the bulk structure, and the A is the area surface.

## Data availability

The authors declare that all data supporting the finding of this study are available within the paper and its supplementary information files. All raw data generated during the current study are available from the corresponding authors upon request. Source data are provided with this paper.

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

## Acknowledgements

This work was supported by National Key Research and Development Project (2019YFA0705702), National Natural Science Foundation of China (22075328, 22379168), Guangdong Basic and Applied Basic

Research Foundation (2021B1515120002), and the Natural Science Foundation of Guangdong Province (2022A1515010405). Computational resources were provided by the Tianhe–2 National Supercomputer Center in Guangzhou.

## Author contributions

J.Q.C. and X.L. conceived the project. J.Q.C. performed the materials synthesis and electrochemical testing. Y.S.S. performed the computational calculations. A.S.G collected the XPS spectra. M.D. and Y.F.Z performed the in-situ XRD tests. J.Q.C., Y.S.S., G.Y.D., M.H.C., G.Y.Q., X.Y.L., F.Y.X., and Y.S. performed the characterizations. J.Q.C. and X.L. analyzed the data and proposed the mechanisms. J.Q.C. wrote the draft that was revised and finalized by X.L. All authors discussed the results and commented on the manuscript.

## Competing interests

The authors declare no competing interests.
