## [Peer Review File · Nature Communications]

REVIEWER COMMENTS

Reviewer #1 (Remarks to the Author):

The manuscript reports that adding Zn in molten Li improves its wettability on Cu foil, thereby facilitating the preparation of Li metal anodes, particularly at very low thicknesses. Besides, Li metal anodes containing Zn exhibit superior electrochemical performance when compared to pure Li anodes. This work contains several interesting phenomena with application potential in Li metal anode batteries. However, the experiments and the explanation on the phenomena contain significant flaws. Prior to considering the publication, the following concerns must be adequately addressed.

1. The authors described the LiZn alloy as a pillar-like structure in the Li/Li-Zn@Cu anode (Figure 2k). However, it appears that no experimental results, such as the SEM images, match this description. It should be clarified whether the LiZn particles are separately dispersed among Li particles or steadily planted on the Cu foil. Besides, the effects of LiZn and Li₂ZnCu₃ phases must be clearly distinguished and summarized based on the experimental results.
2. The authors investigate the wettability of molten Li on Cu in Figures 1 and 6. The contents appear to be highly interrelated and should be combined for a coherent narrative.
3. Could the authors experimentally measure the contact angles of molten Li and Li/LiZn on Cu or provide a supplementary video on the wetting process? The photographs in Figure 1b and 1c cannot be regarded as credible evidence.
4. In Li/LiZn@Cu, the EDS mapping (Figure 1j) indicates that Zn are contained in the bcc Li phase. Would Zn atoms dissolved in the Li phase have an effect on the reaction behavior?
5. The authors claimed that the Zn-related phases contribute to a flat morphology when plating Li on Li/Li-Zn@Cu (Figure 2d-h). There should be control experiments of plating Li on Li@Cu to understand the differences.
6. The authors omitted the XRD peaks of Li₂ZnCu₃ from the in-situ XRD results (Figure 2j). Would the Li₂ZnCu₃ phase changes during redox reactions?
7. In the electrochemical experiments (Figure 3-5), how did the authors prepare the ultrathin Li@Cu anode? Would the different preparation process affect the material properties (e.g., grain size, dislocation, voids) and electrode performance?
8. How did the authors determine the dendrites in the electrodeposits in Figure 5b-c? The highlighted spots in the SEM image appear to be flakes rather than conventional Li dendrites.
9. In Figure 6a, why does the waste rate of Li resource become 0 when $N/P \leq 3$? Since the cathode (NMC or LCO) contains sufficient Li ions at the beginning, any Li on the anode ($N/P > 0$) would be a theoretical excess Li resource.

Reviewer #2 (Remarks to the Author):

In the present manuscript, the authors have delineated the fabrication of a composite anode utilizing a Li/LiZn@Cu alloy configuration. The manuscript posits that the strategic inclusion of other metallic constituents, notably Zn in this case, within the molten Li medium engenders heightened wettability towards the underlying Cu matrix. A commendable feature of this work is the authors' adept elucidation of the rationale behind the selection of Zn, an aspect thoughtfully expounded upon in the introductory segment. Furthermore, the manuscript combines both static and dynamic calculations, providing a comprehensive elucidation of the superior lithium plating characteristics inherent to their devised matrix. Also, an impressive enhancement in electrochemical performance is demonstrated. I appreciate that the authors did a lot work and provided some insights. However, several significant issues give rise to my reservations regarding its suitability for publication within the esteemed confines of a high-impact journal such as Nature Communications.

I. Fundamental errors

1 Kindly ensure consistent usage of the abbreviation for "Figure" throughout the entire manuscript. In certain instances, the abbreviation "Fig." has been utilized, while in others, "Figs." has been employed. Maintaining uniformity in this aspect contributes to the overall clarity and professionalism of the manuscript.

2 An observation has been made regarding the absence of proper journal-title abbreviations for journals of a physical nature, such as "Physical Review B" (which should be abbreviated as "Phys. Rev. B").

3 The Gibbs reaction free energy values presented in this manuscript are derived from meticulous DFT calculations using VASP. It is important to recognize that these calculations are conducted under the assumption of a temperature of 0 K. Therefore, I draw attention to the text fragment at line 341, which references "The alloying reaction between Li and Zn at 250 °C," followed by the accompanying calculated reaction energy. It is pertinent to acknowledge that the disparities between reaction scenarios at significantly elevated and lowered temperatures can be substantial. I propose that the alloy reaction can be spontaneous based on the phase diagram, but urge a thoughtful consideration of this misleading expression.

4 To calculate the interface energy of two interfaces, the authors using the equation " $E_f = (E_{ab} - E_a - E_b)/S$ ", where the E_a and E_b represent the slab energy. However, it should be noted that in the Prof. Qi's work (J. Electrochem. Soc., 163 (3) A592-A598 (2016)), the interface energy is defined as the "The difference between the energy of the fully relaxed interface structure and the energy sum of the two stress free pure phases with the same atomic unit numbers". Therefore, the E_a and E_b ought to be the energy per unit of the stress-free pure A and B bulk structure, respectively.

II. Contradictions

1 DFT calculation results

The authors have employed the VASP package for their Density Functional Theory (DFT) calculations. However, notable disparities between their obtained results and the reported values have been observed. I highlight two distinct aspects that warrant elucidation:

a) Reaction Energy:

The manuscript details the computation of Gibbs reaction free energies for two reactions: Reaction 1 ($\text{Li}+\text{Zn}=\text{LiZn}$) and Reaction 2 ($\text{Li}+\text{LiZn}+3\text{Cu}=\text{Li}_2\text{ZnCu}_3$), yielding calculated values of -0.12 eV and -0.03 eV, respectively. In contrast, calculations performed using the Materials Project module's Reaction Calculator yield values of -0.44 eV and -0.321 eV for the respective reactions. While both sets of results exhibit negative values, indicative of spontaneous reactions, the substantial deviation casts doubts upon the reliability of the DFT calculations in this manuscript. A thorough explanation of the causes underlying these significant discrepancies is imperative.

b) Diffusion Barrier:

It is worth noting that literature encompasses several reports on calculations of Li^+ diffusion barriers on both pure Li and Li-alloy matrices, achieved through the CI-NEB method. Notably, Prof. Liang's work (Energy Storage Mater. 32 (2020) 178–184) has meticulously presented migration barrier values for Li^+ on Li (100) and LiZn (110). Prof. Liang's research demonstrates that the Li^+ migration barrier on Li (100) approximates ~ 0.14 eV, while on LiZn(110), the value stands at ~ 0.023 eV. However, intriguingly, in the present manuscript, the corresponding calculated values are 0.06 eV for Li (100) and 0.03 eV for LiZn (110). This substantial disparity prompts the need for an in-depth clarification concerning the deviation from established values in the literature. A comprehensive elucidation of the methodology, potential sources of error, and other contributing factors is indispensable in order to address this substantial deviation and enhance the credibility of the study's findings.

2 Research subject

This constitutes a central discrepancy within the manuscript's framework.

It appears that the authors advance the proposition that the LiZn alloy is pivotal in governing Li^+ plating behaviors, substantiated by their in-situ characterizations and theoretical derivations. Additionally, the function of Li_2ZnCu_3 is ascribed to aiding the diffusion of Li to the Cu foil. However, scrutiny of Fig. 1k reveals an intriguing pattern wherein the signal intensities of Li_2ZnCu_3 and Cu are notably more pronounced than those of Li and LiZn. This suggests a continuous interaction between LiZn, Li, and Cu, culminating in the formation of Li_2ZnCu_3 upon doctor-blading the mixtures at a temperature of 200 °C. Consequently, it becomes apparent that Li_2ZnCu_3 takes precedence as the principal constituent within the so-called Li-Zn/Cu composite anode.

A noteworthy aspect emerges during subsequent in-situ XRD characterizations, wherein the absence of a Li_2ZnCu_3 peak is perplexing. Closer examination reveals that the electrodes employed for these characterizations are formed by pressing mixtures onto the foil, rather than blade mixing onto the Cu foil. This shift in methodology implies that the predominant subject within the composite anode becomes Li-Zn. However, it is essential to note that the composite electrodes employed for later electrochemical assessments are prepared using the doctor-blading approach. Consequently, it logically follows that Li_2ZnCu_3 , not LiZn, should predominantly influence electrochemical performance in these composite electrodes.

At this juncture, a critical departure from rigor is identified. All calculations presented in the manuscript pivot on LiZn, neglecting the dominant role that Li_2ZnCu_3 assumes in the subsequent electrochemical

evaluations. A reevaluation of characterization results and a meticulous reconstruction of the manuscript's workflow are thus imperative.

III. Other concerns

1 The choice of control group

Upon scrutinizing the methodology section, it becomes evident that the designated control group, denoted as Li@Cu, is fashioned by electrochemically plating a specific quantity of lithium onto the copper foil. While this serves as a comparative baseline for the experimental results, a substantial performance variance between the experimental and control groups is apparent. However, the rationale underpinning this selection of the control group merits reevaluation.

The electrochemical plating process employed to prepare Li@Cu engenders morphological and physicochemical disparities that set it apart from the mechanically-rolled Li-Cu foil. Indeed, this preparation methodology aligns with what can be termed an "anode-free" scenario. In the context of prior investigations involving anode-free cells, it is well-established that electrochemically deposited lithium assumes a rugged morphology. Moreover, the generation of dead lithium within this context is not uncommon, a factor contributing to the degradation of cell performance.

As a constructive suggestion, it would be judicious to prepare or procure the Li-Zn@Cu composite anode and Li@Cu samples devoid of exposure to the electrochemical plating process. Additionally, considering the controllable nature of Li-Zn@Cu's areal capacity, a prudent approach could involve preparing a composite anode with a relatively substantial areal capacity. This strategic maneuver would facilitate seamless comparisons with commercial Li-Cu foil, which boasts a thickness of 50 μm .

2 The performance evaluation by comparing with other reports

This manuscript predominantly focuses on juxtaposing the optimal performance outcomes of the composite anodes against both LIBs and their designated control group. However, in light of the myriad reports concerning scalable composite anodes featuring noteworthy practical areal capacity and elevated capacities, it is essential for the performance metrics outlined in this study to be critically contextualized within the broader landscape of relevant literature. The scope of this comparative analysis should ideally extend beyond the LIBs and control group in order to provide a comprehensive assessment against existing research endeavors. For instance,

- a. "Roll-To-Roll Fabrication of Zero-Volume-Expansion Lithium-Composite Anodes to Realize High-Energy-Density Flexible and Stable Lithium-Metal Batteries" (*Adv.Mater.*2022, 34, 2205677)
- b. "A 10- μm Ultrathin Lithium Metal Composite Anodes with Superior Electrochemical Kinetics and Cycling Stability" (*Energy Environ. Mater.*2023,0, e12598)
- c. "Free-standing ultrathin lithium metal-graphene oxide host foils with controllable thickness for lithium batteries" (*Nat. Energy* 6, 790–798 (2021))

4 The electrolytes employed for in-situ XRD characterizations differ from those utilized for electrochemical performance evaluations.

5 The LiCoO₂ full cell is charged to 4.5 V. Has the surface of LiCoO₂ been modified?

6 In order to enhance the overall quality of this article, a more comprehensive and detailed analysis and characterization are warranted. Given that the strategy of introducing heteroatoms to enhance wettability is extensively documented in existing literature, a deeper exploration is imperative.

Reply to the comments

Reviewer #1 (Remarks to the Author):

The manuscript reports that adding Zn in molten Li improves its wettability on Cu foil, thereby facilitating the preparation of Li metal anodes, particularly at very low thicknesses. Besides, Li metal anodes containing Zn exhibit superior electrochemical performance when compared to pure Li anodes. This work contains several interesting phenomena with application potential in Li metal anode batteries. However, the experiments and the explanation on the phenomena contain significant flaws. Prior to considering the publication, the following concerns must be adequately addressed.

1. The authors described the LiZn alloy as a pillar-like structure in the Li/Li-Zn@Cu anode (Figure 2k). However, it appears that no experimental results, such as the SEM images, match this description. It should be clarified whether the LiZn particles are separately dispersed among Li particles or steadily planted on the Cu foil. Besides, the effects of LiZn and Li_2ZnCu_3 phases must be clearly distinguished and summarized based on the experimental results.

Reply:

Thanks for your comments.

We have supplemented more SEM images of the Li/Li-Zn@Cu anode. As shown in Fig. R1a-b, excepting for the nanodot-like structures, there are obvious nanorod-like (i.e. pillar-like) structures present on the Li/Li-Zn@Cu surface. It is worth noting that these nanodots correspond to the nanorods buried in the Li layer, resulting in the exposure of the top part only. After partial Li stripping, more previously buried alloy-based nanorods can be observed (Fig. R1g). Due to a higher oxidization potential of LiZn alloy with respect to the metallic Li, the LiZn alloy will maintain its composition and crystal structure before the complete stripping of the surrounding Li ions to act as a framework. Such buried nanorods can also be observed in the cross-section SEM images of the Li/LiZn@Cu anode (Fig. R1k-l). Thus, the 3D alloy skeletons and metallic Li together form the Li/LiZn layer on the upper part of the anode. Moreover, as shown in the EDS results, both Zn and Cu elements are distributed on the nanodot-like and nanorod-like structures, indicating that these nanostructures consist of LiZn and Li_2ZnCu_3 alloys (Fig. R1c-e and h-j). According to Fig. R1f and the chemical formula of Li_2ZnCu_3 , the

content of Li_2ZnCu_3 alloy in these nanostructures is only approx. 4.1%, implying that the LiZn alloy is the main component in the Li/LiZn layer. The Li_2ZnCu_3 alloy probably presents at the interface between the Li/LiZn layer and Cu substrate, namely, the surface of the Cu substrate (Fig. R11-n).

Fig. R1. (a-c) SEM top-view images, (d-e) the corresponding EDS mappings, and (f) compositional results of the Li/LiZn@Cu . (g-h) SEM top-view images and (i-j) the corresponding EDS mappings of the Li/LiZn@Cu after partial Li stripping. (k) SEM cross-section image and (l-n) the corresponding EDS mappings of the interface between the Li/LiZn layer and Cu substrate.

Besides, according to the phase diagram of the Li-Zn system, the Li/LiZn can be regarded as a mixture of the intermetallic compound LiZn and metallic Li at a concentration range of Li atoms from 50% to 99%. The atom concentration of Li in our designed Li/LiZn layer is $\sim 97.3\%$. After a cooling process, the Li-Zn mixture will occur a phase-segregation where the LiZn alloy can subsequently form a 3D framework and the metallic Li fills the surrounding area of the framework, similar to the many reported

Li-alloy-based anode (Adv. Mater. 2019, 31, 1804815, Adv. Mater. 2020, 32, 2000952, and J. Energy Chem. 2020, 51, 285-292).

Based on the above analysis, it can be concluded that the LiZn alloy is not steadily planted on the Cu foil but separately dispersed among Li particles. Consequently, the whole Li/LiZn@Cu anode can be divided into three parts from top to bottom, that is, (1) the top Li/LiZn layer, which consists of 3D Li alloy-based (LiZn and a small amount of Li₂ZnCu₃) nanoskeletons and metallic Li, (2) the intermediate layer of Li₂ZnCu₃ alloy, which is located at the interface between Li/LiZn layer and Cu substrate and (3) the bottom Cu substrate.

Fig. R2 (Supplementary Fig. 17d-l in SI). (a) SEM top-view image, (b-c) the corresponding EDS mappings, and (d) compositional results of e-Li/LiZn@Cu. (e-f) SEM cross-section images and (g-h) the corresponding EDS mappings of e-Li/LiZn@Cu. (i) XRD pattern of the e-Li/LiZn@Cu.

To better characterize the Li₂ZnCu₃ intermediate layer, through alcohol etching, the Li/LiZn active

layer can be removed and the Li_2ZnCu_3 layer is maintained (named as $e\text{-Li/LiZn@Cu}$). As shown in Fig. R2a, the surface of the $e\text{-Li/LiZn@Cu}$ presents coarse morphology with numerous pyramid-shaped structures, which can effectively improve the molten Li wettability. The uniform distribution of Zn and Cu elements with a close atomic ratio of 1:3 between Zn and Cu evidences the surface composition of $e\text{-Li/LiZn@Cu}$ is Li_2ZnCu_3 alloy (Fig. R2b-d). The SEM cross-section images and corresponding EDS mappings indicate that the thickness of Li_2ZnCu_3 is $\sim 2 \mu\text{m}$ (Fig. R2e-h). The strong Li_2ZnCu_3 peaks of the XRD pattern in Fig. R2i further verify the preservation of the Li_2ZnCu_3 layer. Thus, since Li_2ZnCu_3 mainly exists on the Cu surface rather than in the LiZn layer, its influence on electrochemical reactions is minimal. Besides, the excellent electrochemical stability of Li_2ZnCu_3 makes it not phase change during cycles (this part is placed in **Question 6**).

However, given that there is indeed a small amount of Li_2ZnCu_3 alloy present in the Li/LiZn layer, we still used theoretical calculation to discuss Li_2ZnCu_3 's role in the Li deposition behaviors. Due to the limited literature on Li_2ZnCu_3 alloy, the surface energies of different crystal planes of Li_2ZnCu_3 were studied to construct the most stable surface. As shown in Table R1, the Li_2ZnCu_3 (001) possesses the lowest surface energy so it is selected for Li adsorption and diffusion calculations.

Table R1 (Supplementary Table 5 in SI). Calculated surface energies of Li_2ZnCu_3 .

Composition	Surface miller indices	Surface energy (J m^{-2})
Li_2ZnCu_3	(001)	0.694
	(100)	0.852
	(110)	0.924
	(111)	0.952

As shown in Fig. R3a-d, the Li absorption energy of Li_2ZnCu_3 (001) surface is -2.78 eV with two possible adsorption configurations towards Li, which is even lower than the LiZn (110) (-2.07 eV), let alone being compared to the Li (100). It suggests that the Li_2ZnCu_3 alloy possesses a stronger Li^+ ion affinity, which can effectively promote Li nucleation and subsequently suppress the uneven Li deposition. However, as shown in Fig. R3e, the Li_2ZnCu_3 alloy does not have significant advantages in promoting Li surface diffusion compared to LiZn alloy (0.03 eV) and bare Li (0.06 eV), due to the

same value (0.06 eV) of surface diffusion barrier along the Li_2ZnCu_3 (001) and Li (100) surfaces. Consequently, compared with the bare Li, the Li_2ZnCu_3 alloy can improve the uniform Li deposition to some extent due to its superior lithiophilicity.

Fig. R3 (Supplementary Fig. 21 in SI). (a) Li adsorption energies on the Li_2ZnCu_3 (001) surface and (b) corresponding adsorption sites. (c) Another configuration of Li adsorption energies on the Li_2ZnCu_3 (001) surface and (d) corresponding adsorption sites. (e) The diffusion barriers of Li along the Li_2ZnCu_3 (001) surface.

After all, we have modified Fig. 1 as shown below and supplemented the correlated Figure, Table, and discussion in the revised manuscript and SI.

Fig. 1. Fabrication of Li/LiZn@Cu anode. (a) Schematic illustration of the fabrication process. The Li wettability for (b) molten Li and (c) molten Li-Zn mixture on Cu foils. (d) Thickness, (e-f) cross-section SEM images, (g) size, (h-i) SEM top-view images, (j) corresponding EDS mappings of the ultrathin Li/LiZn@Cu anode. (k) The calculated ternary phase diagram of the Li-Zn-Cu system at 473 K. (l) XRD pattern of the ultrathin Li/LiZn@Cu anode.

Pages 6-7 of the revised manuscript:

“Here, the atomic concentration of Li in Li/LiZn is $\sim 97.3\%$. Thus, the nanorod- and nanodot-like structures mainly correspond to the LiZn alloy skeletons as shown in Fig. 1j, which is verified by the

energy dispersive X-ray spectroscopy (EDS) measurement of the Li/LiZn@Cu surface, presenting the concentrative distribution of the Zn element around the nanostructures. As shown in Supplementary Fig. 11, such nanostructures are buried by the metallic Li, forming a 3D composite Li anode supported by alloy skeletons. Interestingly, the Cu element also exists on the nanostructures, where the atomic concentration is 10.91% as listed in Supplementary Table 2. As shown in Fig. 1k and Supplementary Figs. 12-14, the simulated ternary phase diagram and chemical potentials of the Li-Zn-Cu system forecast that the Li-Zn alloy, Li, and Cu substrate would re-alloy and convert into the rarely reported Li_2ZnCu_3 alloy, which is an energy-favorable phase at a wide temperature range. Thus, these nanostructures contain LiZn and a small amount of Li_2ZnCu_3 (~ 4.1%). As shown in Fig. 1l and Supplementary Figs. 15-16, the X-ray diffraction (XRD) peaks indicate the coexistence of LiZn alloy (PDF#03-0954), metallic Li (PDF#15-0401), Cu (PDF#04-0836) substrate and the newly formed Li_2ZnCu_3 alloy in the Li/LiZn@Cu anode. Excepting these peaks, no obvious metallic Zn peaks are detected to indicate the thorough Li and Zn alloying process. Of special importance is that the Li_2ZnCu_3 alloy mainly exists at the interface between the Li/LiZn layer and Cu substrate as shown in Supplementary Fig. 17.”

Page 13 of the revised manuscript:

“Additionally, given that the presence of a small amount of Li_2ZnCu_3 alloy in the Li/LiZn layer and its superior lithiophilicity, it will also exert positive influences on Li plating behaviors as shown in Supplementary Table 5 and Fig. 21.”

Page 23 of the revised manuscript:

“In addition, the e-Li/LiZn@Cu electrode can be obtained after etching off the Li/LiZn layer with alcohol.”

Page 27 of the revised manuscript:

“The surface energy σ of Li_2ZnCu_3 can be obtained as follows:

$$\sigma = \frac{1}{2A} [E_{\text{slab}} - nE_{\text{bulk}}] \quad (6)$$

where the E_{slab} is the energy of relaxed slab models with no restriction of the atomic position, the E_{bulk}

is the total energy of the bulk structure, and the A is the area surface.”

Page 13 of the revised SI:

“Due to the intermetallic compound LiZn alloy in the molten Li-Zn mixture, the LiZn alloy will be phase-segregated and form a 3D framework together with the Li_2ZnCu_3 alloy, in which the metallic Li fills the surrounding area of the framework after a cooling process. Thus, excepting for the nanodot-like structures, there are obvious nanorod-like (i.e. pillar-like) structures present on the Li/Li-Zn@Cu surface. It is worth noting that these nanodots correspond to the nanorods buried in the Li layer, resulting in the exposure of the top part only. As shown in Supplementary Fig. 11a-d, some nanorods can be observed inside the Li/LiZn layer. Moreover, due to a higher oxidization potential of LiZn alloy with respect to the metallic Li, the LiZn alloy will maintain its composition and crystal structure before the complete stripping of the surrounding Li ions to act firstly as a framework. Thus, after partial Li stripping, more previously buried alloy-based nanorods can be observed. (Supplementary Fig. 11e-h).”

Supplementary Fig. 11. (a) Cross-section SEM image and (b-d) the corresponding EDS mapping of the interface between the Li/LiZn layer and Cu substrate. (e-f) Top-view SEM images and (g-h) the corresponding EDS mapping of the Li/LiZn@Cu after partial Li stripping.

Page 14 of the revised SI:

Supplementary Table 2. EDS results of the Fig. 1j.

Element	Atom (%)
Zn	89.09
Cu	10.91

Supplementary Table 2 shows the EDS results of Fig. 1j, in which the concentration of Cu element is only 10.91%. According to the chemical formulas of Li_2ZnCu_3 and LiZn , the content of Li_2ZnCu_3 in alloy-based nanostructures is max. 4.1%.

Page 21 of the revised SI:

“The e-Li/LiZn@Cu electrode was obtained by immersing and rinsing with the alcohol solution. After the drying process, as shown in Supplementary Fig. 17a-c, differing from the red-orange color of the pristine Cu or silver color of the Li/LiZn@Cu, the sample after the detachment of Li/LiZn from Cu foil showed a different surface color. Due to the direct contact between the Cu surface and molten Li-Zn mixture, the re-alloying reaction and Li_2ZnCu_3 alloy mainly existed on the Cu surface. Moreover, the top-view SEM image of the e-Li/LiZn@Cu also displayed a pyramid-shaped morphology, which was completely different from Cu foil (Supplementary Fig. 17d), indicating the reconstruction of the smooth Cu surface by re-alloying reaction. The uniform distribution of Zn and Cu elements with a close atomic ratio of 1:3 between Zn and Cu evidenced that the surface composition of e-Li/LiZn@Cu is Li_2ZnCu_3 alloy (Supplementary Fig. 17e-g). The cross-section SEM and corresponding EDS mapping images indicated that the thickness of Li_2ZnCu_3 is $\sim 2 \mu\text{m}$ (Supplementary Fig. 17h-k). The pronounced Li_2ZnCu_3 peaks of the Rietveld refinement of the XRD pattern of the e-Li/LiZn@Cu sample further verified the preservation of the Li_2ZnCu_3 layer on the Cu surface (Supplementary Fig. 17l and Table 3). Based on the above characterizations and analysis, the Li_2ZnCu_3 alloy mainly present at the interface between the Li/LiZn layer and Cu substrate, not inside the Li/LiZn layer.”

Page 28 of the revised SI:

“Compared with the Li and LiZn, few literatures have focused upon the Li_2ZnCu_3 . The Li_2ZnCu_3 (001) is selected as the stable surface due to its lowest surface energy (Supplementary Table 5). As shown in Supplementary Fig. 21a-d, the Li absorption energy of Li_2ZnCu_3 (001) surface is -2.78 eV (the Li_2ZnCu_3 (001) surface possesses two possible adsorption structures towards Li), which is even lower than that of the LiZn (110) (-2.07 eV), not to mention the Li (100) surface. It suggests that the Li_2ZnCu_3 alloy possesses a stronger Li^+ ion affinity, which can effectively promote the Li nucleation and subsequently suppress the uneven Li deposition. However, as shown in Supplementary Fig. 21e, the

Li_2ZnCu_3 alloy does not have significant advantages in promoting the Li surface diffusion compared to the LiZn alloy (0.03 eV) and bare Li (0.06 eV), due to the same value (0.06 eV) of diffusion barrier alongside the Li_2ZnCu_3 (001) and Li (100) surfaces. Consequently, compared with the bare Li, the Li_2ZnCu_3 alloy can improve the uniform Li deposition to some extent due to its superior lithiophilicity, which can be further proved by the deposition morphology comparisons (plating 3 mAh cm^{-2} Li) of Li@Cu (prepared by electrochemical plating, Supplementary Fig. 22a-b) and Li@e-Li/LiZn@Cu (Li deposited e-Li/LiZn@Cu electrode, Supplementary Fig. 22c-d).”

2. The authors investigate the wettability of molten Li on Cu in Figures 1 and 6. The contents appear to be highly interrelated and should be combined for a coherent narrative.

Reply:

Thanks for your careful reading.

We have combined Figs. 1 and 6 for a coherent narrative in the revised manuscript. The initial Fig. 6 has been changed to Fig. 2.

Fig. 2. Economic Li composite anode by alloying. (a) Calculated gravimetric energy density and waste rate of Li resource of pouch cell with different thicknesses of Li metal anodes. The obtained data is based on Supplementary Table 4. Formation energies of the (b) Cu|Li and (c) Cu|LiZn interfaces. (d) Schematic illustration of the formation of LiZn and Li_2ZnCu_3 alloys and their corresponding Gibbs free energy (ΔG_f). (e) SEM top-view image of the Li_2ZnCu_3 alloy. (f) Factors influence the Li wettability with the LiZn and Li_2ZnCu_3 on the Cu substrate.

3. Could the authors experimentally measure the contact angles of molten Li and Li/LiZn on Cu or provide a supplementary video on the wetting process? The photographs in Figure 1b and 1c cannot be regarded as credible evidence.

Reply:

We have measured the contact angles of molten Li and Li/LiZn on the Cu substrate, as shown in Fig. R4. The contact angle of the molten Li-Zn mixture on the Cu substrate is $\sim 62^\circ$, showing a huge improvement in the Li wettability compared with molten Li ($\sim 110^\circ$).

Fig. R4 (Supplementary Fig. 18 in SI). Optic photos of the contact angles of the (a) molten Li and (b) Li/LiZn on Cu substrate.

The corresponding discussion is added to Page 10 of the revised manuscript as follows:

“Under the above synergetic effects, the contact angle of $\sim 62^\circ$ between molten Li-Zn mixture and Cu substrate is much smaller than that of $\sim 110^\circ$ between molten Li and Cu substrate as shown in Supplementary Fig. 18.”

4. In Li/LiZn@Cu, the EDS mapping (Fig. 1j) indicates that Zn are contained in the bcc Li phase. Would Zn atoms dissolved in the Li phase have an effect on the reaction behavior?

Reply:

*The EDS mapping (Fig. 1j) of the Zn element can not directly indicate that the Zn is contained in the bcc Li phase. As discussed in **Question 1**, the phase-segregation of LiZn alloy will construct the 3D framework in the Li/LiZn layer. As shown in Fig. R1a-c and k-n, some nanostructures consisting of LiZn and Li₂ZnCu₃ alloys can be observed on the surface of pristine Li/LiZn@Cu anode, while some nanostructures exist inside the Li/LiZn layer. After partial Li stripping, more buried Li alloy-based crisscrossed skeletons can be overserved (Fig. R1g). In fact, as discussed in **Question 1**, the phases of*

LiZn and Li are separated. Namely, the Li/LiZn@Cu anode has a composite structure, where the 3D LiZn alloy-based skeletons are distributed within the LiZn layer. As for the EDS characterization, although the information it collected is planar, it has a certain exploration depth, and the superimposed information reflects the distribution of Zn elements in 3D space. Therefore, the EDS mapping in Fig. 1j of the initial manuscript shows plausibly that the elemental Zn is contained in the bcc Li phase. In addition, because of such a unique composite structure containing the LiZn alloy-based framework, the Li/LiZn@Cu anode has positive effects, including the volume change alleviation and the Li plating/stripping regulation during the electrochemical reactions.

5. The authors claimed that the Zn-related phases contribute to a flat morphology when plating Li on Li/Li-Zn@Cu (Figure 2d-h). There should be control experiments of plating Li on Li@Cu to understand the differences.

Reply:

*We have performed the control experiments of plating Li on Li@Cu (prepared by electrochemical plating). Moreover, the control experiments of Li@Cu prepared by mechanical pressing (M-Li@Cu, the detailed discussion is demonstrated in **Question 7**) are also provided. As shown in Fig. R5, benefiting from the participation of LiZn alloy in the Li deposition process, more uniform Li morphology can be readily obtained compared with Li@Cu and M-Li@Cu anodes.*

Fig. R5 (Fig. R5a-b is the Supplementary Fig. 22a-b in SI, Fig. R5c-d is the Supplementary Fig. 34f-g in SI, and Fig. R5e-f is the Fig. 3h-I in the revised manuscript). SEM images of (a-b) Li@Cu, (c-d) M-Li@Cu, and (e-f) Li/LiZn@Cu anodes after Li deposition at 3 mAh cm^{-2} .

The corresponding part is modified on Page 13 of the revised manuscript and Page 44 of the revised SI as follows:

Page 13 of the revised manuscript:

“As shown in Fig. 3d-i, k, this is further attested from the SEM images of Li/LiZn@Cu after quantitative Li plating, where at a Li deposition of 0.5 mAh cm^{-2} , the homogeneous Li nuclei appear on the LiZn alloy surface and further at 1 and 3 mAh cm^{-2} , a dense and smooth Li layer is planted on the Li/LiZn@Cu anode without obvious dendrite growth, in sharp contrast to the Li@Cu anode as shown in Supplementary Fig. 22a-b.”

Page 43 of the revised SI

“As shown in Supplementary Fig. 34f-g, the surface of the M-Li@Cu electrode presents a Li intertwined morphology with many pores after plating 3 mAh cm^{-2} Li. Such an uneven Li morphology with a high exposure area to electrolyte will bring about excessive SEI formation and active material consumption, accompanied by the aggravated generation of the “dead Li” in subsequent cycles.”

6. The authors omitted the XRD peaks of Li_2ZnCu_3 from the in-situ XRD results (Fig. 2j). Would the Li_2ZnCu_3 phase changes during redox reactions?

Reply:

The Li_2ZnCu_3 phase would not change during redox reactions due to its excellent electrochemical stability.

To verify this viewpoint, we specially assembled $e\text{-Li/LiZn@Cu||Cu}$ (i.e. working electrode: $e\text{-Li/LiZn@Cu}$; counter electrode: Cu) and Cu||Cu cells, following a charging process at a constant charging current density for Li^+ ion extraction from Li_2ZnCu_3 alloy as much as possible ($25 \mu\text{A}$, a much lower value than the applied current density in symmetric and full cell tests in initial manuscript). As shown in Fig. R6a-b, the voltage profiles of $e\text{-Li/LiZn@Cu||Cu}$ and Cu||Cu cells are similar, both rapidly rising to 1.5 V in a short time, which indicates that the Li^+ ion can not be dealloyed from Li_2ZnCu_3 alloy. Interestingly, such voltage profiles are also similar to the charging of a capacitor. This capacitive ion storage behavior is related to the electric double-layer capacitance on the electrode

surface, and its current response is basically independent of the battery, which does not involve the phase transition caused by ion extraction in the bulk phase (*Energy Environ. Sci.* 2011, 4, 2614-2624). Moreover, the half cells with $\text{Li}||\text{e-Li/LiZn@Cu}$ and $\text{Li}||\text{Cu}$ configurations were further assembled to verify the stability of Li_2ZnCu_3 alloy during the discharging process. The $\text{Li}||\text{e-Li/LiZn@Cu}$ cell also shows a similar voltage profile to the $\text{Li}||\text{Cu}$ cell, in which the platform of voltage drop can be ascribed to the decomposition of electrolyte and formation of SEI (Fig. R6d-e). Furthermore, as shown in Fig. R6c, f, all disassembled electrodes do not significantly change in XRD patterns compared to the pristine e-Li/LiZn@Cu (Fig. R2i).

Fig. R6 (Supplementary Fig. 23 in SI). Voltage profiles of (a) $\text{e-Li/LiZn@Cu}||\text{Cu}$ and (b) $\text{Cu}||\text{Cu}$ cells after charging to 1.5 V at $25 \mu\text{A}$. (c) XRD patterns of the dissembled e-Li/LiZn@Cu electrodes after stripping to 1.5 V. Voltage profiles of (d) $\text{e-Li/LiZn@Cu}||\text{Cu}$ and (e) $\text{Cu}||\text{Cu}$ cells after discharging to 0 V at $10 \mu\text{A cm}^{-2}$. (f) XRD patterns of the dissembled e-Li/LiZn@Cu electrodes after discharging to 0 V.

To better validate the above results, we further assembled the in-situ $\text{Li}||\text{e-Li/LiZn@Cu}$ cell to study the electrochemical stability of Li_2ZnCu_3 alloy in depth. A thinner Cu foil ($6 \mu\text{m}$) was used as the substrate for better X-ray diffraction observation. The whole testing process can be divided into two parts, one of which is the Li discharging and charging with a voltage range from 0 to 1 V at $50 \mu\text{A cm}^{-2}$ and the other of which involves the Li plating with a fixed Li amount of 3 mA cm^{-2} and stripping to 1.5 V at 1 mA cm^{-2} . As shown in Fig. R7a-b, the peak intensities and positions assigned to the Li_2ZnCu_3 alloy are highly consistent without peak weakening or shifting during the Li discharging and charging

process, indicating that the Li_2ZnCu_3 is electrochemical inactive within the above-operated voltage. Moreover, the Li_2ZnCu_3 also maintains a stable phase throughout the Li plating and stripping process, even when the cut-off voltage is up to 1.5 V (Fig. R7c-d). According to the above comprehensive XRD and electrochemical analysis, we can conclude that the Li_2ZnCu_3 is an inert alloy in electrochemical tests, just like the Cu substrate, so that will not participate in electrochemical reactions like LiZn alloy. In addition, a shallow peak belonging to Li (110) gradually emerges during Li plating and fades after Li stripping (Fig. R7e). Such a low signal of Li can be attributed to the low content on the Cu substrate. Thus, due to the excellent electrochemical stability of the Li_2ZnCu_3 alloy and to reduce the interference of the Cu substrate on LiZn and Li characteristic peaks to better clarify the phase changes of the Li/LiZn layer during the Li stripping/plating process, we ignored the Li_2ZnCu_3 alloy and Cu substrate in our in-situ XRD characterization as shown in Fig. 2j of the initial manuscript. Consequently, based on the above comprehensive analysis, we can conclude that the Li_2ZnCu_3 alloy would not change during the redox reactions.

Fig. R7 (Supplementary Fig. 24 in SI). (a) *In-situ* XRD analysis of e-Li/LiZn@Cu electrode and (b) the corresponding contour maps during Li discharging and charging with a voltage range from 0 to 1 V at $50 \mu\text{A cm}^{-2}$. (c) *In-situ* XRD analysis of e-Li/LiZn@Cu electrode, (d) the corresponding contour maps, and (e) selected region of Li peaks during Li plating with a fixed Li amount of 3 mA cm^{-2} and stripping to 1.5 V at 1 mA cm^{-2} .

We supplemented the correlated discussion in the revised manuscript and SI as follows.

Page 14 of the revised manuscript:

“Furthermore, the *in-situ* XRD characterization furnishes an in-depth understanding of the Li stripping/plating behaviors in the Li/LiZn@Cu anode. Due to the excellent electrochemical stability of Li_2ZnCu_3 alloy and to prevent interference of Cu substrate on XRD signals as shown in Supplementary Figs. 23-24, the working electrode is set to be the Li/LiZn anode for simplification.”

Page 24 of the revised manuscript:

“In the in-situ Li||e-Li/LiZn@Cu cell, the cathode was e-Li/LiZn@Cu (Cu thickness: 6 μm) and the anode was Li foil.”

Page 29-31 of the revised SI:

“The phase stability of Li_2ZnCu_3 alloy during the electrochemical process is explored based on the e-Li/LiZn@Cu electrode. Firstly, the e-Li/LiZn@Cu||Cu (i.e. working electrode: e-Li/LiZn@Cu; counter electrode: Cu) and Cu||Cu cells were assembled to verify the stability during the Li charging process. A low constant charging current of 25 μA is applied for Li^+ ion extraction from the Li_2ZnCu_3 alloy as much as possible. As shown in Supplementary Fig. 23a-b, the voltage profiles of e-Li/LiZn@Cu||Cu and Cu||Cu cells are similar, both rapidly rising to 1.5 V in a short time, which indicates that Li^+ ions can not be dealloyed from the Li_2ZnCu_3 alloy. Interestingly, such voltage profiles demonstrate the capacitive behavior. This capacitive ion storage behavior is a non-Faradic process, and its current response is basically independent of the battery.² Moreover, the half cells with Li||e-Li/LiZn@Cu and Li||Cu configurations were further assembled to verify the stability of Li_2ZnCu_3 alloy during discharging process. The Li||e-Li/LiZn@Cu cell also shows a similar voltage profile to the Li||Cu cell, in which the platform of voltage drop can be ascribed to the decomposition of electrolyte and formation of SEI (Supplementary Fig. 23d-e). Furthermore, as shown in Supplementary Fig. 23c,f, all disassembled electrodes do not significantly change in XRD patterns compared to the pristine e-Li/LiZn@Cu (Supplementary Fig. 17l).”

“The *in-situ* Li||e-Li/LiZn@Cu cell is further assembled to study the electrochemical stability of Li_2ZnCu_3 alloy in depth. A thinner Cu foil (6 μm) was used as the substrate for better X-ray diffraction observation. The whole testing process can be divided into two parts, one of which is the Li discharging and charging with a voltage range from 0 to 1 V at 50 $\mu\text{A cm}^{-2}$ and the other of which involves the Li plating with a fixed Li amount of 3 mA cm^{-2} and stripping to 1.5 V at 1 mA cm^{-2} . As shown in Supplementary Fig. 24a-b, the peak intensities and positions belonging to Li_2ZnCu_3 alloy are highly consistent without peak weakening or shifting during the Li discharging and charging process, indicating that Li_2ZnCu_3 is electrochemical inactive within the above-operated voltage. Moreover, the

Li_2ZnCu_3 also maintains a stable phase throughout the Li plating and stripping process, even when the cut-off voltage is up to 1.5 V (Supplementary Fig. 24c-d). According to the above comprehensive XRD and electrochemical analysis, it can be concluded that the Li_2ZnCu_3 alloy is an inert alloy during the electrochemical tests, just like Cu substrate, so that will not participate in electrochemical cycling. In addition, a shallow peak assigning to the Li (110) plane gradually emerges during Li plating and fades after Li stripping (Supplementary Fig. 24e). Such a low signal of Li can be attributed to the low content on Cu substrate. Due to the excellent electrochemical stability of the Li_2ZnCu_3 alloy and to reduce the interference of the Cu substrate on LiZn and Li characteristic peaks to better clarify the phase changes of the Li/LiZn layer during the Li stripping/plating process, the working electrode of in-situ XRD characterization in Fig. 3j is set using the Li/LiZn anode for simplification.”

7. In the electrochemical experiments (Figure 3-5), how did the authors prepare the ultrathin Li@Cu anode? Would the different preparation process affect the material properties (e.g., grain size, dislocation, voids) and electrode performance?

Reply:

The ultrathin Li@Cu anode was prepared by the electrochemical plating method. Using Cu foil as the working electrode and Li foil as the counter electrode to assemble the Li||Cu half cell. Then a constant current density is applied for a discharging process to electrodeposit the Li with a fixed amount onto the Cu foil by controlling the electrodeposition time, and the ultrathin Li@Cu anode is obtained accordingly.

To make the preparation of Li@Cu more clear, we have added a more detailed preparation process in the experimental section (Page 24 of the revised manuscript) as follows

“To acquire the Li@Cu anode, a half cell using Cu foil as the working electrode and Li foil as the counter electrode was assembled, and then 4.5 mAh cm^{-2} Li was electroplated on the Cu surface at 0.5 mA cm^{-2} . After disassembling the half cell and rinsing the Li-plated Cu foil with DOL, the preparation of the Li@Cu anode was completed.”

Generally speaking, there are three methods for preparing the ultrathin Li anodes, including the electrodeposition fabrication, the molten Li reprocessing, and the mechanical rolling. Different

preparation processes can affect the material properties and electrode performance to a certain extent. For example, compared with the molten Li reprocessing and the mechanical rolling, the electrodeposition-based Li anode will be exposed to the electrolyte during fabrication, inevitably introducing unnecessary by-products into the Li anode and making the pre-loss of active Li. Moreover, the electrodeposited Li on the Cu current collector can not maintain a homogeneous morphology but a rugged Li surface with voids, thereby stimulating the generation of “dead Li”. Therefore, especially in the ultrathin Li anode, the inactive Li occupies a greater proportion of the anode, giving an inferior cycling performance to the molten Li reprocessing- and mechanical rolling-based ultrathin Li anodes. For the mechanical rolling, some deep-rooted problems in the metallic Li when it withstands mechanical deformation. Whereby the strong influence of diffusion creep caused by high homologous temperature, the metallic Li becomes highly sticky and poor mechanical processibility, posing a great challenge in producing ultrathin Li anodes. Especially when preparing Li anode with a thickness of less than 50 μm , the requirements for the accuracy of the rolling machine and preparation cost and difficulty will increase sharply. Moreover, the Li can easily induce mechanical scratches on its soft surface during the rolling processing, resulting in the Li^+ ions tending to deposit at these uneven protrusions. As for the molten Li reprocessing, tuning the surface tension of molten Li is a crucial step in achieving ultrathin Li anode. Currently, doping molten Li with impurities to form the Li-based alloys is one of the popular methods, whereas the Li-based alloys usually have better mechanical strength and deposited Li regulation compared to the pure Li.

Given that the ultrathin Li anodes prepared by electrodeposition and molten Li reprocessing have been systematically characterized in the manuscript previously, we added the mechanical rolling-based ultrathin Li@Cu anode (named as the M-Li@Cu and purchased from China Energy Lithium Co., Ltd) for further research. As shown in Fig. R8a-b, the M-Li@Cu possesses visible mechanical indentations on the surface. The XRD pattern exhibits that the M-Li@Cu only contains the metallic Cu and Li (Fig. R8c). The thickness of M-Li@Cu is $\sim 34 \mu\text{m}$ which is similar to the Li/LiZn@Cu electrode ($35\mu\text{m}$), in which the bottom region is $10 \mu\text{m}$ -thick Cu substrate and the top region is $24 \mu\text{m}$ -thick metallic Li layer (Fig. R8d-e). Compared with the Li/LiZn layer, the bare Li layer can not realize the ordered Li grow regulation due to its poor Li adsorption ability and large diffusion barrier for Li lateral deposition, as verified in Fig.3a-c of the revised manuscript. As shown in Fig. R8f-g, the surface of the M-Li@Cu

electrode presents a Li intertwined morphology with many pores after plating 3 mAh cm^{-2} Li. Such an uneven Li morphology with a high exposure area to electrolyte will bring about excess SEI formation and active material consumption, accompanied by the aggravated generation of the “dead Li” in subsequent cycles. Consequently, after 50 cycles at $1 \text{ mA cm}^{-2}/1 \text{ mAh cm}^{-2}$ in the symmetric cell with $\text{M-Li@Cu}||\text{M-Li@Cu}$ configuration, the worse Li morphology can be observed with the partial Li layer detaches from the substrate (Fig. R8h-k). This explains the poor long-term cycling performance of the $\text{M-Li@Cu}||\text{M-Li@Cu}$ cells, in which the voltage violently fluctuates after only 250 h, whereas the symmetric Li/LiZn@Cu cell delivers a stable voltage platform for 690 h (Fig. R8i). It is worth mentioning that the cycling performance of M-Li@Cu is much better than that of Li@Cu which is prepared by electrochemical plating. As the reviewer pointed out, the Li@Cu prepared by electrochemical plating will inevitably introduce a rugged morphology and “dead Li”, thereby resulting in an inferior Li/electrolyte interface and electrochemical performance than M-Li@Cu in long-term cycles. The EIS analysis also demonstrates this viewpoint (Fig. R8n-p). Although both M-Li@Cu and Li@Cu exhibit an increase in R_{SEI} (corresponding to the semicircle at high frequency) after 50 cycles, the M-Li@Cu anode still has a smaller value than Li@Cu one, as well as its better rate performance in Fig. R8m. Furthermore, the M-Li@Cu -based full cell with a high-loading LFP cathode (2 mAh cm^{-2}) was assembled to conduct a systematic evaluation of M-Li@Cu for electrochemical stability. As shown in Fig. R8q, the $\text{M-Li@Cu}||\text{LFP}$ cell delivers a higher capacity retention of $\sim 80.0\%$ after 65 cycles than $\text{Li@Cu}||\text{LFP}$ (54.4% , 60 cycles), but is much inferior to Li/LiZn@Cu cell.

To sum up, benefiting from the more uniform morphology of initial Li and no obvious generation of “dead Li” in the preparation process, the M-Li@Cu exhibits a better electrochemical performance than that of the Li@Cu anode, no matter in symmetric or full cell configuration. However, the poor Li adsorption and surface diffusion ability of M-Li@Cu also restrict its cycling performance, showing a significant gap compared with Li/LiZn@Cu electrode.

Fig. R8. (a) SEM top-view images, (b) XRD pattern, and (c) SEM cross-section images of M-Li@Cu composite anode. Morphology evolution of M-Li@Cu after (f-g) plating 3 mAh cm^{-2} Li and (h-i) 50 cycles in symmetric cell at $1 \text{ mA cm}^{-2}/1 \text{ mAh cm}^{-2}$ and (j-k) corresponding SEM cross-section SEM images. Comparisons of cycling performance of M-Li@Cu-, Li@Cu-, and Li/LiZn@Cu-based symmetric cells in (l) long-term cycling, (m) rate, and (n-p) EIS tests. (q) Comparisons of the cycling performance of M-Li@Cu-, Li@Cu-, and Li/LiZn@Cu-based full cells.

While, based on the comments from the referee, the related discussion and Figure have been added in the revised manuscript and SI as follows.

Page 3 of the revised manuscript

“In addition, it is easy to generate mechanical scratches on the soft Li surface during the rolling process, resulting in the Li^+ ions tending to deposit at these uneven protrusions. Thus, when preparing Li anode with a thickness of less than $50 \mu\text{m}$, the requirements for the accuracy of the rolling machine and preparation cost and difficulty will increase sharply.”

Page 20 of the revised manuscript

“Additionally, considering that the inevitable introduction of rugged Li morphology and “dead Li” in

Li@Cu anode during electrochemical preparation, which will exert innate disadvantages in full cell tests, the mechanical rolling-based ultrathin Li@Cu (M-Li@Cu, $\sim 24 \mu\text{m}$ thick Li) anode is further set to be the control group as shown in Supplementary Fig. 34. When paired with high-loading LFP (13.1 mg cm^{-2} , corresponding to $\sim 2 \text{ mAh cm}^{-2}$, N/P = 2.2), the prominent performance of Li/LiZn@Cu||LFP full cell is also inherited, where it displays enhanced capacity retention of 90.1% and a prolonged lifespan with respect to those of the Li@Cu||LFP and M-Li@Cu||LFP cells as shown in Supplementary Fig. 34o.”

Page 24 of the revised manuscript

“The ultrathin M-Li@Cu anode (thickness: $\sim 24 \mu\text{m}$ -thick Li+ $10 \mu\text{m}$ -thick Cu foil) was purchased from China Energy Lithium Co., Ltd.”

Page 42-44 of the revised SI

“The electrodeposition-based Li@Cu will be exposed to the electrolyte during fabrication, inevitably introducing unnecessary by-products into Li anode and making the pre-loss of active Li. To exclude the introduction of rugged Li and “dead Li” in preparation, the mechanical rolling-based ultrathin Li@Cu anode (named the M-Li@Cu and purchased from China Energy Lithium Co., Ltd) is also considered as the control group. As shown in Supplementary Fig. 34a-b, the M-Li@Cu possesses visible mechanical indentations on the surface. The XRD pattern exhibits that the M-Li@Cu only contains the metallic Cu and Li (Supplementary Fig. 34c). The thickness of M-Li@Cu is $\sim 34 \mu\text{m}$ which is similar to the Li/LiZn@Cu electrode ($35 \mu\text{m}$), in which the bottom region is $10 \mu\text{m}$ -thick Cu substrate and the top region is $24 \mu\text{m}$ -thick metallic Li layer (Supplementary Fig. 34d-e). Compared with the Li/LiZn layer, the bare Li layer can not realize the ordered Li growth regulation due to its poor Li adsorption ability and large diffusion barrier for the Li lateral deposition, as verified in Fig. 3a-c. As shown in Supplementary Fig. 34f-g, the surface of the M-Li@Cu electrode presents a Li intertwined morphology with many pores after plating 3 mAh cm^{-2} Li. Such an uneven Li morphology with a high exposure area to electrolyte will bring about excessive SEI formation and active material consumption, accompanied by the aggravated generation of the “dead Li” in subsequent cycles. Consequently, after 50 cycles at $1 \text{ mA cm}^{-2}/1 \text{ mAh cm}^{-2}$ in the symmetric cell with M-Li@Cu||M-

Li@Cu configuration, the worse Li morphology can be observed with the partial Li layer detaches from the substrate (Supplementary Fig. 34h-k). This explains the poor long-term cycling performance of the M-Li@Cu||M-Li@Cu cells in Supplementary Fig. 34l, in which the voltage violently fluctuates after only 250 h, whereas the symmetric Li/LiZn@Cu cell delivers a stable voltage platform for 690 h. It is worth mentioning that the cycling performance of M-Li@Cu anode is much better than that of Li@Cu one which is prepared by electrochemical plating. It can be ascribed to the more dense and uniform Li layer on the pristine M-Li@Cu than that of Li@Cu, thereby resulting in a superior Li/electrolyte interface and electrochemical performance than that of the Li@Cu anode in long-term cycles. The EIS analysis also demonstrates this viewpoint. As shown in Supplementary Fig. 34n, although both M-Li@Cu and Li@Cu exhibit an increase in R_{SEI} (the semicircle at high frequency) after 50 cycles, the M-Li@Cu still has a smaller value than that of the Li@Cu (Supplementary Fig. 31a), as well as its better rate performance in Supplementary Fig. 34m. Furthermore, the M-Li@Cu-based full cell with a high-loading LFP cathode (2 mAh cm^{-2}) was assembled to conduct a systematic evaluation of M-Li@Cu for electrochemical stability. As shown in Supplementary Fig. 34o, the M-Li@Cu||LFP cell delivers a higher capacity retention of $\sim 80.0\%$ after 65 cycles than the 54.4% after 60 cycles of Li@Cu||LFP cell, but is much inferior to the Li/LiZn@Cu cell.

Benefiting from the uniform morphology of initial Li and no generation of “dead Li” in the preparation process, the M-Li@Cu exhibits a better electrochemical performance than that of the Li@Cu, no matter in symmetric or full cell configuration. However, the poor Li adsorption and surface diffusion ability of M-Li@Cu also restrict its cycling performance, showing a significant gap compared with the Li/LiZn@Cu electrode.”

Supplementary Fig. 34. (a) SEM top-view images, (b) XRD pattern, and (c) SEM cross-section images of M-Li@Cu. Morphology evolutions of M-Li@Cu after (f-g) plating 3 mAh cm⁻² Li and (h-i) 50 cycles in symmetric cell at 1 mA cm⁻²/1 mAh cm⁻² and (j-k) corresponding SEM cross-section images. Comparisons of the M-Li@Cu-, Li@Cu-, and Li/LiZn@Cu-based symmetric cells in (l) long-term cycling, (m) rate, and (n) EIS tests. (o) Comparisons of the cycling performance of M-Li@Cu-, Li@Cu-, and Li/LiZn@Cu-based full cells with a high-loading LFP cathode (~2 mAh cm⁻²).

8. How did the authors determine the dendrites in the electrodeposits in Figure 5b-c? The highlighted spots in the SEM image appear to be flakes rather than conventional Li dendrites.

Reply:

We have already modified the highlighted content as shown in Fig. 5b-c to express it accurately. As shown in Fig. R9b-c, the “Li dendrites” in the highlighted spots has been replaced with “Uneven morphology”, while the “No dendrites” has been replaced with “Even morphology”.

Fig. R9 (Fig. 6 of the revised manuscript). Electrochemical performance of Li@Cu and Li/LiZn@Cu full cells. (a) Cycling performance of Li@Cu||LFP and Li/LiZn@Cu||LFP full cells with an N/P ratio of ~ 3.4 at 0.5 C between 2.8 and 4.2 V. SEM top-view images of the (b) Li@Cu and (c) Li/LiZn@Cu anodes after the 30th cycle in the LFP-based full cells. (d) Rate performance of Li@Cu||LFP and Li/LiZn@Cu||LFP full cells. The thickness of Cu foil is ~ 10 μm . (e) Cycling performance and (f-g) corresponding voltage profiles of Li@Cu||LCO and Li/LiZn@Cu||LCO full cells with an N/P ratio of ~ 2.8 at 0.5 C between 2.8 and 4.5 V. (h) Cycling performance of Li/LiZn@Cu||LCO full cell with a lean electrolyte of 12 μL and (i) corresponding voltage profiles for selected cycles at 0.1 C. (j) Schematic illustration and (k) corresponding cycling performance of the Li/LiZn@Cu||LCO pouch cell. The thickness of Cu foil is ~ 6 μm .

9. In Figure 6a, why does the waste rate of Li resource become 0 when $N/P \leq 3$? Since the cathode (NMC or LCO) contains sufficient Li ions at the beginning, any Li on the anode ($N/P > 0$) would be a theoretical excess Li resource.

Reply:

Thanks for your comment. We have modified Fig. 6a and set the waste rate of Li resource to 0 when the N/P ratio is 0 as follows.

Fig. R10. Calculated gravimetric energy density and waste rate of Li resource of pouch cell with different thicknesses of Li metal anodes.

The corresponding Figure is modified in Fig. 2 (see Question 2) of the revised manuscript and Pages 22-23 of the revised SI as follows:

“Take Li metal anode with a maximum thickness of 400 μm as an example. The areal capacity is

$$\frac{400}{4.85} \text{mAh cm}^{-2} = 82.47 \text{mAh cm}^{-2}$$

For the waste rate of Li resource (R_w , when N/P=0, the R_w is 0), the R_w is defined as follows

$$R_w = \frac{n \times 3.11}{82.47}$$

where the n is the value of the applied N/P ratio.

Thus, the R_w of 400 μm-thick Li metal anode is 100%, while the value is 11.3% when the N/P ratio is 3 (corresponding to 45.3 μm-thick Li metal anode).”

“If the thickness of the Li metal anode can be further reduced to less than 50 μm, the corresponding N/P ratio is ~3.3, close to the ideal value of less than 3. At that time, the energy density of the pouch cell can increase to more than 315 Wh kg⁻¹. The energy density can be further enhanced after increasing the stacked layers and optimizing the weight of the electrolyte, cathode, current collector, etc.”

Reviewer #2 (Remarks to the Author):

In the present manuscript, the authors have delineated the fabrication of a composite anode utilizing a Li/LiZn@Cu alloy configuration. The manuscript posits that the strategic inclusion of other metallic constituents, notably Zn in this case, within the molten Li medium engenders heightened wettability towards the underlying Cu matrix. A commendable feature of this work is the authors' adept elucidation of the rationale behind the selection of Zn, an aspect thoughtfully expounded upon in the introductory segment. Furthermore, the manuscript combines both static and dynamic calculations, providing a comprehensive elucidation of the superior lithium plating characteristics inherent to their devised matrix. Also, an impressive enhancement in electrochemical performance is demonstrated. I appreciate that the authors did a lot work and provided some insights. However, several significant issues give rise to my reservations regarding its suitability for publication within the esteemed confines of a high-impact journal such as Nature Communications.

I. Fundamental errors

1 Kindly ensure consistent usage of the abbreviation for "Fig." throughout the entire manuscript. In certain instances, the abbreviation "Fig." has been utilized, while in others, "Figs." has been employed. Maintaining uniformity in this aspect contributes to the overall clarity and professionalism of the manuscript.

Reply:

Thanks for your suggestion.

The "Figs." is an abbreviation for "Figures". The uniformity of the abbreviation has been checked again.

2 An observation has been made regarding the absence of proper journal-title abbreviations for journals of a physical nature, such as "Physical Review B" (which should be abbreviated as "Phys. Rev. B").

Reply:

Thanks for your careful reading.

We have modified the unabbreviated journal name with the correct abbreviations.

3 The Gibbs reaction free energy values presented in this manuscript are derived from meticulous DFT calculations using VASP. It is important to recognize that these calculations are conducted under the assumption of a temperature of 0 K. Therefore, I draw attention to the text fragment at line 341, which references "The alloying reaction between Li and Zn at 250 °C," followed by the accompanying calculated reaction energy. It is pertinent to acknowledge that the disparities between reaction scenarios at significantly elevated and lowered temperatures can be substantial. I propose that the alloy reaction can be spontaneous based on the phase diagram, but urge a thoughtful consideration of this misleading expression.

Reply:

The comments by the reviewer are substantially meaningful in characterizing the materials here, which are actually the different knowledge between experimental and calculation methods. In this manuscript, the DFT calculations are conducted at 0 K. While, the material synthesis and chemical reactions basically occur at elevated temperatures, at least above the room temperature. The temperature, as an important factor, plays a vital role in materials synthesis and chemical reaction at all time.

In fact, obtaining the Gibbs formation energies $\Delta G_f(T)$ of potential materials is critical for predicting the synthesizability and stability of materials at conditions of interest for numerous applications. The ab initio computational approaches for determining the $\Delta G_f(T)$, which involve calculating the vibrational contributions to $G(T)$ as a function of volume, have benefited to recent advances (Angew. Chem. Int. Ed. 2010, 49, 5242–5266). Moreover, the use of machine learning and data analytics to accelerate materials design and discovery through descriptor-based property prediction has become a standard approach in materials science. Bartel and his coworker predict the Gibbs energies at elevated temperatures of inorganic crystalline solids using the SISO (sure independence screening and sparsifying operator) approach (Nature Communications, (2018) 9:4168). These techniques have been integrated into the pymatgen as the `GibbsComputedStructureEntry` class. Note that this implementation also includes the changes in elemental chemical potentials as a function of temperature, as well as the thermochemistry data for some known cases.

Here, the reaction energy of the alloying reaction between Li and Zn at 250 °C was calculated to explore the possibility of the synthesizability of LiZn alloy. Although the alloy reaction can be spontaneous based on the phase diagram (see Supplementary Figs. 12-14 in SI), it is necessary for Li

and LiZn to become molten conditions to advance the atomic diffusion and ensure the entire alloying reaction. The chemical reaction requires both thermodynamic and kinetic-driven forces, which can be greatly enhanced at a higher temperature.

To make this presentation clearer, the related discussion is supplemented in Pages 9 and 26 of the revised manuscript as follows:

Page 9 of the revised manuscript

“Note that the released Gibbs free energy (ΔG_r) at high temperature is estimated using machine learning model SISO (sure independence screening and sparsifying operator) approach.³⁹”

Page 26 of the revised manuscript

“The Gibbs free energy $G(T)$ of the formation of solids at high temperatures (200 or 250 °C) can be estimated using the machine learning model SISO (sure independence screening and sparsifying operator) approach.³⁹”

4 To calculate the interface energy of two interfaces, the authors using the equation ” $E_f = (E_{ab} - E_a - E_b)/S$ ”, where the E_a and E_b represent the slab energy. However, it should be noted that in the Prof. Qi ‘s work (J. Electrochem. Soc., 163 (3) A592-A598 (2016)), the interface energy is defined as the “The difference between the energy of the fully relaxed interface structure and the energy sum of the two stress free pure phases with the same atomic unit numbers”. Therefore, the E_a and E_b ought to be the energy per unit of the stress-free pure A and B bulk structure, respectively.

Reply:

Thank the reviewer for the kind reminder. This is actually not a big problem in the Li alloy system, because it is very easy to relax to a stress-free pure bulk phase, or surface structure for subsequent simulations at the beginning of the calculation.

A dispute is that an interface energy of two thermodynamic interfaces might sometimes not correspond to a real chemical reaction process during the electrochemical cycles, cause the Li always demonstrates the fluid-like feature and experiences the violent charge transfer and structural

adjustments. This makes it difficult to accurately determine the interface energy of Li-dominated systems.

Then, the $E_f = (E_{ab} - E_a - E_b)/S$, where the E_a and E_b represent the slab energy, was conducted to calculate the interface energy of the two interfaces. We agree with that “the difference between the energy of the fully relaxed interface structure and the energy sum of the two stress free pure phases with the same atomic unit numbers” as a standard model for calculating the slab energies of E_a and E_b (*J. Electrochem. Soc.*, 163 (3) A592-A598 (2016)).

In detail, for an interfacial supercell (contains A and B atoms), the interface formation energy, E_f , can be written as

$$E_f = E_{AB} - N_A E_A - N_B E_B \quad (1)$$

where E_{AB} is the total energy of the fully relaxed interfacial supercell, containing N_A units of A and N_B units of B. The E_A and E_B are the energy per unit of the stress-free pure A and B bulk structure, respectively.

The interfacial energy can be then calculated by

$$\sigma = \frac{E_{AB(xy z)} - N_A E_{A(z)} - N_B E_{B(z)}}{2S} \quad (2)$$

where $E_{AB(xy z)}$ is the fully-relaxed total energy of the interfacial structure. The $E_{A(z)}$ and $E_{B(z)}$ are the energies per atomic layer of the pure A and B bulk structures after constrained relaxation along interface normal direction (z direction) with fixed x and y lattice vectors.

The former definition gives the energy about the joined A and B slab to form a coherent interface, and the latter one demonstrates that the interface formation energy is separated into two parts (strain energy and interfacial energy). The prerequisite is that the lattice mismatch of A and B slabs needs to be small (generally less than 5.0 %).

To fully evaluate the compatibility of Cu|Li and Cu|Li-Zn interfaces, the interface formation energy E_f and interfacial energy σ , are also provided. Using the method as mentioned above, the LiZn interfaces show only a quarter of the interfacial energy of Li, which indicates that the Cu current collector is more mechanically stably covered by the LiZn alloy, other than the Li metal (Fig. R11).

Fig. R11. Atomic structures of the as-constructed supercells of (a) Cu(001)/Li(001) interface and (b) Cu(001)/LiZn(110) interface with the corresponding interfacial energy.

The reference J. Electrochem. Soc., 163 (3) A592-A598 (2016) is cited as Ref. 40.

[40] Liu Z, Qi Y, Lin Y, Chen L, Lu P, Chen L. Interfacial study on solid electrolyte interphase at Li metal anode: implication for Li dendrite growth. *J Electrochem Soc* **163**, A592 (2016).

II. Contradictions

1 DFT calculation results

The authors have employed the VASP package for their Density Functional Theory (DFT) calculations. However, notable disparities between their obtained results and the reported values have been observed. I highlight two distinct aspects that warrant elucidation:

a) Reaction Energy:

The manuscript details the computation of Gibbs reaction free energies for two reactions: Reaction 1 ($\text{Li} + \text{Zn} = \text{LiZn}$) and Reaction 2 ($\text{Li} + \text{LiZn} + 3\text{Cu} = \text{Li}_2\text{ZnCu}_3$), yielding calculated values of -0.12 eV and -0.03 eV, respectively. In contrast, calculations performed using the Materials Project module's Reaction Calculator yield values of -0.44 eV and -0.321 eV for the respective reactions. While both sets of results exhibit negative values, indicative of spontaneous reactions, the substantial deviation

casts doubts upon the reliability of the DFT calculations in this manuscript. A thorough explanation of the causes underlying these significant discrepancies is imperative.

Reply:

Thanks for your interesting.

First of all, the thermodynamic properties of the materials containing Li, Zn, and Cu elements at 0 K were obtained using Density Functional Theory (GGA/GGA+U) calculations.

Table R2. The thermodynamic properties of the materials contain Li, Zn, and Cu elements at 0 K.

	formula	form_energy (eV/atom)	decomp_enthalpy (eV/atom)
0	LiZn	-0.218680	-0.081384
1	LiZn3	-0.171035	-0.060308
2	Li2ZnCu3	-0.126813	-0.030800
3	Zn8Cu5	-0.108454	-0.010717
4	Zn35Cu17	-0.101228	-0.004592
5	ZnCu	-0.090755	-0.002636
6	Zn3Cu	-0.080880	-0.003470
7	LiCu3	-0.034679	-0.034679
8	Li	0.000000	0.000000
9	Cu	0.000000	0.000000
10	Zn	0.000000	0.000000

Then, the released Gibbs free energy (ΔG_r) about the formation of LiZn and Li₂ZnCu₃ at 0 K are calculated as follows.

Next step is incorporating the temperature-dependence with a previously derived machine learning (ML) model (SISSO) and experimental data.

The Materials Project (MP) contains the DFT-calculated energies for compounds at 0 K. Since most experimental syntheses take place well above 0 K, it is useful to have some idea of how the phase

diagram (and hence reaction energies, relative stabilities, etc.) change with increasing temperatures. A recent work by Bartel et al.^[1] derived an equation to estimate the Gibbs free energy of the formation of solids, $\Delta G_f(T)$, as a function of temperature using the MP formation enthalpy as a reference. Note that the formation enthalpy calculated from MP phase diagrams is actually an estimate of the formation enthalpy at $T = 298\text{ K}$ due to the specific corrections applied to the raw DFT energies. This has been implemented within pymatgen as the ‘GibbsComputedStructureEntry’ class. This implementation also includes the change in elemental chemical potentials as a function of temperature, as well as the thermochemistry data for some known gases.

Fig. R12. The convex hull about the formation of LiZn at 523 K using the ‘InterfacialReactivity’ class in pymatgen and the calculated reaction energy.

The phase diagram slice approach depicted in the manuscript has been implemented into pymatgen within the ‘InterfacialReactivity’ class, and the convex hull about the formation of LiZn at 523 K is plotted in Fig. R12. The reaction energy of -0.123 eV/atom was obtained after using the ML model (SISSO). In the table, we see the reactions as a function of the normalized mixing ratio, along with their energy in kJ per mole of reaction, as well as eV per (reactant) atom.

The free energies of the materials containing Li, Zn, and Cu elements at 523 K were calculated as

follows.

GibbsComputedStructureEntry mp-30-GGA - Cu1

Gibbs Free Energy (Formation) = 0.0000

GibbsComputedStructureEntry mp-79-GGA - Zn2

Gibbs Free Energy (Formation) = 0.0000

GibbsComputedStructureEntry mp-1222617-GGA - Li2 Zn1 Cu3

Gibbs Free Energy (Formation) = -0.4166

GibbsComputedStructureEntry mp-1934-GGA - Li2 Zn2

Gibbs Free Energy (Formation) = -0.4912

*Then the the released Gibbs free energies (ΔG_r) at 523 K were shown below. Reaction 1 (Li+Zn=LiZn) and Reaction 2 (Li+LiZn+3Cu=Li₂ZnCu₃), yielding calculated values of **-0.12 eV/atom** and **-0.03 eV/atom**, respectively, as shown below:*

Although it involves the greatest decrease in free energy, the reaction given by the red star is still favorable and may occur at higher temperatures/later times as shown in Fig. R12.

The detailed calculation method and corresponding part are supplemented into the revised manuscript to make this part clearer as follows:

Page 9-10 of the revised manuscript

“The alloying reaction between Li and Zn at 250 °C is as follows:

Note that the released Gibbs free energy (ΔG_r) at high temperature is estimated using machine learning model SISSO (sure independence screening and sparsifying operator) approach.³⁹”

“As shown in Fig. 2d, this improved Cu|Li-Zn interface contacts help the subsequent alloying process of Li-Zn, Cu, and excessive Li at 200 °C to generate the ternary Li₂ZnCu₃ phase as follows:

“The Gibbs free energy $G(T)$ of the formation of solids at high temperatures (200 or 250 °C) can be estimated using the machine learning model SISSO (sure independence screening and sparsifying operator) approach.³⁹”

Reference:

[1] Bartel, C. J.; Millican, S. L.; Deml, A. M.; Rumpitz, J. R.; Tumas, W.; Weimer, A. W.; Lany, S.; Stevanović, V.; Musgrave, C. B.; Holder, A. M. Physical descriptor for the Gibbs energy of inorganic crystalline solids and temperature-dependent materials chemistry. *Nat. Commun.* 2018, 9 (1), 4168. DOI: 10.1038/s41467-018-06682-4.

b) Diffusion Barrier:

It is worth noting that literature encompasses several reports on calculations of Li⁺ diffusion barriers on both pure Li and Li-alloy matrices, achieved through the CI-NEB method. Notably, Prof. Liang's work (*Energy Storage Mater.* 32 (2020) 178–184) has meticulously presented migration barrier values for Li⁺ on Li (100) and LiZn (110). Prof. Liang's research demonstrates that the Li⁺ migration barrier on Li (100) approximates ~0.14 eV, while on LiZn(110), the value stands at ~0.023 eV. However, intriguingly, in the present manuscript, the corresponding calculated values are 0.06 eV for Li (100) and 0.03 eV for LiZn (110). This substantial disparity prompts the need for an in-depth clarification concerning the deviation from established values in the literature. A comprehensive elucidation of the methodology, potential sources of error, and other contributing factors is indispensable in order to address this substantial deviation and enhance the credibility of the study's findings.

Reply:

We agree that the credibility of Li⁺ ion diffusion barriers needs to be clarified urgently. Note that the slab model, choice of exchange-correlation Functional, and convergence criteria will influence the reproducibility of material properties.

*The reviewer raises the deviation between Prof. Liang's research (*Energy Storage Mater.* 32 (2020) 178–184) and our calculation results. Hence, we drive into the details of the computational method and the slab model. However, Prof. Liang's paper does not disclose the detailed method. While, the*

references are provided in the discussion section. The migration energy barrier calculated by Liang et al. is highly consistent with the previous literature (*J. Chem. Phys.* 141 (17) (2014) 174710–174717, *Energy Environ. Sci.* 11 (12) (2018) 3400–3407). The original calculation methods present as follow “The Brillouin zone integration for bulk calculations has been performed on a $9 \times 9 \times 9$ k -point grid. The electrode surfaces are modeled by 5-layer slabs.” In each case, the three bottom layers were fixed at bulk positions while all other layers were allowed to relax.

Fig. R13. (a) The energy barriers of lithium diffusion on the surface of lithium metal (100). (b) The calculated corresponding minimum energy paths (hollow-bridge-hollow).

As a result, the diffusion barrier of 0.15 eV at the Li (100) surface is reproduced with the 5-layer slab as shown in Fig. R13, which agrees with Prof. Liang's results. While, the 5-layer slab is too thin to simulate the surface migration process, which is also a physically incorrect result for a real surface. Moreover, the choice of the slab thickness will have a great impact on the calculation results.

Based on the above consideration, the optimized 8-layer slabs were utilized for the subsequent calculation in this work. The four bottom layers were fixed at bulk positions while all other layers were allowed to relax. The calculated values of 0.06 eV for Li (100) and 0.03 eV for LiZn (110) are presented in Fig. 3b-c in the revised manuscript. In addition, our calculated values are in perfect accordance with the literature (*ACS Energy Lett.* 2019, 4, 2952-2959).

The detailed calculation method is added to Page 26 of the revised manuscript as follows:

“The optimized 8-layer slab was utilized for the calculation of the Li-ion migration barriers, where the four bottom layers were fixed at bulk positions while all other layers were allowed to relax.”

2 Research subject

This constitutes a central discrepancy within the manuscript's framework.

It appears that the authors advance the proposition that the LiZn alloy is pivotal in governing Li⁺ plating behaviors, substantiated by their in-situ characterizations and theoretical derivations. Additionally, the function of Li₂ZnCu₃ is ascribed to aiding the diffusion of Li to the Cu foil. However, scrutiny of Fig. 1k reveals an intriguing pattern wherein the signal intensities of Li₂ZnCu₃ and Cu are notably more pronounced than those of Li and LiZn. This suggests a continuous interaction between LiZn, Li, and Cu, culminating in the formation of Li₂ZnCu₃ upon doctor-blading the mixtures at a temperature of 200 °C. Consequently, it becomes apparent that Li₂ZnCu₃ takes precedence as the principal constituent within the so-called Li-Zn/Cu composite anode.

A noteworthy aspect emerges during subsequent in-situ XRD characterizations, wherein the absence of a Li₂ZnCu₃ peak is perplexing. Closer examination reveals that the electrodes employed for these characterizations are formed by pressing mixtures onto the foil, rather than blade mixing onto the Cu foil. This shift in methodology implies that the predominant subject within the composite anode becomes Li-Zn. However, it is essential to note that the composite electrodes employed for later electrochemical assessments are prepared using the doctor-blading approach. Consequently, it logically follows that Li₂ZnCu₃, not LiZn, should predominantly influence electrochemical performance in these composite electrodes.

At this juncture, a critical departure from rigor is identified. All calculations presented in the manuscript pivot on LiZn, neglecting the dominant role that Li₂ZnCu₃ assumes in the subsequent electrochemical evaluations. A reevaluation of characterization results and a meticulous reconstruction of the manuscript's workflow are thus imperative.

Reply:

Thanks for your careful review.

At the beginning, we primarily emphasized the important role of LiZn during Li deposition/stripping, substantiated by their in-situ XRD characterizations and theoretical derivations. Indeed, in Fig. 1k of the initial manuscript, the peak intensities of Li₂ZnCu₃ and Cu are notably more pronounced than those of Li and LiZn components. However, this only indicates that the Li₂ZnCu₃ alloy presents within the Li/LiZn@Cu composite anode, but it does not demonstrate the direct correlation with the Li deposition/stripping process. The supplementary experiments have been comprehensively conducted to address the above issues. First of all, as shown in Fig. R14a-c, numerous nanorod-like and nanodot-like structures are distributed randomly onto the flat Li/LiZn@Cu substrate. According to the EDS

mappings (Fig. R14d-e), such nanostructures consist of Zn and Cu elements, which can be ascribed to the formation of LiZn and/or Li_2ZnCu_3 alloys upon doctor-blading the mixtures at a temperature of 200 °C. Although the existence of the Li_2ZnCu_3 alloy on the Li/LiZn@Cu surface, its content is much less than that of the LiZn. According to Fig. R14f and the chemical formula of Li_2ZnCu_3 , the content of Li_2ZnCu_3 alloy in these nanostructures is only approx. 4.1%, implying that the Li_2ZnCu_3 alloy is not the main component in the electrochemically active layer of the Li/LiZn@Cu anode (i.e. Li/LiZn layer).

Fig. R14 (Fig. 1h-j in the revised manuscript and Supplementary Table 2 in SI). SEM top-view images, (d-e) the corresponding EDS mappings, and (f) compositional results of the Li/LiZn@Cu.

In addition to the surface of the Li/LiZn@Cu anode, as shown in Fig. R15a, a visible Li_2ZnCu_3 layer exists at the contact of the Li/LiZn layer and Cu substrate, which is verified by the EDS mapping images (Fig. R15b-d). Besides, in the region of the LiZn layer, excepting for the obvious overlap of Zn element distribution with the internal nanostructures, the content of Cu element is very low, further indicating that the Li_2ZnCu_3 alloy is not the majority in Li/LiZn active layer. Such results collectively mean the fact that the Li_2ZnCu_3 alloy mainly exists between the Cu and Li/LiZn active layer but not in the Li/LiZn active layer, which can be attributed to the direct contact of the molten Li-Zn mixture and Cu surface during the doctor-blading process.

Fig. R15 (Supplementary Fig. 11 in SI). (a) SEM cross-section image and (b-d) the corresponding EDS mappings of the interface between the Li/LiZn layer and Cu substrate.

To further verify this point of view, we characterize the Li_2ZnCu_3 layer. Through alcohol etching, the Li/LiZn active layer can be removed and the Li_2ZnCu_3 layer is maintained (named as e-Li/LiZn@Cu). The color of the Li_2ZnCu_3 layer is light grey (Fig. R16a-c). The morphology and crystal structures of the e-Li/LiZn@Cu are also characterized by SEM and XRD. As shown in Fig. R16d, the surface of the e-Li/LiZn@Cu presents coarse morphology with numerous pyramid-shaped structures, which can effectively improve the molten Li wettability. The uniform distribution of Zn and Cu elements with a close atomic ratio of 1:3 between Zn and Cu evidences the surface composition of e-Li/LiZn@Cu is Li_2ZnCu_3 alloy (Fig. R16e-g). The SEM cross-section images and corresponding EDS mappings indicate that the thickness of Li_2ZnCu_3 is $\sim 2 \mu\text{m}$ (Fig. R16h-k). Then, we compare the XRD patterns of Li/LiZn@Cu (Fig. 11 of the revised manuscript) and e-Li/LiZn@Cu (Fig. R16l). After the alcohol immersion, the peak intensities of Li_2ZnCu_3 remain strong enough. Therefore, the above results suggest that the initial pronounced XRD peaks of Li_2ZnCu_3 mainly stem from the Li_2ZnCu_3 layer on the surface of Cu substrate. Namely, the Li_2ZnCu_3 alloy distributes within the Li/LiZn@Cu composite anode, but not in the Li/LiZn active layer. As a matter of fact, the Li_2CuZn_3 acts as a good conductor to transport electrons and also contributes to the homogeneous deposition of Li, which are the main roles during electrochemical cycles, not participating in the Li alloy/dealloy process.

Fig. R16. Optic images of the (a) Cu foil, (b) Li/LiZn@Cu, and (c) Li/LiZn@Cu after alcohol immersion (e-Li/LiZn@Cu). (d) SEM top-view image, (e-f) the corresponding EDS mappings, and (g) compositional results of e-Li/LiZn@Cu. (h-i) SEM cross-section images and (j-k) the corresponding EDS mappings of e-Li/LiZn@Cu. (l) XRD pattern of the e-Li/LiZn@Cu.

At the beginning, as shown by the in-situ XRD characterization in Fig. 2j (initial manuscript), we did use the electrode formed by pressing mixtures, rather than the blade mixing onto the Cu foil. The main reason is that the Li_2ZnCu_3 alloy possesses excellent electrochemical stability, which will not form a new phase during the charging or discharging process. To verify this viewpoint, we specially assembled e-Li/LiZn@Cu||Cu (i.e. working electrode: e-Li/LiZn@Cu; counter electrode: Cu) and Cu||Cu cells, following a charging process at a constant charging current density for Li^+ ion extraction from Li_2ZnCu_3 alloy as much as possible ($25 \mu\text{A}$, a much lower value than the applied current density in symmetric and full cell tests in initial manuscript). As shown in Fig. R17a-b, the voltage profiles of e-Li/LiZn@Cu||Cu and Cu||Cu cells are similar, both rapidly rising to 1.5 V in a short time, which indicates that the Li^+ ion can not be dealloyed from Li_2ZnCu_3 alloy. Interestingly, such voltage profiles

are also similar to the charging of a capacitor. This capacitive ion storage behavior is related to the electric double-layer capacitance on the electrode surface, and its current response is basically independent of the battery, which does not involve the phase transition caused by ion extraction in the bulk phase (*Energy Environ. Sci.* 2011, 4, 2614-2624). Moreover, the half cells with $\text{Li}||\text{e-Li/LiZn@Cu}$ and $\text{Li}||\text{Cu}$ configurations were further assembled to verify the stability of Li_2ZnCu_3 alloy during the discharging process. The $\text{Li}||\text{e-Li/LiZn@Cu}$ cell also shows a similar voltage profile to the $\text{Li}||\text{Cu}$ cell, in which the platform of voltage drop can be ascribed to the decomposition of electrolyte and formation of SEI (Fig. R17d-e). Furthermore, as shown in Fig. R17c, f, all disassembled electrodes do not significantly change in XRD patterns compared to the pristine e-Li/LiZn@Cu (Fig. R16l).

Fig. R17 (Supplementary Fig. 23 in SI). Voltage profiles of (a) $\text{e-Li/LiZn@Cu}||\text{Cu}$ and (b) $\text{Cu}||\text{Cu}$ cells after charging to 1.5 V at $25 \mu\text{A}$. (c) XRD patterns of the disassembled e-Li/LiZn@Cu electrodes after stripping to 1.5 V. Voltage profiles of (d) $\text{e-Li/LiZn@Cu}||\text{Cu}$ and (e) $\text{Cu}||\text{Cu}$ cells after discharging to 0 V at $10 \mu\text{A cm}^{-2}$. (f) XRD patterns of the disassembled e-Li/LiZn@Cu electrodes after discharging to 0 V.

To better validate the above results, we further assembled the in-situ $\text{Li}||\text{e-Li/LiZn@Cu}$ cell to study the electrochemical stability of Li_2ZnCu_3 alloy in depth. A thinner Cu foil ($6 \mu\text{m}$) was used as the substrate for better X-ray diffraction observation. The whole testing process can be divided into two parts, one of which is the Li discharging and charging with a voltage range from 0 to 1 V at $50 \mu\text{A cm}^{-2}$ and the other of which involves the Li plating with a fixed Li amount of 3mA cm^{-2} and stripping to 1.5 V at 1mA cm^{-2} . As shown in Fig. R18a-b, the peak intensities and positions assigned to the Li_2ZnCu_3 alloy are highly consistent without peak weakening or shifting during the Li discharging and

charging process, indicating that the Li_2ZnCu_3 is electrochemical inactive within the above-operated voltage. Moreover, the Li_2ZnCu_3 also maintains a stable phase throughout the Li plating and stripping process, even when the cut-off voltage is up to 1.5 V (Fig. R18c-d). According to the above comprehensive XRD and electrochemical analysis, we can conclude that the Li_2ZnCu_3 is an inert alloy in electrochemical tests, just like Cu substrate, so that will not participate in electrochemical reactions like LiZn alloy. In addition, a shallow peak belonging to Li (110) gradually emerges during Li plating and fades after Li stripping (Fig. R18e). Such a low signal of Li can be attributed to the low content on Cu substrate. Thus, due to the excellent electrochemical stability of the Li_2ZnCu_3 alloy and to reduce the interference of the Cu substrate on LiZn and Li characteristic peaks to better clarify the phase changes of the Li/LiZn layer during the Li stripping/plating process, we ignored the Li_2ZnCu_3 alloy and Cu substrate in our in-situ XRD characterization as shown in Fig. 2j of the initial manuscript.

Fig. R18 (Supplementary Fig. 24 in SI). (a) *In-situ* XRD analysis of e-Li/LiZn@Cu electrode and (b) the corresponding contour maps during Li discharging and charging with a voltage range from 0 to 1 V at 50 $\mu\text{A cm}^{-2}$. (c) *In-situ* XRD analysis of e-Li/LiZn@Cu electrode, (d) the corresponding contour maps, and (e) selected region of Li peaks during Li plating with a fixed Li amount of 3 mA cm^{-2} and stripping to 1.5 V at 1 mA cm^{-2} .

However, given that there is indeed a small amount of Li_2ZnCu_3 alloy present in the Li/LiZn layer, we still used theoretical calculation to discuss Li_2ZnCu_3 's role in the Li deposition behaviors. Due to the limited literature on Li_2ZnCu_3 alloy, the surface energies of different crystal planes of Li_2ZnCu_3 were studied to construct the most stable surface. As shown in Table R3, the Li_2ZnCu_3 (001) possesses the lowest surface energy so it is selected for Li adsorption and diffusion calculations.

Table R3 (Supplementary Table 5 in SI). Calculated surface energies of Li_2ZnCu_3 .

Composition	Surface miller indices	Surface energy (J m^{-2})
Li_2ZnCu_3	(001)	0.694
	(100)	0.852
	(110)	0.924
	(111)	0.952

As shown in Fig. R19a-d, the Li absorption energy of Li_2ZnCu_3 (001) surface is -2.78 eV with two possible adsorption configurations towards Li, which is even lower than the LiZn (110) (-2.07 eV), let alone being compared to the Li (100). It suggests that the Li_2ZnCu_3 alloy possesses a stronger Li^+ ion affinity, which can effectively promote Li nucleation and subsequently suppress the uneven Li deposition. However, as shown in Fig. R19e, the Li_2ZnCu_3 alloy does not have significant advantages in promoting Li surface diffusion compared to LiZn alloy (0.03 eV) and bare Li (0.06 eV), due to the same value (0.06 eV) of surface diffusion barrier along the Li_2ZnCu_3 (001) surface with Li (100). Consequently, compared with the bare Li, the Li_2ZnCu_3 alloy can improve the uniform Li deposition to some extent due to its superior lithiophilicity, which can be further proved by the deposition morphology comparisons (plating 3 mAh cm^{-2} Li) of Li@Cu (prepared by electrochemical plating), M-Li@Cu (prepared by mechanical rolling), and Li@e-Li/LiZn@Cu (Li deposited e-Li/LiZn@Cu electrode) in Fig. R20.

Fig. R19 (Supplementary Fig. 34 in SI). (a) Li adsorption energies on the Li_2ZnCu_3 (001) surface and (b) corresponding adsorption sites. (c) Another configuration of Li adsorption energies on the Li_2ZnCu_3 (001) surface and (d) corresponding adsorption sites. (e) The diffusion barriers of Li along the Li_2ZnCu_3 (001) surface.

Fig. R20 (Fig. R20a-b is the Supplementary Fig. 22a-b in SI, Fig. R34f-g is the Supplementary Fig. 22a-b in SI, and Fig. R20a-b is the Supplementary Fig. 22c-d in SI). SEM images of (a-b) Li@Cu , (c-d) M-Li@Cu , and (e-f) Li@e-Li/LiZn@Cu anodes after Li deposition at 3 mAh cm^{-2} .

To sum up, we have modified Fig. 1 as shown below and supplemented the correlated Figure, Table, and discussion in the revised manuscript and SI.

Page 6-7 of the revised manuscript:

“Here, the atom concentration of Li in Li/LiZn is $\sim 97.3\%$. Thus, the nanorod-like and nanodot-like nanostructures mainly correspond to the LiZn alloy skeletons as shown in Fig. 1j, which is verified by the energy dispersive X-ray spectroscopy (EDS) measurement of the Li/LiZn@Cu surface, presenting the concentrative distribution of the Zn element around the nanostructures. As shown in Supplementary Fig. 11, such nanostructures are buried by the metallic Li, forming a 3D composite Li anode supported by alloy skeletons. Interestingly, the Cu element also exists on the nanostructures, where the atom concentration is 10.91% as listed in Supplementary Table 2. As shown in Fig. 1k and Supplementary Figs. 12-14, the simulated ternary phase diagram and chemical potentials of the Li-Zn-Cu system forecast that the Li-Zn alloy, Li, and Cu substrate would re-alloy and convert into the rarely reported Li_2ZnCu_3 alloy, which is an energy-favorable phase at a wide temperature range. Thus, these nanostructures contain LiZn and a small amount of Li_2ZnCu_3 ($\sim 4.1\%$). As shown in Fig. 11 and S15-16, the X-ray diffraction (XRD) peaks indicate the coexistence of LiZn alloy (PDF#03-0954), metallic Li (PDF#15-0401), Cu (PDF#04-0836) substrate and the newly formed Li_2ZnCu_3 alloy in the Li/LiZn@Cu anode. Excepting these peaks, no obvious metallic Zn peaks are detected to indicate the thorough alloying process between Li and Zn. Of special importance is that the Li_2ZnCu_3 alloy mainly exists at the interface between the Li/LiZn layer and Cu substrate as shown in Supplementary Fig. 17.”

Page 13-14 of the revised manuscript:

“Additionally, given that the presence of a small amount of Li_2ZnCu_3 alloy in the Li/LiZn layer and its superior lithiophilicity, it will also exert positive influences on Li plating behaviors as shown in Supplementary Table 5 and Fig. 21. As shown in Fig. 3d-i, k, this is further attested by the SEM images of Li/LiZn@Cu after quantitative Li plating, where at a Li deposition of 0.5 mAh cm^{-2} , the homogeneous Li nuclei appear on the LiZn alloy surface and further at 1 and 3 mAh cm^{-2} , a dense and smooth Li layer is planted on the Li/LiZn@Cu anode without dendrite growth, which is in sharp contrast to Li@Cu anode as shown in Supplementary Fig. 22a-b.

Furthermore, the *in-situ* XRD characterization furnishes an in-depth understanding of the Li

stripping/plating behaviors in the Li/LiZn@Cu anode. Due to the excellent electrochemical stability of Li_2ZnCu_3 alloy and to prevent interference of Cu substrate on XRD signals as shown in Supplementary Figs. 23-24, the working electrode is set using the Li/LiZn anode for simplification.”

Fig. 1. Fabrication of Li/LiZn@Cu anode. (a) Schematic illustration of the fabrication process. The Li wettability for (b) molten Li and (c) molten Li-Zn mixture on Cu foils. (d) Thickness, (e-f) cross-section SEM images, (g) size, (h-i) SEM top-view images, (j) corresponding EDS mappings of the ultrathin Li/LiZn@Cu anode. (k) The calculated ternary phase diagram of the Li-Zn-Cu system at 473 K. (l) XRD pattern of the ultrathin Li/LiZn@Cu anode.

Page 23-24 of the revised manuscript:

“In addition, the e-Li/LiZn@Cu electrode can be obtained after etching off the Li/LiZn layer with

alcohol.”

“In the *in-situ* Li||e-Li/LiZn@Cu cell, the cathode was e-Li/LiZn@Cu (Cu thickness: 6 μm) and the anode was Li foil.”

Page 27 of the revised manuscript:

“The surface energy σ of Li_2ZnCu_3 can be obtained as follows:

$$\sigma = \frac{1}{2A} [E_{\text{slab}} - nE_{\text{bulk}}] \quad (6)$$

where the E_{slab} is the energy of relaxed slab models with no restriction of the atomic position, the E_{bulk} is the total energy of the bulk structure, and the A is the area surface.”

Page 21 of the revised SI:

“The e-Li/LiZn@Cu electrode was obtained by immersing and rinsing with the alcohol solution. After the drying process, as shown in Supplementary Fig. 17a-c, differing from the red-orange color of the pristine Cu or silver color of the Li/LiZn@Cu, the sample after the detachment of Li/LiZn from Cu foil showed a different surface color. Due to the direct contact between the Cu surface and molten Li-Zn mixture, the re-alloying reaction and Li_2ZnCu_3 alloy mainly existed on the Cu surface. Moreover, the top-view SEM image of the e-Li/LiZn@Cu also displayed a pyramid-shaped morphology, which was completely different from Cu foil (Supplementary Fig. 17d), indicating the reconstruction of the smooth Cu surface by re-alloying reaction. The uniform distribution of Zn and Cu elements with a close atomic ratio of 1:3 between Zn and Cu evidenced that the surface composition of e-Li/LiZn@Cu is Li_2ZnCu_3 alloy (Supplementary Fig. 17e-g). The cross-section SEM and corresponding EDS mapping images indicated that the thickness of Li_2ZnCu_3 is $\sim 2 \mu\text{m}$ (Supplementary Fig. 17h-k). The pronounced Li_2ZnCu_3 peaks of the Rietveld refinement of the XRD pattern of the e-Li/LiZn@Cu sample further verified the preservation of the Li_2ZnCu_3 layer on the Cu surface (Supplementary Fig. 17l and Table 3). Based on the above characterizations and analysis, the Li_2ZnCu_3 alloy mainly present at the interface between the Li/LiZn layer and Cu substrate, not inside the Li/LiZn layer.”

Supplementary Fig. 17. Optic images of the (a) Cu foil, (b) Li/LiZn@Cu, and (c) Li/LiZn@Cu after alcohol immersion (e-Li/LiZn@Cu). (d) Top-view SEM images, (e-f) the corresponding EDS mapping, and (g) results of e-Li/LiZn@Cu. (h-i) Cross-section SEM images and (j-k) the corresponding EDS mapping of e-Li/LiZn@Cu. (l) Rietveld refinement of XRD pattern of the e-Li/LiZn@Cu sample.

Supplementary Table 3. Rietveld refinement results of XRD data for the alcohol-treated Li/LiZn@Cu sample.

Overall composition: Li ₂ ZnCu ₃ (conventional cell). Space group: R $\bar{3}m$, No. 166					
Lattice constants: a = b = 4.974 Å, c = 12.212 Å, V = 261.634 Å ³ , $\alpha = 90^\circ$, $\beta = 90^\circ$, $\gamma = 120^\circ$					
atom	site	x	y	z	Frac.
Li	6c	0	0	-0.32994	1
Zn	3b	0	0	-0.5	1
Cu	9e	-0.16667	0.16667	-0.33333	1
Overall composition: Cu. Space group: Fm $\bar{3}m$, No. 225					
Lattice constants: a = b = c = 3.615 Å, V = 47.259 Å ³					
atom	site	x	y	z	Frac.

Page 28-31 of the revised SI:

“Compared with the Li and LiZn, few literatures have focused upon the Li_2ZnCu_3 . The Li_2ZnCu_3 (001) is selected as the stable surface due to its lowest surface energy (Supplementary Table 5). As shown in Supplementary Fig. 21a-d, the Li absorption energy of Li_2ZnCu_3 (001) surface is -2.78 eV (the Li_2ZnCu_3 (001) surface possesses two possible adsorption structures towards Li), which is even lower than that of the LiZn (110) (-2.07 eV), not to mention the Li (100) surface. It suggests that the Li_2ZnCu_3 alloy possesses a stronger Li^+ ion affinity, which can effectively promote the Li nucleation and subsequently suppress the uneven Li deposition. However, as shown in Supplementary Fig. 21e, the Li_2ZnCu_3 alloy does not have significant advantages in promoting the Li surface diffusion compared to the LiZn alloy (0.03 eV) and bare Li (0.06 eV), due to the same value (0.06 eV) of diffusion barrier alongside the Li_2ZnCu_3 (001) and Li (100) surfaces. Consequently, compared with the bare Li, the Li_2ZnCu_3 alloy can improve the uniform Li deposition to some extent due to its superior lithiophilicity, which can be further proved by the deposition morphology comparisons (plating 3 mAh cm^{-2} Li) of Li@Cu (prepared by electrochemical plating, Supplementary Fig. 22a-b) and Li@e-Li/LiZn@Cu (Li deposited e-Li/LiZn@Cu electrode, Supplementary Fig. 22c-d).”

“The phase stability of Li_2ZnCu_3 alloy during the electrochemical process is explored based on the e-Li/LiZn@Cu electrode. Firstly, the e-Li/LiZn@Cu||Cu (i.e. working electrode: e-Li/LiZn@Cu; counter electrode: Cu) and Cu||Cu cells were assembled to verify the stability during the Li charging process. A low constant charging current of 25 μA is applied for Li^+ ion extraction from the Li_2ZnCu_3 alloy as much as possible. As shown in Supplementary Fig. 23a-b, the voltage profiles of e-Li/LiZn@Cu||Cu and Cu||Cu cells are similar, both rapidly rising to 1.5 V in a short time, which indicates that Li^+ ions can not be dealloyed from the Li_2ZnCu_3 alloy. Interestingly, such voltage profiles demonstrate the capacitive behavior. This capacitive ion storage behavior is a non-Faradic process, and its current response is basically independent of the battery.² Moreover, the half cells with Li||e-Li/LiZn@Cu and Li||Cu configurations were further assembled to verify the stability of Li_2ZnCu_3 alloy during discharging process. The Li||e-Li/LiZn@Cu cell also shows a similar voltage profile to

the Li||Cu cell, in which the platform of voltage drop can be ascribed to the decomposition of electrolyte and formation of SEI (Supplementary Fig. 23d-e). Furthermore, as shown in Supplementary Fig. 23c,f, all disassembled electrodes do not significantly change in XRD patterns compared to the pristine e-Li/LiZn@Cu (Supplementary Fig. 171).”

“The *in-situ* Li||e-Li/LiZn@Cu cell is further assembled to study the electrochemical stability of Li_2ZnCu_3 alloy in depth. A thinner Cu foil (6 μm) was used as the substrate for better X-ray diffraction observation. The whole testing process can be divided into two parts, one of which is the Li discharging and charging with a voltage range from 0 to 1 V at $50 \mu\text{A cm}^{-2}$ and the other of which involves the Li plating with a fixed Li amount of 3 mA cm^{-2} and stripping to 1.5 V at 1 mA cm^{-2} . As shown in Supplementary Fig. 24a-b, the peak intensities and positions belonging to Li_2ZnCu_3 alloy are highly consistent without peak weakening or shifting during the Li discharging and charging process, indicating that Li_2ZnCu_3 is electrochemical inactive within the above-operated voltage. Moreover, the Li_2ZnCu_3 also maintains a stable phase throughout the Li plating and stripping process, even when the cut-off voltage is up to 1.5 V (Supplementary Fig. 24c-d). According to the above comprehensive XRD and electrochemical analysis, it can be concluded that the Li_2ZnCu_3 alloy is an inert alloy during the electrochemical tests, just like Cu substrate, so that will not participate in electrochemical cycling. In addition, a shallow peak assigned to the Li (110) plane gradually emerges during Li plating and fades after Li stripping (Supplementary Fig. 24e). Such a low signal of Li can be attributed to the low content on Cu substrate. Due to the excellent electrochemical stability of the Li_2ZnCu_3 alloy and to reduce the interference of the Cu substrate on LiZn and Li characteristic peaks to better clarify the phase changes of the Li/LiZn layer during the Li stripping/plating process, the working electrode of *in-situ* XRD characterization in Fig. 3j is set using the Li/LiZn anode for simplification.”

III. Other concerns

1 The choice of control group

Upon scrutinizing the methodology section, it becomes evident that the designated control group, denoted as Li@Cu, is fashioned by electrochemically plating a specific quantity of lithium onto the copper foil. While this serves as a comparative baseline for the experimental results, a substantial

performance variance between the experimental and control groups is apparent. However, the rationale underpinning this selection of the control group merits reevaluation.

The electrochemical plating process employed to prepare Li@Cu engenders morphological and physicochemical disparities that set it apart from the mechanically-rolled Li-Cu foil. Indeed, this preparation methodology aligns with what can be termed an "anode-free" scenario. In the context of prior investigations involving anode-free cells, it is well-established that electrochemically deposited lithium assumes a rugged morphology. Moreover, the generation of dead lithium within this context is not uncommon, a factor contributing to the degradation of cell performance.

As a constructive suggestion, it would be judicious to prepare or procure the Li-Zn@Cu composite anode and Li@Cu samples devoid of exposure to the electrochemical plating process. Additionally, considering the controllable nature of Li-Zn@Cu's areal capacity, a prudent approach could involve preparing a composite anode with a relatively substantial areal capacity. This strategic maneuver would facilitate seamless comparisons with commercial Li-Cu foil, which boasts a thickness of 50 μm .

Reply:

Thanks for your suggestions.

We prepared the Li@Cu electrode by mechanical rolling method (named M-Li@Cu and purchased from China Energy Lithium Co., Ltd) without exposure to the electrochemical plating process. As shown in Fig. R21a-b, the M-Li@Cu possesses visible mechanical indentations on the surface. The XRD pattern exhibits that the M-Li@Cu only contains the metallic Cu and Li (Fig. R21c). The thickness of M-Li@Cu is $\sim 34 \mu\text{m}$ which is similar to the Li/LiZn@Cu electrode ($35\mu\text{m}$), in which the bottom region is $10 \mu\text{m}$ -thick Cu substrate and the top region is $24 \mu\text{m}$ -thick metallic Li layer (Fig. R21d-e). Compared with the Li/LiZn layer, the bare Li layer can not realize the ordered Li grow regulation due to its poor Li adsorption ability and large diffusion barrier for Li lateral deposition, as verified in Fig.3a-c of the revised manuscript. As shown in Fig. R21f-g, the surface of the M-Li@Cu electrode presents a Li intertwined morphology with many pores after plating 3mAh cm^{-2} Li. Such an uneven Li morphology with a high exposure area to electrolyte will bring about excess SEI formation and active material consumption, accompanied by the aggravated generation of the "dead Li" in subsequent cycles. Consequently, after 50 cycles at $1 \text{mA cm}^{-2}/1 \text{mAh cm}^{-2}$ in the symmetric cell with M-Li@Cu||M-Li@Cu configuration, the worse Li morphology can be observed with the partial Li

layer detaches from the substrate (Fig. R21h-k). This explains the poor long-term cycling performance of the M-Li@Cu||M-Li@Cu cells, in which the voltage violently fluctuates after only 250 h, whereas the symmetric Li/LiZn@Cu cell delivers a stable voltage platform for 690 h (Fig. R21l). It is worth mentioning that the cycling performance of M-Li@Cu is much better than that of Li@Cu which is prepared by electrochemical plating. As the reviewer pointed out, the Li@Cu prepared by electrochemical plating will inevitably introduce a rugged morphology and “dead Li”, thereby resulting in an inferior Li/electrolyte interface and electrochemical performance than M-Li@Cu in long-term cycles. The EIS analysis also demonstrates this viewpoint (Fig. R21n-p). Although both M-Li@Cu and Li@Cu exhibit an increase in R_{SEI} (corresponding to the semicircle at high frequency) after 50 cycles, the M-Li@Cu anode still has a smaller value than Li@Cu one, as well as its better rate performance in Fig. R21m. Furthermore, the M-Li@Cu-based full cell with a high-loading LFP cathode (2 mAh cm^{-2}) was assembled to conduct a systematic evaluation of M-Li@Cu for electrochemical stability. As shown in Fig. R21q, the M-Li@Cu||LFP cell delivers a higher capacity retention of $\sim 80.0\%$ after 65 cycles than Li@Cu||LFP (54.4% , 60 cycles), but is much inferior to Li/LiZn@Cu cell.

To sum up, benefiting from the more uniform morphology of initial Li and no obvious generation of “dead Li” in the preparation process, the M-Li@Cu exhibits a better electrochemical performance than that of the Li@Cu anode, no matter in symmetric or full cell configuration. However, the poor Li adsorption and surface diffusion ability of M-Li@Cu also restrict its cycling performance, showing a significant gap compared with Li/LiZn@Cu electrode.

Fig. R21. (a) SEM top-view images, (b) XRD pattern, and (c) SEM cross-section images of M-Li@Cu composite anode. Morphology evolution of M-Li@Cu after (f-g) plating 3 mAh cm⁻² Li and (h-i) 50 cycles in symmetric cell at 1 mA cm⁻²/1 mAh cm⁻² and (j-k) corresponding SEM cross-section SEM images. Comparisons of cycling performance of M-Li@Cu-, Li@Cu-, and Li/LiZn@Cu-based symmetric cells in (l) long-term cycling, (m) rate, and (n-p) EIS tests. (q) Comparisons of the cycling performance of M-Li@Cu-, Li@Cu-, and Li/LiZn@Cu-based full cells.

Then, considering the suggestions of the reviewer about “Additionally, considering the controllable nature of Li-Zn@Cu's areal capacity, a prudent approach could involve preparing a composite anode with a relatively substantial areal capacity. This strategic maneuver would facilitate seamless comparisons with commercial Li-Cu foil, which boasts a thickness of 50 μm”, we further prepared the Li-Zn@Cu electrode with thicker thickness and high capacity. As shown in Fig. R22a-d, the 50 μm-thick (Li/LiZn: 40 μm; Cu: 10 μm) and 58 μm-thick (Li/LiZn: 48 μm; Cu: 10 μm) Li-Zn@Cu electrodes were successfully fabricated. The areal capacity of 58 μm-thick Li-Zn@Cu electrode is ~ 8.7 mAh cm⁻² (Fig. R22e).

Fig. R22 (Supplementary Fig. 5 in SI). SEM cross-section images of Li/LiZn@Cu with (a-b) 40 μm -thick and (c-d) 48 μm -thick Li/LiZn layers. (e) Voltage profile of the stripping process for the Li/LiZn@Cu with 48 μm -thick Li/LiZn layer.

The related discussion and Figure are supplemented into the revised manuscript and SI as follows:

Page 6 of the revised manuscript

“What is more, the thickness of the cast Li/LiZn layer is adjustable (range from 5 to 48 μm , corresponding to the capacity from 0.89 to 8.7 mAh cm^{-2}), the delivered capacities of which are compatible with most of the available cathodes as shown in Supplementary Figs. 3-5.”

Page 20 of the revised manuscript

“Additionally, considering that the inevitable introduction of the rugged Li morphology and “dead Li” in Li@Cu anode during electrochemical preparation, which will exert innate disadvantages in full cell tests, the mechanical rolling-based ultrathin Li@Cu (M-Li@Cu, $\sim 24 \mu\text{m}$ -thick Li) anode is further set to be as control group as shown in Supplementary Fig. 34. When paired with high-loading LFP (13.1 mg cm^{-2} , corresponding to $\sim 2 \text{mAh cm}^{-2}$, N/P = 2.2), the prominent performance of Li/LiZn@Cu||LFP full cell is also inherited, where it displays enhanced capacity retention of 90.1% and a prolonged lifespan with respect to those of the Li@Cu||LFP and M-Li@Cu||LFP cells as shown in Supplementary Fig. 34o.”

Page 24 of the revised manuscript

“The ultrathin M-Li@Cu anode (thickness: ~24 μm -thick Li+10 μm -thick Cu foil) was purchased from China Energy Lithium Co., Ltd.”

Page 42-44 of the revised SI

“The electrodeposition-based Li@Cu will be exposed to the electrolyte during fabrication, inevitably introducing unnecessary by-products into Li anode and making the pre-loss of active Li. To exclude the introduction of rugged Li and “dead Li” in preparation, the mechanical rolling-based ultrathin Li@Cu anode (named the M-Li@Cu and purchased from China Energy Lithium Co., Ltd) is also considered as the control group. As shown in Supplementary Fig. 34a-b, the M-Li@Cu possesses visible mechanical indentations on the surface. The XRD pattern exhibits that the M-Li@Cu only contains the metallic Cu and Li (Supplementary Fig. 34c). The thickness of M-Li@Cu is ~ 34 μm which is similar to the Li/LiZn@Cu electrode (35 μm), in which the bottom region is 10 μm -thick Cu substrate and the top region is 24 μm -thick metallic Li layer (Supplementary Fig. 34d-e). Compared with the Li/LiZn layer, the bare Li layer can not realize the ordered Li growth regulation due to its poor Li adsorption ability and large diffusion barrier for the Li lateral deposition, as verified in Fig. 3a-c. As shown in Supplementary Fig. 34f-g, the surface of the M-Li@Cu electrode presents a Li intertwined morphology with many pores after plating 3 mAh cm^{-2} Li. Such an uneven Li morphology with a high exposure area to electrolyte will bring about excessive SEI formation and active material consumption, accompanied by the aggravated generation of the “dead Li” in subsequent cycles. Consequently, after 50 cycles at 1 mA cm^{-2} /1 mAh cm^{-2} in the symmetric cell with M-Li@Cu||M-Li@Cu configuration, the worse Li morphology can be observed with the partial Li layer detaches from the substrate (Supplementary Fig. 34h-k). This explains the poor long-term cycling performance of the M-Li@Cu||M-Li@Cu cells in Fig. 34l, in which the voltage violently fluctuates after only 250 h, whereas the symmetric Li/LiZn@Cu cell delivers a stable voltage platform for 690 h. It is worth mentioning that the cycling performance of M-Li@Cu anode is much better than that of Li@Cu one which is prepared by electrochemical plating. It can be ascribed to the more dense and uniform Li layer on the pristine M-Li@Cu than that of Li@Cu, thereby resulting in a superior Li/electrolyte interface and electrochemical performance than that of the Li@Cu anode in long-term cycles. The EIS analysis also demonstrates this viewpoint. As shown in Supplementary Fig. 34n, although both M-Li@Cu and

Li@Cu exhibit an increase in R_{SEI} (the semicircle at high frequency) after 50 cycles, the M-Li@Cu still has a smaller value than that of the Li@Cu (Supplementary Fig. 31a), as well as its better rate performance in Supplementary Fig. 34m. Furthermore, the M-Li@Cu-based full cell with a high-loading LFP cathode (2 mAh cm^{-2}) was assembled to conduct a systematic evaluation of M-Li@Cu for electrochemical stability. As shown in Supplementary Fig. 34o, the M-Li@Cu||LFP cell delivers a higher capacity retention of $\sim 80.0\%$ after 65 cycles than the 54.4% after 60 cycles of Li@Cu||LFP cell, but is much inferior to the Li/LiZn@Cu cell.

Benefiting from the uniform morphology of initial Li and no generation of “dead Li” in the preparation process, the M-Li@Cu exhibits a better electrochemical performance than that of the Li@Cu, no matter in symmetric or full cell configuration. However, the poor Li adsorption and surface diffusion ability of M-Li@Cu also restrict its cycling performance, showing a significant gap compared with the Li/LiZn@Cu electrode.”

Supplementary Fig. 34. (a) SEM top-view images, (b) XRD pattern, and (c) SEM cross-section images of M-Li@Cu. Morphology evolutions of M-Li@Cu after (f-g) plating 3 mAh cm^{-2} Li and (h-i) 50 cycles in symmetric cell at $1 \text{ mA cm}^{-2}/1 \text{ mAh cm}^{-2}$ and (j-k) corresponding SEM cross-section images. Comparisons of the M-Li@Cu-, Li@Cu-, and Li/LiZn@Cu-based symmetric cells in (l) long-term cycling, (m) rate, and (n) EIS tests. (o) Comparisons of the cycling performance of M-Li@Cu-, Li@Cu-, and Li/LiZn@Cu-based full cells with a high-loading LFP cathode ($\sim 2 \text{ mAh cm}^{-2}$).

2 The performance evaluation by comparing with other reports

This manuscript predominantly focuses on juxtaposing the optimal performance outcomes of the composite anodes against both LIBs and their designated control group. However, in light of the myriad reports concerning scalable composite anodes featuring noteworthy practical areal capacity and elevated capacities, it is essential for the performance metrics outlined in this study to be critically contextualized within the broader landscape of relevant literature. The scope of this comparative analysis should ideally extend beyond the LIBs and control group in order to provide a comprehensive assessment against existing research endeavors. For instance,

- a. “Roll-To-Roll Fabrication of Zero-Volume-Expansion Lithium-Composite Anodes to Realize High-Energy-Density Flexible and Stable Lithium-Metal Batteries” (Adv.Mater. 2022, 34, 2205677)
- b. “A 10- μm Ultrathin Lithium Metal Composite Anodes with Superior Electrochemical Kinetics and Cycling Stability” (Energy Environ. Mater. 2023, 0, e12598)
- c. “Free-standing ultrathin lithium metal–graphene oxide host foils with controllable thickness for lithium batteries” (Nat. Energy 6, 790–798 (2021))

Reply:

Thanks for your suggestion.

In reference a (Adv. Mater. 2022, 34, 2205677), Luo et al. reported a sandwich-like Li anode (zeroVE-Li), which is from top to bottom consisting of EI film, 3D CuCM framework, and LiMg alloy, respectively. Such a unique composite structure provides not only improved cycling stability over 200 cycles with 63% capacity retention in high-loading (3.7 mAh cm^{-2}) NCM811-based full cell, but also excellent battery performance during 3000 bending cycles in flexible pouch cell. However, compared with our design, this zeroVE-Li anode has a complex fabrication process, as well as poor thickness adjustability. The thick zeroVE-Li ($\sim 104 \mu\text{m}$) leads to the excessive introduction of inactive materials, resulting in low energy density of the assembled battery (214 Wh kg^{-1}) compared to our battery (284.0 Wh kg^{-1} or 366.5 Wh kg^{-1} when containing 1 or 10 stacked layers, respectively).

In reference b (Energy Environ. Mater. 2023, 0, e12598) and c (Nat. Energy 6, 790–798 (2021)), although the reduced thickness of their Li anode (reference b: $10 \mu\text{m}$; reference c: $0.5\text{--}20 \mu\text{m}$), they utilized expensive materials (carbon nanotube and reduced graphene oxide) with complex synthesis process to serve as 3D inactive frameworks and accommodate Li infusion. In sharp contrast, the

preparation of ultrathin Li/LiZn@Cu anode only involves common and low-cost Li and Zn foils without complex preparation conditions, which is suitable for large-scale fabrication.

We have discussed and cited these references appropriately, and supplemented them in the Introduction section. In addition, we have also added the data of the symmetric batteries including cycled capacity/applied current density, depth of discharge (the ratio between the cycled capacity and total capacity), and lifespan and cycling voltage range of the full cells based on the reported anodes in Supplementary Table 6.

The related discussion and references are added on Pages 2-3 and 15 of the revised manuscript and Supplementary Table 6 is listed in the revised SI as follows:

Page 2 of the revised manuscript

“Despite the better cycling performance they have achieved, the overstock LMAs can not reasonably match the capacity of the current commercial cathodes (e.g. 3 mAh cm⁻²) in a low N/P ratio (< 3), resulting in a waste of most Li sources with compromising the high energy density.¹⁹”

Page 3 of the revised manuscript

“At this point, the thickness of the Li composite anode is closely related to that of the 3D host. By controlling the thicknesses of the 3D hosts, the ultrathin carbon nanotube-based Li-MnO_x/CNT (10 μm) and reduced graphene oxide-based Li@eGF (0.5-20 μm) with the limited Li sources can be readily prepared.^{26,27} However, the preparation for the ultrathin 3D hosts is complex and high-cost. Another feasible and cost-effective way is directly reshaping the thickness of Li when it is in the liquid state.”

Page 15 of the revised manuscript

“The cycled capacity is fixed at 1 mAh cm⁻², corresponding to a discharge of depth of 22.7%, which is greater than that of the reported works as listed in Supplementary Table 6.”

Cited references

[19] Luo, C. et al. Roll-To-Roll Fabrication of Zero-Volume-Expansion Lithium-Composite Anodes to

Realize High-Energy-Density Flexible and Stable Lithium-Metal Batteries. *Adv. Mater.* 34, 2205677 (2022).

[26] Chen, H. et al. Free-standing ultrathin lithium metal–graphene oxide host foils with controllable thickness for lithium batteries. *Nat. Energy* 6, 790-798 (2021).

[27] Zhang, G. et al. A 10- μm Ultrathin Lithium Metal Composite Anodes with Superior Electrochemical Kinetics and Cycling Stability. *Energy Environ. Mater.* 6, e12598, (2023).

S/C composites	~1.27 mAh cm ⁻²	~44.4	0.5 C	200 cycles	1.7-2.8 V	72.4%	3
LFP	3 mAh cm ⁻²	3.3	0.5 C	130 cycles	2.8-3.8 V	~87%	4
LFP	1.9 mAh cm ⁻²	>16.8	0.5 C	200 cycles	2.4-3.8 V	93.6%	5
NCM811	~4 mAh cm ⁻²	5	1 C	200 cycles	3.0-4.3 V	87.66%	6
LFP	~0.34 mAh cm ⁻²	~58.8	0.2 C	200 cycles	2.5-4.0 V	94.5%	7
NCM111	~1.3 mAh cm ⁻²	~39.6	1 C	150 cycles	2.7-4.2 V	80.01%	8
LFP	~2 mAh cm ⁻²	~46.5	0.2 C	100 cycles	2.0-4.0 V	94%	9
NMC811	3.7 mAh cm ⁻²	3.6	0.54 C	200 cycles	/	63%	10
LCO	~0.4 mAh cm ⁻²	4	0.5 C	150 cycles	3.0-4.35 V	80.3%	11
LFP	3.24 mAh cm ⁻²	~1.14	0.5 C	200 cycles	3.0-4.0 V	81%	12
LFP	1.3 mAh cm ⁻²	~3.4	0.5 C	230 cycles	2.4-4.2 V	98.0%	Our work
LFP	2 mAh cm ⁻²	~2.2	0.5 C	130 cycles	2.4-4.2 V	90.1%	Our work
LCO	1.8 mAh cm ⁻²	~2.5	0.5 C	125 cycles	2.8-4.5 V	74%	Our work
LCO (pouch cell)	3.27 mAh cm ⁻²	~1.35	0.1 C	40 cycles	2.8-4.5 V	90% (compared with the highest capacity)	Our work

4 The electrolytes employed for in-situ XRD characterizations differ from those utilized for electrochemical performance evaluations.

Reply:

The electrolyte employed for in-situ XRD characterizations is an ether-based electrolyte consisting of 1 M lithium bis(trifluoromethanesulphonyl) imide (LiTFSI) in DOL/1,2-dimethoxyethane (DME) (v/v = 1:1) with 1 wt% LiNO₃, which is consistent with the electrolyte utilized for half and symmetric cells. Such an ether-based electrolyte is better compatible with metallic Li during Li stripping/plating for stable voltage profiles. While the electrolyte for full cells is a carbonate-based electrolyte with 1 M lithium hexafluorophosphate (LiPF₆) in ethylene carbonate (EC)/dimethyl carbonate (DMC)/ethyl methyl carbonate (EMC) (v/v/v = 1:1:1) with 5% fluoroethylene carbonate (FEC). Compared with ether-based electrolytes decomposing oxidatively beyond 4.0 V [Angew. Chem. Int. Ed. 2021, 60, 26837-26846], this carbonate-based electrolyte possesses excellent stability at high voltage, even stable at voltages as high as 4.5V, which is suitable for LFP-based (2.4 - 4.2 V) and LCO-based (2.8 - 4.5 V) cells to pursue higher energy density.

Switching the electrolytes for different battery systems (i.e. using ether-based electrolytes for half and symmetric cells and carbonate-based electrolytes for full cells) is a common strategy to preferably reinforce the battery performance, which has been applied in numerous reported works as listed in Table R4. Thus, we use different electrolyte systems for better performance considerations for different battery systems.

Table R4. Electrolyte systems for symmetric and full cells in reported Li anode.

Method	Electrolyte recipes		Ref
	Half cell/Symmetric cell	Full cell	
Li@ZDDP	1.0 M LiTFSI in DME:DOL (v/v =1:1) with 2.0% LiNO ₃	1.0 M LiPF ₆ in EC/EMC/FEC (v/v/v=3:7:1)	13
LaF ₃ -Cu	4 M LiFSI in DME	0.6 M LiDFOB and 0.6 M LiBF ₄ in FEC and DEC (v/v =1:2)	14
Self-assembled monolayers	1.0 M LiTFSI in DME:DOL (v/v = 1:1) with 1.0% LiNO ₃	1 M LiPF ₆ in EC:DEC/EMC (v/v/v=1:1:1) with 1 wt% FEC	15
rGO@Li	1.0 M LiTFSI in DME:DOL (v/v = 1:1) with 2.0% LiNO ₃	1 M LiPF ₆ in EC/DMC (v/v=1:1)	16
CaLi-Li	1.0 M LiTFSI in DME:DOL (v/v = 1:1) with 1.0% LiNO ₃	1 M LiPF ₆ in EC/DEC (v/v=1:1)	5

FGLi	1.0 M LiTFSI in DME:DOL (v/v = 1:1) with 1.0% LiNO ₃	1 M LiPF ₆ in EC:DMC/EMC (v/v/v=1:1:1) with 1 wt% FEC	17
Li-MnO _x /CNT	1.0 M LiTFSI in DME:DOL (v/v = 1:1) with 1.0% LiNO ₃	1 M LiPF ₆ in EC/DMC/EMC (v/v/v=1:1:1)	11

5 The LiCoO₂ full cell is charged to 4.5 V. Has the surface of LiCoO₂ been modified?

Reply:

Yes, in our manuscript, the surface of LiCoO₂ has been optimized to achieve stable operation under a high voltage of 4.5 V.

6 In order to enhance the overall quality of this article, a more comprehensive and detailed analysis and characterization are warranted. Given that the strategy of introducing heteroatoms to enhance wettability is extensively documented in existing literature, a deeper exploration is imperative.

Reply:

Thanks for your suggestion.

The wettability of molten Li towards the substrate is mainly determined by two factors, i.e. the surface tension of molten Li and the surface texture of substrate.

The surface tension of molten Li can be tuned by heteroatom doping and temperature. It has been widely accepted that heteroatom doping can effectively reduce the interatomic force of Li, which can be ascribed to the weaker bonding force between Li and heteroatoms than Li-Li (Nat. Commun. 2019, 10, 4930 and Adv. Sci. 2020, 7, 2002212). Moreover, the reaction between Li and hetero elements can release a negative Gibbs free energy (ΔG_r), which can further reduce the contact angle between molten Li and substrate as follows (J. Mater. Sci. 1991, 26, 3400-3408 and J. Mater. Sci. 2010, 45, 4256-4264):

$$\cos\theta^1 = \cos\theta^0 - \Delta\gamma_{sl}/\gamma_{gl} - \Delta G_r/\gamma_{gl} \quad (7)$$

in which the θ^1 and θ^0 denote the contact angle after or before the reaction, the $\Delta\gamma_{sl}$ is a change in solid-liquid interfacial energy by the reaction, and the γ_{gl} is the gas-liquid interfacial energy. The smaller the θ , the better the wettability. Thus, the exothermal reaction (negative ΔG_r) of Li and hetero elements could decrease the θ to enhance the Li wettability. As for temperature (T), as the T rises, the wettability of molten Li toward the substrate increases, which has been demonstrated by Cui and co-

workers (*Energy Storage Mater.* 2018, 14, 345-350). It may be attributed to that entropy increases with temperature, leading to a decrease in surface tension and viscosity of molten Li. The surface tension (γ_T) of Li at T can be expressed as follows (*Adv. Sci.* 2020, 7, 2002212):

$$\gamma_T = \frac{d\gamma}{dT}(T - T_m) + \gamma_m \quad (8)$$

where the $\frac{d\gamma}{dT} < 0$, the T_m is the melting point, and the γ_m is the surface tension at T_m . Namely, high temperature is helpful for the molten Li spreading along the substrate. However, it is not advisable to simply raise the temperature, as an increase in temperature represents an increase in operational risk. In addition, at high temperatures, the Li is more likely to react with the trace impurities in the glove box, generating passivation layers such as the thin Li_2O covers, which hinders the improvement of wettability (*Energy Storage Mater.* 2018, 14, 345-350).

The surface texture of the substrate also exerts a significant influence on the molten Li wettability, which is related to the surface composition and roughness. Similar to the improvement of wettability caused by the reaction between Li and doped elements, when the substrate surface can react with molten Li, it can also stimulate the release of negative ΔG_r and drive the spreading of molten Li. Besides, the surface roughness can influence the Laplace pressure (ΔP) to alter the molten Li wettability, as shown below (*Small.* 2023, 19, 2205653):

$$\Delta P = 2\sigma/r \quad (9)$$

where the σ is the surface tension of molten Li and the r refers to the capillary radius. A smaller r can induce a larger ΔP , as well as a higher liquid level, responding to the improved Li wettability. Moreover, as verified by Guo and co-workers, compared with ordinary vertical capillaries, the conical capillaries have better wettability enhancement (*Nano Energy.* 2020, 73, 104731). Because the Laplace pressure on the high curvature site of the conical capillary is greater than that in the low curvature site, the as-generated non-equilibrium Laplace pressure will better drive the molten Li spreading. Accordingly, increasing the roughness of the substrate surface can improve the molten Li wettability. However, it must be noted that such a situation only applies when the liquid and substrate are in a Wenzel state (Wenzel state means that there are no bubbles between the liquid and the substrate surface, which implies it is in a wetting state on the substrate surface, *Ind. Eng. Chem.* 1936, 28, 988-994). According to Young's model, the contact angle is defined as

$$\text{Cos}\theta = (\gamma_s - \gamma_{sl})/\gamma_l \quad (10)$$

in which the γ_s , γ_{sl} , and γ_l are the surface tensions of the substrate, solid-liquid surface, and liquid, respectively. In Wenzel state, the contact angle can be corrected to

$$\text{Cos}\theta_m = r(\gamma_s - \gamma_{sl})/\gamma_l = r\text{Cos}\theta \quad (11)$$

in which the θ_m is the measured contact angle and r is the roughness factor (r : the ratio between actual and projected surface areas of the substrate). As a result, when the substrate surface is already lithiophilicity (i.e. $\theta < 90^\circ$), increasing surface roughness will further enhance the wettability of molten Li. If the gas exists between the liquid and rough surface of the substrate, it is in the Cassie-Baxter state (Trans. Faraday Soc. 1944, 40, 546-551). At this point, an increase in roughness suppresses the wetting of the substrate by the liquid.

Recently, for the reinforcement of the molten Li wettability, introducing the elemental additives that can alloy with Li is a prevalent and effective strategy to tune Li wettability. However, almost all reported literature only focuses on the surface tension of molten Li, due to the weak interaction between the doped molten Li and substrate. In our manuscript, we selected Zn as the heteroatom doping to modulate the molten Li wettability towards the Cu substrate, which has a different wettability mechanism from the currently reported literature.

The molten Li-Zn mixture can optimize the surface tension of molten Li and the surface texture of the substrate simultaneously to achieve fast and uniform Li wetting on Cu foil at a low heating operating temperature. First of all, similar to the other Li-alloy formation, the LiZn alloy can also reduce the surface tension of molten Li due to the weaker interaction of Li-Zn with regard to the Li-Li bonds and negative ΔG_r ($\text{Li} + \text{Zn} \rightarrow \text{LiZn}$, $\Delta G_r = -23.6 \text{ kJ mol}^{-1}$). Moreover, different from the other heteroatom doping-induced Li wettability, the molten Li-Zn mixture can re-alloy with the Cu substrate and improve surface roughness for further wettability improvement. According to the ternary phase diagram of the Li-Zn-Cu system at 473 K (Fig. R23a), a new alloy phase of Li_2ZnCu_3 can be generated by the reaction among Li, LiZn, and Cu (Fig. 23b):

Such a negative ΔG_r provides an extra driving force for molten Li spreading. More significantly, the formation of the Li_2ZnCu_3 alloy can transform the morphology of Cu foil and significantly improve its roughness (Fig. R23c). The contact between the molten Li-Zn mixture and Cu substrate can be

regarded as being a Wenzel state due to the compact interface within the Li/LiZn layer and Cu in Li/LiZn@Cu. The pyramidal shape of Li_2ZnCu_3 alloy on the Cu surface also induces a large Laplace pressure to absorb molten Li and promote its diffusion. Under the above synergetic effects of the reduced interatomic force of the Li-Zn bond, the negative ΔG_r from the LiZn and Li_2ZnCu_3 formations, and the pyramidal-shape structures of the surface, the contact angle of 62° between the molten Li-Zn mixture and Cu substrate is much smaller than that of 110° between the molten Li and Cu substrate as shown in Fig. R23d. Therefore, the molten Li-Zn mixture can homogeneously spread along the Cu substrate for doctor blading at 200°C (low melting point at 174°C when the Zn element concentration is 2.7%, while many reported Li alloys need higher temperatures more than 250°C to wet the substrate).

Fig. R23. (a) The calculated ternary phase diagram of Li-Zn-Cu system at 473 K. (b) Schematic illustration of the formation of LiZn and Li_2ZnCu_3 alloys and their corresponding Gibbs free energy (ΔG_r). (c) SEM top-view image of the Li_2ZnCu_3 alloy. (d) Comparison of the contact angles between Molten Li and Li-Zn mixture on the Cu substrate and the factors influence the Li wettability with the LiZn and Li_2ZnCu_3 on the Cu substrate.

After all, we have modified Fig. 2 as shown below and supplemented the correlated discussion in the revised manuscript and Figure in the revised SI.

Pages 9 and 10 of the revised manuscript:

“Substantially, the wettability of liquid towards the substrate mainly involves two factors, including the surface tension of molten Li and the surface texture of the substrate. Firstly, the aforementioned Li-Zn alloys can effectively reduce the surface tension of the molten Li due to the weaker interaction of Li-Zn with regard to the Li-Li bonds, visibly augmenting the Li wettability on the substrate. The alloying reaction between Li and Zn at 250 °C is as follows:

Note that the released Gibbs free energy (ΔG_r) at high temperatures is estimated using the machine learning model SISO (sure independence screening and sparsifying operator) approach.³⁹”

“As shown in Fig. 2b-c,f, the formation energies of Cu|Li and Cu|Li-Zn interfaces are determined to be *ca.* -1.63 and -1.81 J m⁻², respectively, indicating the enhanced chemical contacts between Cu and Li-Zn with respect to that between Cu and Li.⁴⁰ As shown in Fig. 2d, this improved Cu|Li-Zn interface contact helps the subsequent alloying process of Li-Zn, Cu, and excessive Li at 200 °C to generate the ternary Li₂ZnCu₃ phase as follows:

In fact, this exothermal reaction among the molten Li, Li-Zn alloy, and Cu is conducive to further improving the Li wettability on the substrate upon the alloying process.”

“Thus, the exothermal alloying (negative ΔG_r) of Li-Zn and Li₂ZnCu₃ could decrease the θ to enhance the Li wettability on Cu foil, providing an extra driving force for molten Li spreading in line with previous reports.^{29,43} More significantly, the formation of the Li₂ZnCu₃ alloy can transform the morphology of Cu foil and significantly improve its roughness as shown in Fig. 2e. The contact between the molten Li-Zn mixture and Cu substrate can be regarded as a Wenzel state due to the compact interface within the Li/LiZn layer and Cu in Li/LiZn@Cu, which can further improve the surface lithiophilicity by roughness increasing.⁴⁴ Such a pyramidal-shape morphology of Li₂ZnCu₃ alloy demonstrates the abundant nano-structures to form the effective capillary effect, which also contributes to the Li wettability by Laplace pressure (ΔP ; $\Delta P = 2\sigma/r$, where the σ is the surface tension of molten Li and the r refers to the capillary radius), as shown in Fig. 2e, f.⁴⁵ Under the above synergetic effects, the contact angle of $\sim 62^\circ$ between molten Li-Zn mixture and Cu substrate is much

smaller than that of $\sim 110^\circ$ between molten Li and Cu substrate as shown in Supplementary Fig. 18.”

Fig. 2. Economic Li composite anode by alloying. (a) Calculated gravimetric energy density and waste rate of Li resource of pouch cell with different thicknesses of Li metal anodes. The obtained data is based on Supplementary Table 4. Formation energies of the (b) Cu|Li and (c) Cu|LiZn interfaces. (d) Schematic illustration of the formation of LiZn and Li₂ZnCu₃ alloys and their corresponding Gibbs free energy (ΔG_f). (e) SEM top-view image of the Li₂ZnCu₃ alloy. (f) Factors influence the Li wettability with the LiZn and Li₂ZnCu₃ on the Cu substrate.

Supplementary Fig. 18. Optic photos of the contact angles of the (a) molten Li and (b) Li/LiZn on Cu substrate.

Reference

1. Wu J, *et al.* Composite Lithium Metal Anodes with Lithiophilic and Low-Tortuosity Scaffold Enabling Ultrahigh Currents and Capacities in Carbonate Electrolytes. *Adv Funct Mater* **31**, 2009961 (2021).
2. Wu J, *et al.* Polycationic polymer layer for air-stable and dendrite-free Li metal anodes in carbonate electrolytes. *Adv Mater* **33**, 2007428 (2021).
3. Liu P, *et al.* LiBr–LiF-Rich Solid–Electrolyte Interface Layer on Lithiophilic 3D Framework for Enhanced Lithium Metal Anode. *Small Struct*, 2200010 (2022).
4. Li X, *et al.* Thickness-controllable Li–Zn composite anode for high-energy and low-N/P ratio lithium metal batteries. *J Mater Chem A* **10**, 11246-11253 (2022).
5. Zhou Y, *et al.* A novel dual-protection interface based on gallium-lithium alloy enables dendrite-free lithium metal anodes. *Energy Storage Mater* **39**, 403-411 (2021).
6. Zhang Y, *et al.* Enabling 420 Wh kg⁻¹ Stable Lithium Metal Pouch Cells by Lanthanum Doping. *Adv Mater* **35**, 2211032 (2023).
7. Wang A, *et al.* Stable all-solid-state lithium metal batteries enabled by ultrathin LiF/Li₃Sb hybrid interface layer. *Energy Storage Mater* **49**, 246-254 (2022).
8. Pathak R, *et al.* Fluorinated hybrid solid-electrolyte-interphase for dendrite-free lithium deposition. *Nat Commun* **11**, 1-10 (2020).
9. Li S, *et al.* A robust all-organic protective layer towards ultrahigh-rate and large-capacity Li metal anodes. *Nat Nanotechnol* **17**, 613-621 (2022).
10. Luo C, *et al.* Roll-To-Roll Fabrication of Zero-Volume-Expansion Lithium-Composite Anodes to Realize High-Energy-Density Flexible and Stable Lithium-Metal Batteries. *Adv Mater* **34**, 2205677 (2022).
11. Zhang G, *et al.* A 10- μ m Ultrathin Lithium Metal Composite Anodes with Superior Electrochemical Kinetics and Cycling Stability. *Energy Environ Mater* **6**, e12598 (2023).
12. Chen H, *et al.* Free-standing ultrathin lithium metal–graphene oxide host foils with controllable thickness for lithium batteries. *Nat Energy* **6**, 790-798 (2021).
13. Huang S, *et al.* Interfacial friction enabling $\leq 20 \mu\text{m}$ thin free-standing lithium strips for lithium metal batteries. *Nat Commun* **14**, 5678 (2023).
14. Zhang Y, *et al.* Enabling 420 Wh kg⁻¹ Stable Lithium Metal Pouch Cells by Lanthanum Doping. *Adv Mater* **35**, 2211032 (2023).
15. Liu Y, *et al.* Self-assembled monolayers direct a LiF-rich interphase toward long-life lithium metal

batteries. *Science* **375**, 739-745 (2022).

16. Li N, *et al.* Reduced-graphene-oxide-guided directional growth of planar lithium layers. *Adv Mater* **32**, 1907079 (2020).
17. Xu P, *et al.* A Lithium-Metal Anode with Ultra-High Areal Capacity (50 mAh cm⁻²) by Griding Lithium Plating/Stripping. *Energy Storage Mater* **38**, 190-199 (2021).

REVIEWERS' COMMENTS

Reviewer #1 (Remarks to the Author):

All the suggested points have been modified completely. It means that the revised manuscript is now ready for publication.

Reviewer #2 (Remarks to the Author):

In the revised manuscript, the author has effectively addressed my concerns and significantly enhanced the quality of the article.

However, I am still curious about why the peak intensity of LiZn was less pronounced than that of the Li_2ZnCu_3 in the XRD. Can authors provide a possible explanation?

Reply to the comments

Reviewer #1 (Remarks to the Author):

All the suggested points have been modified completely. It means that the revised manuscript is now ready for publication.

Reply:

We sincerely appreciate the valuable feedback provided by the reviewers #1 and the kind recommendation of our research.

Reviewer #2 (Remarks to the Author):

In the revised manuscript, the author has effectively addressed my concerns and significantly enhanced the quality of the article.

However, I am still curious about why the peak intensity of LiZn was less pronounced than that of the Li_2ZnCu_3 in the XRD. Can authors provide a possible explanation?

Reply:

Thanks for the comment.

Fig. R1 (Fig R1a-b is the Fig. 1h-j in the revised manuscript, Fig. R1 c-f is the Supplementary Fig. 11a-d in SI, and Fig. R1 g-j is the Supplementary Fig. 17d, i-k in SI). (a-b) Top-view SEM images, (c) cross-section SEM image and the (d-f) corresponding EDS mapping of the Li/LiZn@Cu anode. (g) Top-view SEM image, (h) cross-section SEM image and the (i-j) corresponding EDS mapping of the e-Li/LiZn@Cu anode.

To understand the weak peak intensity of LiZn in the complex alloy, the morphology is bound to this important phenomenon.

As shown in Fig. R1a-b, the dot- and rod-like LiZn alloys are distributed discretely around or inside the Li/LZn@Cu surface, rather than being densely packed throughout the surface of the alloy. In the cross-sectional SEM of Fig. R1c-f, the LiZn alloys are embedded clearly inside the Li/LiZn layer, acting as a framework. The oriented and non-consecutive morphology of LiZn alloy will demonstrate an important influence on the peak intensity of the final XRD pattern of the alloy anode.

Thus, some of the LiZn alloys are deeply buried into the entire three-dimensional space of the Li/LiZn layer. In this case, the incompact spatial distribution of LiZn alloy may lead to weakening conditions in the XRD diffraction process of LiZn alloy.

In sharp contrast, as shown in Fig. R1g-j, the 2 μm -thick Li_2ZnCu_3 alloy is fully covered on the entire Cu surface, with a dense structure. Such flat and compact morphology is conducive to the strong peak intensity of XRD.

Last but not least, the content of LiZn alloy is not high. The primary particle size is small enough and the structural stability, especially at the Li-LiZn- Li_2ZnCu_3 phase boundaries is challengeable to be characterized using XRD in the alloy anode.

Consequently, the peak intensity of LiZn is less pronounced than that of the Li_2ZnCu_3 in the XRD.